# A Survey of Hybrid Free Space Optics (FSO) Communication Networks to Achieve 5G Connectivity for Backhauling

**DOI:** 10.3390/e24111573

**Published:** 2022-10-31

**Authors:** Omar Aboelala, It Ee Lee, Gwo Chin Chung

**Affiliations:** Faculty of Engineering, Multimedia University, Cyberjaya 63100, Malaysia

**Keywords:** fifth-generation (5G), multiple-input multiple-output (MIMO), free-space optics (FSO), line-of-sight (LOS), hybrid FSO/radio-frequency (RF)

## Abstract

Increased capacity, higher data rate, decreased latency, and better service quality are examples of the primary objectives or needs that must be catered to in the near future, i.e., fifth-generation (5G) and beyond. To fulfil these needs, cellular network design must be drastically improved. The 5G cellular network design, huge multiple-input multiple-output (MIMO) technology, and device-to-device communication are all highlighted in this comprehensive study. Hence, free-space optics (FSO) is a promising solution to address this field. However, FSO standalone is insufficient during turbulent weather conditions. FSO systems possess some limitations, such as being able to be disturbed by any interference between sender and receiver such as a flying bird and a tree, as it requires line-of-sight (LOS) connectivity. Moreover, it is sensitive to weather conditions; the FSO performance significantly decreases in bad weather conditions such as fog and snow; those factors deteriorate the performance of FSO. This paper conducts a systematic survey on the existing projects in the same area of research such as the hybrid FSO/Radio frequency (RF) communication system by listing each technique used for each model to achieve optimum performance in terms of data rate and Bit Error Rate (BER) to be implemented in 5G networks.

## 1. Introduction

The next phase in mobile communications standards is fifth-generation (5G) communication. It provides new services with ultra-high system capacity, massive device connectivity, ultra-low latency, ultra-high security, ultra-low energy consumption, and extremely high service quality [1,2,3,4,5,6]. 5G communication is predicted to enable ultra-dense heterogeneous networks with a 1000-fold increase in mobile data capacity per area and a 100-fold increase in the number of connected wireless devices compared to current wireless networks [1]. As a result, future networks will need to enable high user data rates, low power consumption, and minimal end-to-end latency [1,2,3,4,5,6]. To provide hyper-dense ultra-fast access networks, high-capacity backhaul connectivity is required for 5G and beyond communications [3].

Furthermore, as the Internet of Things (IoT) idea develops, the rate at which physical devices are connected to the internet is exponentially expanding [7,8,9,10,11]. Radio frequency (RF) is currently widely employed in various wireless applications. However, the currently employed RF spectrum is insufficient to meet the growing demand for 5G wireless bandwidth and serve the IoT paradigm. Existing wireless technologies make extensive use of the electromagnetic spectrum below 10 GHz, which has nearly been exhausted; thus, it is expected that the tremendous connectivity demand of future mobile data traffic will not be supplied by existing wireless technologies [12]. Furthermore, this range (below 10 GHz) has restrictions such as a narrow spectrum band, spectrum use laws, and substantial interference between close RF access points. As a result, researchers are investigating new complementing spectra for wire-free communication connectivity, such as millimeter and nanometer waves [13]. The International Telecommunication Union presented 11 additional candidate bands for International Mobile Telecommunication-2020, i.e., 5G communication, at the World Radio Conference 2015.

Local and international authorities carefully regulate the usage of this band. Most RF sub-bands are licensed to specific operators, such as cellular phone companies, television broadcasters, and point-to-point microwave lines [14]. RF communication is used in many applications [15], the crowd in RF spectrum for the mobile data services requests new high-speed wireless network [16]. RF communication systems have limitations such as relative low data rate, poor security, and consuming a lot of power [17]. On the other hand, RF communications can handle bad weather conditions and do not require LOS connectivity.

The optical spectrum is a promising solution for future high-density and high-capacity networks. Free Space Optical (FSO) refers to wireless connectivity based on the optical spectrum. FSO-based network technologies have distinct advantages over RF-based network technologies. For communication lengths ranging from a few nanometers to more than 10,000 km, FSO systems can deliver high-data-rate services. They are suitable for both indoor and outdoor services. FSO systems, on the other hand, suffer from their sensitivity to obstacle blocking and restricted transmitted power. As a result, combining FSO and RF systems could provide a viable answer for the massive demands of future 5G and beyond communication systems.

Over the last few decades, FSO communication has been widely explored as a promising alternative to RF. In FSO, data are used to modulate a light beam in the same way it is in fiber optics. However, the light beam travels from one point to another wireless. The fact that FSO combines the high bandwidth of optical communication systems with the flexibility of wireless technologies has sparked a surge in interest in the technology.

The Near Infrared (NIR), Visible Light (VL), and Ultraviolet (UV) bands are all covered by FSO technology, terrestrial and space FSO communications, such as fiber-optic systems, often operate in the near-infrared spectrum [18]. As demonstrated in the following sections, terrestrial systems can also function in the VL [19] and UV [20] frequency bands. Indoor FSO lines, on the other hand, often use the NIR [21] and VL [22] bands, whilst underwater FSO systems use the NIR [23] and VL [24] bands. FSO detectors are immune to multipath fading (i.e., substantial changes in received signal magnitude and phase) due to the relatively short wavelengths (i.e., high frequencies) at which they operate, as opposed to RF connections, which are particularly sensitive to multipath fading. This is due to the spatial diversity created by the FSO detector regions being extremely large compared to the wavelengths [17]. Aside from the unregulated spectrum, most optical components used in FSO lines are less expensive, smaller, lighter, and consume less power than RF components, resulting in cost and energy savings [25,26,27,28]. Although most FSO components are less expensive, lighter, and smaller than RF lines, it is important to remember that FSO networking solutions are not as mature or commercially available as RF networking solutions. The cost of FSO systems is likely to decrease as the technology becomes more mainstream and as market competition increases. On the other hand, as technology advances, FSO solution designers will be able to establish the best design principles, which will have an impact on the size of the modules used in FSO systems.

Because FSO and RF do not interfere, FSO technology has also been explored to complement the existing RF systems [29]. This feature is critical for applications where RF interference must be avoided, such as hospitals and personal entertainment systems aboard commercial airplanes, to avoid interfering with RF-sensitive navigation and avionics electronic equipment [30]. Furthermore, along with RF technology, the next generations of wireless communication systems (e.g., 5G, fifth-generation (6G)) contain numerous complimentary access technologies, including FSO [31,32].

To overcome the disadvantages of FSO technology, hybrid FSO/RF communication is utilized as a substitutional nonstop method to use when the primary FSO link is under bad weather conditions. As shown in Figure 1, the sender switches between the links based on the feedback received from the receiver using a hybrid FSO/RF system which gives excellent advantages to the communication system. The system becomes highly reliable and always available for wireless communication between sender and receiver, even under extreme weather conditions. A hybrid structure of FSO with RF backup combines the advantages of both the technologies, such as low cost, non-line-of-sight applications, high data rate, and low latency [16]. The hybrid system provides dual transceivers RF and FSO by switching between them based on the weather conditions and electromagnetic interference levels. Mixing the FSO link with the RF link is a solution to provide a high-speed data rate for backhaul wirelessly. Rayleigh scattering occurs when particle size is much smaller than the wavelength, such as in snow [33]. Mie scattering applies to particles that have a comparable size to the wavelength, such as water droplets in clouds and fog. Non-selective scattering can be used for particle sizes much greater than the wavelength, such as raindrops [34,35]. In long-range RF connections, some atmospheric conditions can easily disturb the connection. Using an access point to amplify the signals through a short-range RF link to a long-range FSO link solves this problem without additional power consumption [36].

In addition, the FSO communication system solves the last mile access problem. Therefore, many researchers are interested in its field [37]. The last mile mainly characterizes the final networking part that connects the end-user to the internet. The last mile problem occurs due to funneling all the network traffic in this link, i.e., bottlenecking of the network throughput. Consequently, the bandwidth available in this link will limit the amount of data transmitted to the respective internet service provided [38]. To sum up, Table 1 summarizes the main differences between FSO and RF communication systems [39,40,41]. Most terrestrial FSO links are between 300 m and 5 km, although depending on the data rate and availability needed, greater lengths, such as 8–11 km, may be implemented [42]. On the contrary, with a power of 1 W, the typical RF link ranges with a frequency range of 400 MHz are up to 30 km, and with a power of 10 W, up to 80 km [43].

### Motivation and Contribution

For a long time, the benefits of FSO technology have been well-known. Recent researches and developments in FSO enabling technologies on the other hand, made it easier to make use of these benefits. As a result, many new FSO-related research publications have recently been released. Given that most FSO technology classification attempts occurred in the late 1990s, we feel that existing FSO technology classifications are outmoded [17,29,44].

The majority of the previous categorization attempts focused solely on reviewing and differentiating FSO systems, with little regard for the establishment of new/future FSO links. As a result, fitting some emerging and future configuration classes into existing single-level classification schemes may be challenging, if not impossible. As a result, numerous survey studies must incorporate new classes, causing the overall classification scheme to become inconsistent and non-systematic in its expansion. Consider the quasi (multi-spot) diffuse system [45,46], which, despite its similarity to diffuse systems [17,30,47,48], has been propagated as a different class. Furthermore, many recent advancements in FSO have resulted in inconsistencies, and in some cases, contradictions, between multiple classifications and definitions, such as in their naming conventions and operating principles. The terms line-of-sight (LOS), directed LOS, non-directed LOS and point-to-point (P2P), for example, all refer to the same FSO link configuration [17,29,30,44,47,49].

FSO systems are an excellent solution for existing projects to enhance their performance as a full-duplex FSO system can achieve 10 Gbps of data rate. In addition, FSO communication systems can be used in different applications for last-mile access. By comparing the FSO system with existing systems, we can conclude that the FSO system outperforms all of them, as explained in Section 3.

This article offers a comprehensive examination of hybrid FSO communication systems, which make use of a variety of approaches and links to accomplish the best possible levels of performance. In recent years, there has been an increase in the instances in which these technologies have been utilized.

The organization of this article is broken down as follows; Section 2 covers the essential conditions, in addition to covering fundamental concepts, Section 3 analyzes the performance of FSO in comparison to other communication methods for 5G backhauling, Section 4 explains multiplexing approach, Section 5 discusses communication networks that make use of relay nodes, Section 6 provides an overview of the various hybrid FSO systems, followed by Section 7 which explores projects that exclusively utilized FSO links, and ultimately Section 8 goes into details on models for the fourth-generation (4G), 5G, and 6G communication systems.

## 2. Basic Requirements and Primary Concepts

Preliminaries and basic concepts linked to optical wireless communication are discussed in this section. We address the optical wireless technology’s naming convention because it has been discovered that researchers refer to the technology by multiple names in the literature. Preliminaries and essential components of a general FSO link, such as light sources, photodetectors, and modulation methods, are also briefly discussed. The specific components utilized in optical communication systems, as well as breakthroughs in research linked to these components, are outside the focus of this work. Papers and publications on the theory of operation, variations, and advancement of various types of light sources and photodetectors are available for interested readers [50,51,52,53,54,55,56,57,58]. In [59,60,61,62], there is a discussion of eye safety and existing legislation. Furthermore, refs. [47,63,64] provide excellent summaries and taxonomies of modulation schemes in FSO.

### 2.1. Nomenclature

Optical wireless and fiber-optic communication technologies operate in the same band of the electromagnetic spectrum and have comparable transmission bandwidth capacities; hence, optical wireless communication was formerly known as fiber-less optics. New names for fiber-less optics technology arose in the literature as it advanced and was employed in new sectors, such as Lasercom, Optical Wireless Communication (OWC), and FSO. The terms “OWC” and “FSO” have become commonly used in recent decades, although “fiber-less optics” and “lasercom” are now regarded as obsolete [65].

Kaushal and Kaddoum [27] utilize the symbol FSO to refer to fiber-less optics technology in a recent classification and survey. The authors divide FSO technology into two categories: indoor systems and outdoor systems. Terrestrial Links and Space Links are two subcategories of the FSO system. Because the technique uses an unguided channel in both the terrestrial atmosphere and the vacuum, FSO refers to outside links (outer space). This is also true in indoor and underwater locations where unguided channels are used by fiber-less optical systems. It has been discovered that FSO in the underwater environment is commonly known as Underwater OWC.

### 2.2. Light Source

Laser Diodes (LDs) and Light Emitting Diodes (LEDs) are the most prevalent light sources in FSO systems because of their high optical power outputs and wider modulation bandwidths. LDs are popular in applications requiring large data rates. To mitigate potential eye and skin safety hazards, there are standards and power restrictions controlling the use of the LDs [64].

**Advantages of LDs:** Coherent lights are those in which all individual light waves are properly aligned with one another, and all waves travel in the same direction, in the same manner, and at the same time [66]. Due to its coherent properties, laser light is collimated and directed forward. As a result, laser light has the ability to traverse enormous distances. Furthermore, because of this, LDs produce less interference and deliver a higher data rate than LED lights.

**Disadvantages of LDs:** The aperture of LD is small. With LD, only point-to-point communication is possible. The concept of employing LD-based FSO communication has been around for a long time. IR LDs have already been used to show high-data-rate communication for mobile access [67]. However, due to potential health risks (e.g., eye injury, hyperthermia, coagulation, and ablation due to thermal effects of laser radiation [68]), expense and color mixing complications [69], LDs are not widely used.

LEDs, on the other hand, are favored in indoor applications with low/medium data rates. This is because LEDs are both cheaper and more reliable than LDs. LEDs are also long-lasting sources with large-area emitters. As a result, even at relatively high levels, LEDs can be operated securely. LEDs support lower data rates than LDs [70,71]. Data speeds of up to 1 Gbps have been attained utilizing LEDs and rate-adaptive discrete multi-tone modulation [72]. Tsonev et al. [69] demonstrate a 3 Gbps FSO link using a single 50-micrometers gallium nitride LED and an Orthogonal Frequency Division Multiplexing (OFDM) modulation technique in the visible light range.

**Advantages of LEDs:** Due to recent developments in solid-state lighting, there has been a tendency over the past decade to replace incandescent and fluorescent bulbs with high-intensity white solid-state LEDs. LEDs offer advantages such as great energy efficiency, long lifespan, small form factor, lower heat generation, reduced use of hazardous materials in design, and improved color interpretation without hazardous chemicals [22]. LED adoption has been steadily increasing due to these advantages, with approximately 75% of all illumination predicted to be delivered by LEDs by 2030 [73]. Another significant advantage of LEDs is their ability to instantly transition between different light levels. Because of the rapid switching rate, LEDs can be used as FSO transmitters for high-speed communication and as a highly efficient lighting source simultaneously. As a result, LEDs can perform two functions: very efficient illumination and extremely fast communication.

**Disadvantages of LEDs:** All of the light produced by an LED is incoherent. As a result, all of the waves are out of phase, and the optical power communicated by an LED is comparatively low. Natural and artificial light sources also cause LED source light to interfere.

### 2.3. Photodetector

A photodetector is a semiconductor device that converts light’s photon energy into an electrical signal by releasing and speeding current-conducting carriers within the semiconductors. The Positive-Intrinsic-Negative photodiode (PIN) and the Avalanche Photodiode (APD) are the two photodiodes that are most frequently utilized [17,64]. Only two varieties are accessible because of their good quantum efficiency, semiconductor design, and widespread availability in commercial-off-the-shelf (COTS) [74].

FSO receivers can generally be built with amplifiers for a variety of advantages, including the following:The optical preamplifier can be used to increase optical signal strength that has been diminished by different atmospheric circumstances, to successfully increase receiver sensitivity,Overcome eye-limit restrictions on transmitted laser power, andSuppress the limiting effect of the receiver thermal noise generated in the electronic amplifier.

In low-cost, low-data-rate FSO lines, PIN photodetectors are preferred since they are inexpensive, can function at low bias, and can withstand vast temperature fluctuations [17,70]. This results in a significant internal electrical gain, which raises the receiver’s Signal to Noise Ratio (SNR) [17,29]. APD photodetectors outperform PIN, especially in systems with low ambient light noise. As a result, APDs are preferred in FSO systems with high data rates and good performance. APDs, on the other hand, are more expensive and have a temperature-dependent gain. In [64] several noise sources connected to PINs and APDs are analyzed.

Recent breakthroughs in graphene, two-dimensional materials, and (nano)materials, such as plasmonic nanoparticles, semiconductors, and quantum dots, have opened the way for the development of ultrafast photodetectors that work over a wide wavelength range [75,76,77]. These photodetectors help optical communication systems with ultrahigh bandwidth and more incredible data speeds.

### 2.4. Modulation

Transmission dependability, energy efficiency, and spectrum efficiency vary depending on the modulation technique. The type of application determines which modulation method is used. For example, On-Off keying (OOK) modulation is the most widely used modulation method in FSO systems due to its simplicity. However, in more complicated systems that demand a high data rate, such as deep space communication, OOK can be wasteful. Pulse Position Modulation (PPM) or one of its derivatives, such as Variable-PPM, is commonly selected for such applications [64,78,79].

There are coherent and non-coherent optical communication systems. Non-coherent optical transmission systems use amplitude and differential phase modulations, which do not require coherent local oscillator light, whereas coherent optical transmission systems use phase and quadrature amplitude modulations for coherent detection.

Both OOK and PPM are single-carrier pulsed modulation systems. Single-carrier modulation techniques become inefficient as data rates rise due to increased intersymbol interference [80]. PPM also necessitates time-domain equalization, which can be difficult for FSO lines with severe channel characteristics and impairments [47]. Subcarrier Intensity Modulation (SIM) and Multiple SIM (MSIM) are utilized in this instance, as well as OFDM. An optical source is driven by a pre-modulated RF signal conveying the data in SIM-based techniques. Because the LD’s input must be non-negative, a Direct Current (DC) bias is applied to the signal before it is utilized to drive the optical source to maintain an entirely positive amplitude. SIM approaches alleviate channel impairments and enable a more superficial and cost-effective implementation than single-carrier modulation schemes [81]. Furthermore, SIM outperforms PPM approaches in terms of bandwidth efficiency [82].

To avoid non-negative amplitudes, the DC bias (non-information signal) is added to the pre-modulated RF signal, resulting in low power efficiency. The DC bias is required to prevent clipping and nonlinear distortion in the optical domain which may become quite large as the number of carriers increases, as in the MSIM approaches. As a result, the peak-to-average power ratio (PAPR) rises, lowering the power efficiency [82]. Another issue in MSIM approaches is the nonlinearity of the light source [83,84]. Mixed signals and Inter-Modulation Distortion originate from nonlinearity at the light source, which causes interference among the subcarriers and broadening of the signal spectrum intermodulation distortion (IMD). MSIM approaches require a minimal number of carriers to restrict transmit power and reduce IMD. However, this reduces the data transfer rate. Another way to avoid IMD is to send each subcarrier through a different optical source [85].

A PAPR reduction technique can be utilized to improve the performance of MSIM techniques by making the signal less susceptible to a nonlinear distortion [86]. Another option is to use pre-distortion or post-distortion to adjust for nonlinearities [87,88]. Hassan et al. [82] provide a thorough examination of SIM techniques as it go over the benefits and drawbacks of SIM/MSIM.

## 3. FSO Systems for 5G Backhauling Network

FSO can be deployed in many applications such as for the last-mile access when a place is not suitable for constructing fiber network, providing a backup link when the fiber network experiences failure, extending/enlarging an existing fiber network as it needs short time and can be easily deployed or backhauled as FSO systems are capable of transferring data between antenna towers and public switched telephone network with high data rate speed [89]. It can also be used for bridging wide area network access as it supports high-speed data rate for end-users and works as a backbone for trunking networks with high-speed [90], connecting between P2P links such as two buildings or point-to-multipoint (P2MP) links such as satellite to the ground [91], and in military applications as it is a secure system that can connect large areas with basic planning in short time [92].

FSO systems have a lot of advantages, such as providing high transmission speed and takes less than 30 min to be easily installed [35], can start it up with low initial investment [89], does not need spectrum license as radio and microwave systems do, secure with no security upgrade required, and super low error rate even with high data rate, as for OFC. Moreover, its narrow laser beam allows limitless FSO links to be installed in certain areas. Additionally, it is immune to radio frequency interference [93] as electromagnetic/radio magnetic interference does not affect the FSO transmission which allows huge spatial reuse and has low power usage with high bandwidth [94]. The transmission has the speed of light since the optical beam is transmitted in air [95]. Table 2 illustrates a comparison between FSO with different communication systems. We can conclude that the FSO system has all the benefits of optical fiber, microwave radio, and coaxial cable systems, such as high security and high data rate with low maintenance [96,97].

FSO systems also have their disadvantages, further explained as follows. First, is scintillation loss which is the sensitivity to temperature changes from the earth’s heat rise. This causes “image dancing” on the receiver node [35]. Second, geometric loss, or power attenuation from beam spreading [93]. The third is absorption loss, where water molecules or carbon dioxide absorb photons’ power [98]. Fourth is atmospheric attenuation, in which at 1550 nm, haze attenuation is lower than other wavelengths, but fog is wavelength-independent [99]. Another disadvantage includes FSO systems scattering when the optical beam hits a scatterer thus, reducing the beam’s intensity for a longer distance.

FSO is a viable technique for providing 5G communication at high data rates and enormous IoT connectivity. Communication systems in 5G and beyond must have the capabilities to integrate ultra-dense heterogeneous networks. Making the cells smaller is a simple but incredibly effective approach to enhancing network capacity [6]. For 5G and future communications, networks quickly evolve to contain nested tiny cells such as picocells and femtocells. To satisfy the demands of 5G, visible light communication (VLC) and light fidelity (LiFi) can deliver ultra-dense small cell hotspot services. FSO, LiFi, and VLC can also successfully handle high-capacity backhaul for 5G and future communication systems. The low power consumption of FSO technologies is a critical need for 5G [2]. FSO can connect a vast and diversified collection of 5G devices for indoor and outdoor communications and deliver secure communications as required by 5G. Improving spectral efficiency will provide extreme densification and offloading. FSO technology can provide vast connectivity with low-power LED technologies to support the IoT paradigm. When the primary FSO link is down due to inclement weather, RF/FSO communication can be used as a backup, nonstop method, as shown in Figure 1.

**Table 2 entropy-24-01573-t002:** Comparison of FSO with Different Systems [97,100].

Parameters	FSO	Optical Fiber	Microwave Radio	Coaxial Cable
Installation	Moderate	Difficult	Difficult	Moderate
Data Rate	Gbps	Independent	Gbps	Mbps
Security	Good	Very good	Poor	Good
Connectivity	P2P, P2MP short and long reach	P2P, P2MP short and long reach	P2P short reach	Multidrop short reach
Cost	Low	Very High	Very High	High
Maintenance	Low	Low	Low	Moderate
Bandwidth	Very High	Very High	Medium	High
Spectrum License	Not required	Not required	Required	Required

Fiber-optic access networks are well-suited to be extended by E-band radios (71–86 GHz). Hilt in [100] demonstrated that modern MMW radios can attain Gbps speed owing to the broad radio bandwidth made possible by the frequency allocation method [101]. The estimated lifespan of the E-band radio connections is 3–5 years. However, when optical cable reaches a radio node, the investment is not wasted. As 5G networks are spreading everywhere, 5G aims to reach gigabit-level in cellular networks. However, there are a lot of challenges that disallow 5G wireless backhaul networks to work effectively. Ge et al. [102] analyze wireless backhaul networks by using small cell, millimeter-wave (MMW) and Multiple-Input Multiple-Output (MIMO) communication technologies to attain Gigabit transmission rates in 5G networks strategies for promoting 5G wireless backhaul networks in a way that uses low energy and offers high throughput. The wireless backhaul traffic in upcoming 5G+ networks is set to be examined using two common small cell scenarios. Furthermore, two common small cell scenarios are used to examine the energy efficiency of wireless backhaul networks. According to numerical findings, in 5G wireless backhaul networks, the distribution solution uses less energy than the central solution. If the new distribution network design is used in the next 5G wireless backhaul networks, a real difficulty would materialize. However, there are solutions to overcome 5G network challenges which can be summarized as follows; distribution of cell architecture and protocols to allow wireless data forwarding in ultra-dense small cell networks, make use of MMW communications that are highly recommended as they transfer huge traffic in 5G networks, utilize cooperative small cell groups as it will decrease handover failure rate in small cell networks and applying FSO into the system which have high energy-efficient data transmission schemes [35]. Results imply that the distribution solution is more energy-efficient compared to the central solution in 5G networks, where distribution solution is all backhaul communication that is forwarded to a specific small cell base station (BS) because there is no macrocell BS to gather all backhaul traffic from small cells. Small cells resemble WiFi in many ways, but they can use cellular standards, unlike WiFi. Depending on the cell’s bandwidth, maximum output power, coverage area, user count, deployment scenario, and connection to the backhaul, small cells can be categorized as femto, pico, or micro. Table 3 below compares small cells with macrocells.

## 4. Multiplexing

Due to the increasing advancement of wireless communication systems, a dependable system that can provide larger channel capacity and data transfer rates for users is essential. MIMO systems achieve this because their multiple antennas on both the transmitter and receiver sides allow for spatial diversity and spatial multiplexing techniques. MIMO is a practical solution to avoid losing data in the channel due to fading and errors, it ensures that the receiver gains more than one version of the data sent and all of them are identical. It enhances the reliability and performance of the system with high spectral efficiency and low energy per bit [105]. The goal of this research is to compare and contrast the capacity of single-input single-output (SISO), single-input multiple-output (SIMO), multiple-input single-output (MISO), and MIMO systems. We shall concentrate on the MIMO system in this research because of its increased capacity and better data transfer rates. These characteristics of MIMO systems makes it ideal for modern communication technologies. Table 4 summarizes the main differences between SISO, SIMO, MISO and MIMO communication models.

### 4.1. Multiple-Input Multiple-Output

Fang et al. [106] deploy polar coded MIMO FSO communication systems. Polar codes are error-correcting codes to achieve the capacity of binary-input memoryless symmetric channels. In simpler words, the sender can transmit data at the fastest data rate. Moreover, encoding and decoding are accomplished at a low complexity rate [107]. Photodetectors in this project neglect ambient light. Simulation results confirms that the polar-coded multiple photodiodes technique reduces turbulence-induced fading beneficially. Moreover, polar codes perform better than Low-density parity-check (LDPC) codes. On top of that, MIMO is a technology that utilizes multiple antennas at transceivers to sustain numerous paths for data to reach the receiver. Nevertheless, Fang et al. [108] propose a decode-and-forward (DF) relay that executes an end-to-end hybrid RF/FSO multi-hop utilizing multiple pulse position modulation (MPPM) coding with a MIMO scheme as shown in Figure 2, which illustrates a one-way communication (represented by unidirectional arrows). The DF relay decodes the received signal from the source to reach the destination successfully [109]. The connection between the source and the DF relay is done in RF communication. Contrarily, both DF relay and destination are using a MIMO FSO communication. Results confirm that the complexity of serial amplify-and-forward (AF) multi-hop is the main problem and requires focused studies and analysis to enhance its performance.

However, Al-Eryani et al. [110] propose a mixed RF and hybrid RF/FSO half-duplex relay network model, where authors assume that each user has only one antenna in their device for sending and receiving packets, and a DF relay requires more antennas than the number of users X as illustrated in Figure 3. If any error occurs during the relay’s transmission, the destination will recover the lost signal from the combined signal from the source and relay. Or else; the extra users will be blocked until an active user becomes idle or look for another relay to serve him/her. On the relay-destination link, the primary link is the FSO with the technology of heterodyne detection (HD) as it has low noise, and the RF is the backup link that uses the same frequency as the user-relay link. Moreover, the system utilizes the N2 × N3 MIMO system, and the N3 “number of antennas at destination in relay-destination link” must be greater than the N2 “number of antennas at relay in relay-destination link”. The significance of this project is that there is a buffer with a maximum size of L, where L is the number of packets stored. The buffer is in the physical layer of the relay node for storing data of users temporarily until the condition of the channel becomes suitable for data transmission in the relay-destination link. Results prove that buffering in the physical layer allows the system to have more robust performance in terms of the Outage Probability (OP), error rate, and ergodic capacity. Moreover, the proposed MIMO system grants many users with low complexity.

Liang et al. [111], interrogate an asymmetric dual-hop RF/FSO system that utilizes a DF relay. Each link takes the advantage of MIMO technology in the relay node to compensate for the loss due to the fading. The relay transforms the electrical signals received by the RF antennas into optical signals using SIM to be transmitted over the FSO link. However, each link has a different number of antennas to focus on the performance of the diversity multiplexing. FSO link has photo aperture transmitters that adopt repetition coding and photodetectors receivers that occupy an equal gain combining scheme. In addition, OP, Bit Error Rate (BER), and average ergodic capacity are calculated to evaluate the performance and stability of the system. This model has excellent characteristics such as magnificent error performance, high energy efficiency, and robustness. This project is an excellent solution for future large-capacity and high-speed communication systems.

Yousif et al. [112], enhance the capacity of hybrid FSO/RF that utilizes orbital angular momentum (OAM) based on MIMO/spatial mode multiplexing (SMM) using M-ary MPPM and spatial PPM (SPPM). The results highlight new advances using OAM multiplexing that supports the high capacity for FSO and RF communication systems. The authors advance the MPPM by using the source Gaussian model and SMM by coming up with a new form of hybrid SPPM and MPPM that reduces the BER and complexity of the system. Moreover, they use SMM with OAM-Quadrature Amplitude Modulation (QAM) with MIMO scheme under weak and strong weather conditions. In addition, the numerical results confirm that the OAM-based MIMO surpasses the conventional MIMO communication systems with a long-distance between sender and receiver. Furthermore, transmitter lenses boost the OAM beams with bigger mode spacing and decrease power loss and power penalty. This project maintains perfect SNR under strong weather turbulence. The authors confirm that the SPPM and MPPM using MIMO scheme decreases BER and the complexity of the spatial mode multiplexing. In addition, MPPM boosts the capacity and data transmission of RF/FSO link. Moreover, power penalties can be improved by using OAM based on MIMO system.

### 4.2. Single-Input Multiple-Output

Shah et al. [113], examine a hybrid FSO/RF connection between a ground station (GS) and Low-Earth Orbit (LEO) satellite that uses MPSK scheme. The proposed system utilizes an adaptive-combining-based switching scheme for upload and download transmission where maximum ratio combination (MRC) occurs at both FSO and RF links as long as the FSO link is not working with high efficiency. The results show the performance of the proposed communication system in terms of OP and average symbol error rate (ASER) and compare the results with single-link FSO SATCOM and single-threshold-based systems. The results prove that the system with RF backup link has better performance than a single-hop and dual-hop FSO system. The authors consider a high-altitude platform station (HAPS) as a DF relay between the sender and receiver that enhances the performance of the end-to-end system, as shown below in Figure 4. Moreover, the authors conclude that a dual-hop hybrid system performs better than a single-hop communication system. In addition, a system with HAPS as a relay node between sender and receiver has better system performance in terms of reliability. Ergodic capacity analysis for the adaptive combining scheme will be the future work for authors to focus on in terms of OP and ASER for FSO and RF fading models obeying Malaga-M and α−η−κ−μ distributions, respectively.

The hybrid SIMO-RF/FSO communication system is introduced by Shi et al. [114], who compare it with the conventional single input single output RF/FSO (SISO-RF/FSO) communication system, as shown in Figure 5. The AF relay follows the MRC theory to process the collected signal. Nevertheless, the authors derive the OP, average bit error rate (ABER), and average capacity to inspect the project’s performance. The MRC scheme is used to analyze the information that is received by the relay’s multiple receiving antennas, which are used to collect signals from the source. The source-relay hop communicates using an RF link that has numerous receiving antennas to ensure that the transmitted signals reach relay even if one or more antennas are unable to receive the supplied data. On the other hand, the relay-destination hop uses an FSO link to communicate to send and receive optical signals. However, FSO links are susceptible to the conditions of the surrounding atmosphere, and misalignment faults have the potential to rapidly compromise the quality of optical communications. To evaluate the model’s effectiveness, the authors of this work take measurements of the OP, ABER, and average capacity. An index that measures the capability of a channel to transmit data is called the average capacity of the channel. According to the numerical findings, the performance of the system is improved by the addition of SIMO-RF links in terms of diversity gain and system capacity. On the other side, it adds another layer of complexity to the system. Because of this, the optimal range for the number of antennas to use in constructing a practical application is between two and four.

### 4.3. Multiple-Input Single-Output

Liang et al. [115], propose a system that consists of dual-hop utilizing MISO RF and also a SIMO FSO relaying system. The system has Alamouti-type space-time block coding and transmitter antenna selection (TAS) schemes between the source and the relay. The relay is an AF relay with fixed and variable gain protocols. The relay will be fixed on a mountain or on the top of a building; since supplying energy is insufficient; therefore, the relay must be constructed with less complexity in both hardware and software by using a single receiving antenna and single transmitting aperture. The authors calculate OP, outage and channel capacity, and ASER to measure the system’s performance under different conditions. The presented mixed MISO RF/SIMO FSO relaying system (MSM-RS) has superior performance to the mixed MISO RF/SISO FSO relaying system (MSS-RS) as it accomplishes lower ASER and OP under strong turbulence. In other words, MSM-RS is stronger than MSS-RS when the weather worsens. In addition, an AF relay with a fixed gain is much better than a variable gain concerning outage capacity.

## 5. FSO Networks with Relay Assist

By permitting the transmitted data to use a relay node instead of a direct route to the destination, which is greatly hampered by atmospheric turbulence, relay transmission can be used to overcome atmospheric turbulence [89]. Serial (i.e., multi-hop transmission) and parallel (i.e., cooperative diversity) relaying are the two types of relaying configurations [116]. Multi-hop relaying is commonly used to increase the range of a broadcast having a limited range. The signal is serially passed from one relay node to the next in this method. The sending node transmits the data to the receiving node and a relay node, which then retransmits the data to the receiving node in parallel relaying. This kind of transmission functions as a dispersed array of antennas and is referred to as a cooperative diversity strategy [63].

FSO researchers are embracing the techniques and procedures used in RF relay-assisted networks because the notion of relay-assisted networks is established in terms of RF technology. Researchers employed AF [117,118,119,120], DF [121,122], and detect-and-forward [123] protocols for the protocols used to convey the data utilizing the relay nodes. All-optical AF relaying is used to prevent the need for optical-electro-optical conversion at relay stations, which eliminates the need for high-speed circuits and the resulting delay [124,125,126,127,128,129]. The saturated gain optical amplifier (OA) has been proven to be effective in reducing scintillation and air turbulence [127,130,131,132]. Bandele et al. [127] concentrate on FSO OA communication systems that are cascaded. The results show that even in the lack of channel state information (CSI) and without using a complicated adaptive decision threshold, cascaded OA FSO connections can suppress scintillation. As a result, FSO links can reach further distances.

As mentioned in Section 4.1, authors in [108,110,111] have used MIMO relays in their models to assist the communication system for successful transmission with high reliability. However, in this section, we will illustrate other communication systems that also integrated one or more relay nodes into their network. Each model is significantly unique from the others, as described below.

Authors in [133,134] utilize Access Point (AP) in their hybrid FSO/RF. AP allows users to be connected to the relevant BS. Amirabadi and Vakili [133] discusses two projects; known CSI and unknown CSI at the AP. For known CSI, when a signal is received at the AP, it is reproduced and forwarded while for the unknown CSI, the received signal is amplified with a constant gain and forwarded to its destination. The AP decreases the total power consumed and enhances the network capacity, stability, and data rate of the whole system that consists of two links. The relay assists to switch between links based on the current situation. However, Bhowal and Kshetrimayum [134], propose a DF relay-based hybrid FSO/RF communication system for cellular communication with new schemes like hybrid spatial modulation (HSM) and transmit source selection (TSS). HSM with TSS is to be used for the first time in a hybrid system. The mobile user (MU) can communicate with the closest AP using RF signals. The AP can converse with the relevant BS using FSO signals because the BS is the communication system’s source, and the MU is its destination node. The AP serves as a DF relay, converting received RF signals to FSO signals and vice versa. At the same time, all nodes are working in half-duplex communication. A half-duplex relay receives signals in specific time slots and then transmits them in other time slots, which reduces the efficiency of the spectrum [5,135]. The BS consists of X number of lasers. Moreover, the AP has X photodetectors and X antennas for RF signals sent to the MU and several antennas for data transmission. The results confirm that the HSM proposed system is better than other systems by activating the corresponding FSO or RF source based on the message bit, and TSS_HSM is the best method based on performance. The authors will investigate the concept of multi-user using a hybrid FSO/RF communication system using APs in their future work.

As mentioned before, in Section 4.2 [113] has used HAPS as a relay node between the GS and the LEO. Huang et al. [136] inspect their novel framework with massive uplink connection for a satellite-aerial-terrestrial network (SATN) as illustrated in Figure 6, where a HAPS plays an essential role in this network which acts as an AF relay that aids the uplink transmission from the terrestrial user equipment (UE) to satellite, the amplified signal is converted to an optical signal by using SIM. The authors develop a space division multiple access (SDMA) technique that boosts the ergodic sum rate (ESR) in a mixed RF/FSO SATN. In addition, imperfect angular information for each UE is assumed at the HAPS to calculate the analytical expression of the channel correlation matrix, then grouping all UE into groups via a subspace-based user grouping and scheduling scheme, which groups all the UE into groups. Moreover, they present an efficient low complex beamforming (BF) technique for each UE which maximizes ESR. The results obtained, such as ESR of the SATN, illustrate how effective the proposed BF and SDMA schemes are. In addition, as the number of antennas or the transmitting power increase, the performance gets better in terms of ESR.

Sharma et al. [137] implement a DF relay in this dual-hop hybrid FSO/RF system using the architecture of selective combining (SC). FSO link is the primary default option and the RF link is the backup for the primary link if it is interrupted. Beam waist at the transmitter’s output, size of the detector, and jitter are the main factors that affect the transmission of the FSO link. The system utilizes a SC scheme to merge the data collected at the destination from both transmission links. Weather disturbance occurs at both FSO channels source-relay and relay-destination links. The presented project has great performance even when the pointing error is high due to the presence of the RF backup link. Moreover, it outperforms the individual RF and FSO systems.

In another paper, Sharma et al. [138] propose a dual-hop hybrid FSO/RF system that has a DF relay between the source and receiver that adopts selective DF relaying protocol to avoid the shortcoming of the DF relay, and there is a direct link between source and destination that uses FSO link. Moreover, the system utilizes an MRC at the receiver. The switching mechanism shifts between both sub-systems, where MRC is a simple technique that chooses the received signal with the highest SNR “strongest signal” among the received signals [139]. The FSO has the higher priority for data transmission as long as the received SNR is above the calculated threshold; the system activates the RF link once the received SNR falls below the threshold and the primary FSO link becomes on standby mode. The intensity of the laser beam at the FSO link is modulated using a phase shift keying (PSK) modulation. The authors inspect the performance of the proposed system in terms of the exact and asymptotic OP and average SER closed expressions form. The results confirm that the proposed hybrid system with MRC is much better than a single hop hybrid system and cooperative FSO system with MRC due to diversity and the benefits of switching scheme sequentially.

Furthermore, Yongzhi and Jiliang [140], investigate a dual-hop Non-Orthogonal Multiple Access (NOMA) system consisting of FSO/RF links with the aid of an AF relay that converts the optical signals received from the source into electrical signals to be sent to the RF users who utilize a single antenna for data transmission while an eavesdropper tries to attack the RF link in the communication system as illustrated below in Figure 7. The results calculate and analyze the secrecy outage probability (SOP) to evaluate the performance of the proposed system. The results confirm that the HD technique has much higher SOP performance than the intensity modulation with direct detection (IM/DD) technique. Additionally, Jamali and Mahdavifar [141], review the performance of dual-hop uplink NOMA communication that utilizes FSO/RF links for concurrent transmission. The system has dynamic order decoding to prioritize users at the destination based on their instant CSI. The authors calculate the OP of the system at each link to determine the system’s performance. In addition, the system utilizes a dynamic-order decoding scheme that prioritizes users at the destination based on their spontaneous CSI. Results show that the FSO link performs better than the RF link in terms of OP and ergodic capacity.

Wang et al. [142] build a full-duplex relay with self-interference to assist a hybrid RF/FSO system that will be investigated for the first time, with spatial diversity at transceivers to enhance the performance of the system. The model operates when the whole transmission from source to destination happens with the aid of the full-duplex relay while the relay receives the data from the source using RF communication since both of them use the same frequency, then the relay sends it to the destination using RF and FSO communication as Figure 8 illustrates. Self-interference occurs due to the simultaneous RF transmission from source to destination by the full-duplex relay. The first hop consists of RF communication with multiple apertures, and the second hop has a parallel connection of RF and FSO communication with the SC technique. SC scheme receives the signal from the link with the highest electrical SNR [143]. Moreover, self-interference is caused by the loop of the RF link at the relay. The FD relay system has twice the capacity rate of the half-duplex relay system when self-interference is removed. On the other hand, the capacity of the full-duplex relay is even worse than the half-duplex relay when the self-interference is harsh. It highlights the importance of reducing self-interference to achieve a higher capacity rate in a full-duplex relay system.

Tahami et al. [144] accomplishes building architecture with novelty; it utilizes RF and FSO communication links with an AF relay with variable gain. This RF/FSO communication system transmits data between a source and destination with the aid of the relay. IM, HD, and DD are used at the FSO link. The authors compute the OP, which indicates the quality of the communication channel [145] and BER to measure the performance of the proposed novel architecture. This project illustrates an outstanding performance over the single RF/FSO. For instance, the OP at 40 dB of SNR is 2.3 × 10^−6^ compared with the single RF/FSO system is 3 × 10^−3^. The model’s architecture consists of two AF relays in the system with variable gain to create two different isolated routes (P1 and P2). The communication between the source to relay 1 and relay 2 are different: one is FSO link and one is RF link. The collected signals will be amplified by both relays independently to be sent to the desired destination through different links. If the connection between the source and relay was FSO, then the connection between the relay and the destination will be RF and vice versa. A bandpass filter (BPF) filters the received RF signal. It proposes a new Extended Generalized Bivariate Meijer-G Function algorithm that shows an improvement over the two paths due to the spatial diversity. Additionally, the relay helps for data transmission in long-range.

Chen et al. [146] evaluate the performance of FSO/RF that has a relay node between the source and destination with energy harvesting, with pointing error loss on FSO link was taken into account and there is no direct link between source and destination using single aperture. On the other hand, the RF link between the relay and the destination utilize a single antenna which can communicate in a NLOS communication. Source and relay nodes are installed on tall buildings where the destination is a RF user. SIM-based BPSK scheme is used to convert RF signals to optical signals. The proposed FSO-based system has higher and more controllable energy efficiency than traditional RF-based systems. The performance of the system gets better when the photodetector responsivity increases.

Najafi et al. [147], inspect a system which consists of parallel FSO/RF links, where the system has multiple relay nodes as shown in Figure 9. The main advantage of this network is that it can avoid the challenges caused by FSO and RF systems. Since it is utilizing both of them at the same time, the network complements FSO systems which have low reliability and RF systems which have low data rates for example. The authors consider unidirectional transmission from the source to the destination through M relay nodes since M is the number of the relay nodes {1, 2, 3, …, M} and there is no direct link between source and destination. Each FSO link employs OOK with IM/DD and Gaussian signaling for the RF links with SISO transceivers. The source has more than one aperture that transmits light towards the relay. Relay has a photodetector towards the source to detect the transmitted light and an aperture towards destination to retransmit the light. The authors have used a buffer-aided (BA) relay which can store received data from the source to be sent to the destination when the weather condition becomes ideal for data transmission and non-buffer-aided (non-BA) relay which forwards the data immediately to the destination [148]. The future work will focus on implementing a bidirectional communication via a single BA relay as the BA relay has better performance than the non-BA.

Odeyemi and Owolawi show the error performance of a cooperative diversity of a communication system that is mixed of FSO and RF links that utilizes MRC at the destination [149]. Additionally, they derive cumulative distribution function (CDF), moment generating function (MGF), and ABER of the total SNR of the proposed system to evaluate its performance and reliability. The results prove that the system’s performance increases in terms of reliability by increasing the number of parallel relay nodes between source and receiver.

Zedini et al. [150], investigated the performance of an asymmetric dual-hop AF relaying system that consists of both RF and FSO links that work together to transmit the signals successfully from the sender to the receiver over both channels. The relay node between both hops plays an important role by converting RF signals into FSO signals by multiplexing and combining the multiple RF users to be sent over the FSO link. The authors develop a new uniform expression regarding the CDF, probability distribution function (PDF), the MGF, and the moment of end-to-end SNR of the whole communication system in respect of Meijer’s G function. Based on the previous expressions, they developed an exact closed-form expression for the OP, the higher-order amount of fading, and BER of various binary modulations in terms of Meijer’s G function. In addition, a specific expression for end-to-end ergodic capacity regarding the bivariate G function is derived. Moreover, they derive new asymptotic results for the previous expressions mentioned above. Results demonstrate that the performance decreases when the pointing error and/or the weather conditions become grievous.

Upadhya et al. [151], study the effect of in-phase/quadrature-phase imbalance on asymmetric double-hop RF/FSO AF two-way relay (TWR) communication system with multiple co-channel interferers at the relay node. An expression of outage probability and achievable sum rate illustrate the proposed system’s performance, where the results clarify that the system is reliable and stable. However, Sarker et al. [152] test the privacy performance of a dual-hop FSO/RF that utilizes a DF relay. The one-way communication system consists of four essential nodes; source, relay, destination and eavesdropper, as shown in Figure 10. The source-relay is a RF link. However, the relay-destination and relay-eavesdropper are FSO links, where the eavesdropper is somewhere near the destination and tries to overhear the optical data transmission from the relay. The system’s performance is studied in terms of SOP, strictly positive secrecy capacity (SPSC), and intercept probability. Moreover, HD and IM/DD are compared together, and the results show that HD has a better performance than IM/DD.

Alathwary et al. [153], propose a multi-hop hybrid FSO/RF system that utilizes DF relay nodes. The FSO links have the higher priority in all the hops over the RF link for transmitting data as long as the weather quality is above the acceptable level. Nevertheless, data transmission is switched to the RF link if the next node sends back a 1-bit feedback RF channel. The system’s performance is inspected in terms of end-to-end outage probability and end-to-end ergodic capacity which are derived in closed-form expressions under different weather conditions. Results confirm that multi-hop hybrid communication systems have great performance under different strong weather conditions and strong pointing errors. As jitter increases, the system’s overall performance decreases. In addition, placement of the relay node and optimizing the required number of hops will be inspected for multi-hop hybrid FSO/RF system in future work.

Torabi and Effatpanahi [154], present an analysis of the performance of two different FSO/RF systems that employ AF relay with SC technique at the receiver side. The first one which is the “RF–RF/FSO” as shown in Figure 11, the first hop between the sender and relay utilizes RF link and the second hop between the relay and the receiver has RF and FSO links. The second one is “RF–RF/RF–FSO” as shown in Figure 12. It has two relay nodes; links between the source and relays are RF links. However, one link is FSO and another link is RF between the relays and destination. Furthermore, the FSO link utilizes IM/DD and heterodyne detection techniques. The authors examine the system performance in terms of closed-form expressions for the outage probability, average BER and ergodic capacity after deriving expressions for PDF and CDF of the received SNR of both systems. The numerical results confirm that both systems have better performance compared to the classical hybrid RF/FSO systems that employ AF relaying. Additionally, the “RF–RF/RF–FSO” model accomplishes the highest performance compared to the RF/FSO and RF–RF/FSO systems as it has two independent paths to the destination.

Amirabadi and Vakili [155] investigate the performance using OP and BER of a multi-user dual-hop FSO/RF communication system assisted by a fixed gain or adaptive gain AF relay. This system allows users to connect to the respective BS through the relay using RF signals. On the other hand, the relay communicates to the BS using the FSO link. For the first time, the number of users on the performance is studied in negative exponential atmospheric turbulence for a dual-hop hybrid FSO/RF system. Results obtained show that the adaptive gain relaying system has a better performance compared to the fixed gain relaying as it is less sensitive to the number of users which is more applicable in areas with a high population. However, the fixed-gain scheme is less complex but uses a lot of power. To sum up; areas with saturated weather turbulence are recommended to use adaptive gain relaying because of the importance of power consumption and the number of users. Nevertheless, fixed gain or adaptive gain relaying could be implemented at moderate to strong weather turbulence based on the system performance requirement.

Tonk and Yadav [156], inspect the performance of DF relay in asymmetric 5G RF/FSO systems. This dual-hop system uses RF communication in the first hop between source and relay and FSO communication between relay and destination where there is no direct link between source and destination. The results of the proposed model’s behavior verified its accuracy concerning OP and received SNR, which can be beneficial for future wireless communication networks. Moreover, Mogadala et al. [157] proposed a new hybrid FSO/RF communication system that consists of dual-hop with DF relay with no direct link between sender and receiver. This proposed system can be used for 5G wireless backhaul as it meets the requirements of the 5G communications such as capacity, latency, reliability, energy efficiency, and cost-effectiveness that can work with high performance compared to other systems such as FSO and RF standalone system. As shown in Figure 13, the DF relay is placed between the source and destination and utilizes dual transceivers for FSO and RF links to decode and forward the received signals from source to destination over the more suitable channel. The SNR value at destination determines which link will be used for data transmission. If the received SNR falls below the minimum threshold of the FSO link, the sender will switch from FSO to RF till the received SNR goes above the minimum threshold again. The received SNR information is sent to the transmitted over a dedicated feedback link for taking the right decision based on the information received from the receiver. This proposed model can be a valuable solution for the existing backhaul systems to have a reliable communication system.

Roumelas et al. [158], study hybrid FSO/MMW subsystems using BPSK modulation and employing a selection combining scheme, which is serially connected to each other using DF relays. They study and investigate spatial jitter and fading characteristics simultaneously in improving performance in terms of outage probability and BER, and boosting the link distance using a relaying scheme. Results assure that availability and reliability increase by using a hybrid scheme as the presence of DF relay nodes increases the effective link length. This system can be easily used for backhauling 5G/5G+ networks. Since fiber and FSO links can provide high bandwidth, they are suitable to implement 5G networks with low latency and high stability. However, they need proper modeling. Morra et al. [159], present the analytical results of their extended work of mixing FSO with fiber links that utilize AF relay for 5G backhauling systems with considering RF co-channel interference, FSO pointing errors and scintillation, and fiber and FSO modulator nonlinearity are modeled to be considered. The system’s performance is measured in terms of OP, average BER and CDF. The RF/FSO/fiber hybrid system can allow 5G access for end-users. The results in this experiment confirm that there is an optimal average launched optical power into the fiber that gives an outstanding performance. These results provide help for the design of 5G backhauling networks.

Lu et al. [160], implement and design a bi-directional fiber-FSO-5G MMW/5G NR sub-THz convergence with parallel/orthogonally polarized dual-carrier mechanism as shown in Figure 14. The 40 Gbps download data transmission uses four-level pulse amplitude modulation (4-PAM) combined with 100 GHz 5G NR sub-THz 4-PAM signals. The downlink transmission of x- and y-polarized 40 Gbit/s/100 GHz 5G NR sub-THz signals is successfully accomplished via a parallel/orthogonally polarized dual-carrier technique. Phase modulators and remote injection locking distributed feedback laser diodes (DFB LDs) successfully construct the uplink transmission of 10 Gbit/s/28 GHz and 10 Gbit/s/24 GHz 5G MMW signals. To summarize, 5G NR sub-THz and 5G MMW links are used for BS communications for downlink and uplink, respectively. Moreover, the 500 m FSO link and the 20 km single-mode fiber (SMF) link allow the 1 m RF link to reach the core network successfully. Results confirm that the proposed bi-directional system can achieve high-speed and long-reach transmissions with high stability under different weather conditions.

As the number of small cells incredibly increases daily to achieve 5G wireless technology, the challenges increase to achieve effective 5G traffic backhauling. Song et al. [161], investigate the practicality of a multi-hop 5G backhaul system that utilizes FSO connectivity to attain a high data rate with high reliability and resource-efficiency to soothe network traffic by focusing on boosting the throughput with the aid of relay nodes. The proposed system uses IM/DD and employs binary pulse position modulation (BPPM). Moreover, the authors propose an algorithm that uses back-pressure theory to improve transmission efficiency via a DF relay selection. In addition, another algorithm is used for link scheduling that boosts resource utilization by reusing idle time slots on FSO links. The simulation results illustrate that the proposed algorithms can remarkably strengthen the performance of FSO-backhaul systems in terms of reliability and resource utilization, especially in comprehensive data transmission. Their future work will focus on decreasing the delay in the system. Additionally, Trinh et al. [162] studied a MMW RF/FSO system assisted with an AF relay node. This project is a highly upgradeable and cost-effective solution for 5G cellular backhaul networks. In addition, novel precise closed-form expressions for CDF, PDF, and MGF are derived in terms of Meijer’s G functions. All the numerical results evaluate the system’s performance in terms of OP, ASER and average capacity at high SNR under different weather conditions. The results confirm that the proposed system could be a scalable, cost-effective and high-capacity backhaul solution for 5G mobile networks. Moreover, pointing errors with strong atmospheric turbulence downgrades the system performance.

Pattanayak et al. [163] represents a DF TWR communication with a physical layer since eavesdroppers can hear the transmitted data by the nodes. The signal will begin its journey from the source to its destination through the TWR, as shown in Figure 15. The communication must be secure and confidential. The source has a specific antenna for data transmission in optical signals, while there is another transceiver antenna at the relay to build communication with the source. In addition, the RF antenna supplies the relay for transmitting signals to the destination over RF communication. Relay can convert the FSO signals to RF signals and vice versa. IM, HD and DD are analyzed separately to measure the effect of detection. All nodes are operating in half-duplex mode. The authors analyze three scenarios; E attacks the RF links, attacking FSO links, and attacking both. The analysis illustrates that as the number of eavesdroppers increases, the less secure the model becomes. Moreover, Scenario 3 is the worst of the other two scenarios. Nevertheless, HD is much better than IM/DD for optical signal detection for the three scenarios. The system performance can be enhanced if all nodes are working in full-duplex instead of a half-duplex communication.

Jiang et al. [164] propose an end-to-end hybrid FSO/RF system integrated with PPM, BPSK and SIM schemes. The end-to-end mixed RF/FSO system contains three important parts; source, destination and a relay between them. RF link receives the signal and the relay converts it into an optical signal utilizing the technique of SIM, the transformed signal is going to be amplified by a fixed gain and then transmits to the desired destination through the FSO link, and impulse modulation and direct detection are used at the destination to decode the original data sent by the sender. A Hybrid PPM-BPSK-SIM system boosts the performance of a mixed RF/FSO system in terms of OP, BER and average channel capacity under extreme pointing error and powerful turbulence conditions.

Bag et al. [165] utilized a parallel connection for the system, both FSO subsystems with an additional RF link as a backup link to continue the process of transmitting data to the receiver till the FSO link gets back on track after the weather gets clearer. Using RF and FSO together provided an enormous number of advantages to the system to minimize the loss of data during unfavorable climate conditions. FSO has a better data rate in a clear LOS path. Moreover, due to the changes in weather such as fog and rain, the RF link was used as a backup for the primary link to continue transmitting data. The FSO link is primary, and the RF-FSO link was the backup to recover when the primary link was not functioning with high efficiency. As shown in Figure 16, the system transmits the data through the FSO primary link. If the channel is unclear and unstable due to the weather conditions, then the data are transmitted from the sender to destination using RF hop to relay. The relay is responsible for converting RF signals to optical signals and then straight to the destination using the FSO hop. This whole path is called the RF-FSO backup link. If the weather becomes clear and suitable for data transmission, the primary link is reactivated. The overall performance of the proposed system is higher and more stable than the conventional single FSO systems in all weather conditions. Adding parallel MIMO RF links to the FSO links in this project can increase the overall performance.

In another approach, the source sends the data using RF signal to the destination and a quantize-and-encode relay (QER) over the FSO link in [166]. However, the relay-destination link could be either RF or a hybrid FSO/RF link, as shown in Figure 17. The relay detects the signals transmitted from the source to re-modulate it using OOK of the laser beam to be forwarded to the destination. Both relay and destination have signal processing units for RF and FSO. The main significance of this project is that the authors deploy QAM instead of BPSK, which enhances the efficiency of the bandwidth in networks. Moreover, forward error correction (FEC) and bit interleaved coded modulation with iterative decoding are deployed in the relay-destination link. This system is cost-effective and does not waste the RF spectrum. In addition, optimized symbol mapping achieves full diversity even when the FSO link faces unclear weather. Nevertheless, Al-Eryani et al. [167] study the accomplishment of a two-way multi-user mixed RF and FSO system with a relay; the system schedules user opportunities and has asymmetric channel turbulence. The users use RF signals to communicate with the relay. On the other hand, the communication between relay and BS is done in FSO signals. The authors derive a closed-form expression from calculating the outage probability OP for high SNR, average symbol error rate, and average ergodic channel capacity while a heterodyne detection scheme is assumed in the system. Moreover, they compare the performance of TWR and one-way relaying (OWR) networks. Additionally, they examine the effects on the whole network performance of parameters such as the number of users, pointing errors, atmospheric turbulence conditions, and outage probability threshold. The results confirm that the TWR network has double the ergodic capacity compared to the exact outage probability of an OWR network. Moreover, while weak to moderate weather conditions and few pointing errors, the OP at the RF link does not change. However, with severe pointing errors, the OP increases at the FSO for data transmission. The authors confirm that adding a backup RF backhaul link or providing a relay node with a physical layer buffer increases the reliability of a mixed system as the relay can store multiple received signals until the link returns to service again.

Kiran et al. [168] propose a hybrid FSO/RF communication system to provide reliable communication, especially when the link between transceivers is misaligned. The model consists of two media for transmission. The placement of the RF and FSO channels is in parallel. The relay switch plays a vital role in the system by switching from the primary FSO channel to the backup RF channel in case the weather interrupts the transmission. The hybrid model transfers large files between two computers under link misalignment conditions. Moreover, the chosen file by the computer will be transmitted serially by RS-232 protocol. Based on the weather conditions and the availability of both links, the relay will switch between RF and FSO link to transmit the data. The FSO link has different modules like trans-impedance amplifier (TIA), low-cost LASER/LED of 670 nm wavelength for transmitting and ST-IKL3B photodetector. The file is converted from the RS-232 to TTL logic and vice versa by the MAX232 serial interface. Data are then modulated to Laser/LED accordingly by IC 7405 driver to represent the data. The received data become weak, they pass to the TIA, and then to the level converter. A Zigbee transceiver transmits and receives data. The photodetector detects the voltage, the LM358 amplifier compares it with the threshold voltage. On the receiver side, HT12D decoder is used to decode the information to switch the relay between the two paths. When the misalignment happens with the FSO link, the backup RF path starts to transmit data using a 2.4 GHz Zigbee transceiver. This transmission is done by the relay when the received voltage is below the threshold value. The FSO path becomes in a standby mode till the misalignment is over and the received voltage rises above the threshold value.

Almouni et al. [169] study the work performance of a dual-hop system mixed FSO/RF by deriving the OP and average symbol error probability at a high SNR regime; the communication system has an AF relay with only one photodetector that converts the FSO signals from the source that uses multiple apertures to RF signals to be sent to multiple users “destination” since there is no direct link between the RF users and the source “BS” as shown in Figure 18. The authors investigate how well a mixed FSO-RF multi-user relay network performs in the presence of Poisson field interference using aperture selection and opportunistic user scheduling. The user who has the highest SNR among those who are available is chosen to receive the signal from the relay node via opportunistic scheduling. The relay can use HD or IM/DD detection schemes to detect the transmitted optical beam. In addition, due to the multiple receptions at the relay, the relay utilizes maximum ratio transmission. The FSO link between the optical source “BS” and relay follows the Malaga-M fading model with considering pointing errors at the first hop. The RF link has shadowed κ-μ models for LOS and NLOS propagation and opportunistic user scheduling with a Poisson field interference. Results confirm that the proposed formulas for position and power allocation improve the performance in terms of OP.

Nguyen et al. [170], establish an extensive analytical hybrid FSO/RF model based on a cross-layer approach that utilizes Transmission Control Protocol (TCP). Since that, to avoid error in data transmission, the authors employ Reed–Solomon code and Selective repeat automatic repeat request (SR-ARQ) in the proposed project. TCP is implemented for the first time in hybrid FSO/RF. Figure 19 below illustrates the whole connection of the end-to-end network. LEO satellites play an important role in this network, LEO satellites communicate all together using FSO communication and the LEO satellite network acts as a bridge by allowing the end-user to have either the RF or the FSO links to access the internet. In addition, forward error correction FEC, SR-ARQ, and TCP variants (including NewReno, Hybla, Cubic, and High-speed TCP (HSTCP) occur in physical, link and transport layers, respectively. Moreover, the link between the internet and the LEO satellite network is RF communication. The results prove that Cubic TCP is the most convenient throughput performance. However, NewReno is the worst among the four TCP variants. The authors illustrate utilizing TCP in terms of transmission errors at last-mile links, where the results clearly show that Cubic has the highest performance among the TCP variants mentioned above. Moreover, HSTCP and Hybla could be used. However, NewReno is insufficient for high-speed satellite communications specially in strong weather conditions.

Amirabadi et al. [36] present a novel multi-user multi-hop hybrid FSO/RF system that utilizes Differential Binary Phase Shift Keying modulation. The proposed system consists of two main parts; the connection between the mobile user and the source BS and between the source and the receiver BS. Since the mobile user connects to the source BS through a long-range link, thus an AF relay is established to aid the connection of the first part. Nevertheless, a demodulating and forward relaying is in the second part to serve the connection over a multi-hop hybrid parallel FSO/RF link. However, authors derive exact and asymptotic new closed-form expressions for outage probability and BER for the proposed project using amplify and forward relay and decode and forward relay with considering the atmospheric turbulence and pointing error effect with opportunistic selection schemes at each relay to get more accurate results. It has the benefits of FSO, RF, relay-assisted, and multi-user communication systems simultaneously. This project can be used in areas with high population and long-distance range links as the results confirm that the system is not affected by the number of relays or users. Moreover, it does not consume a lot of power with low latency and low complexity. From the simulation results, unknown CSI has better performance than known CSI as the amplification gain is established by choosing it manually considering the worst-case scenario.

In conclusion, Table 5 summarizes all relay assisted models in terms of the maximum data rate, fading distribution, wavelength and frequency utilized in FSO and RF links respectively, the type of relay node used in the model, and the simulation software used to verify the results. NDA and Na respectively indicate no data available and not applicable, respectively. Additionally, GG in the table means the gamma-gamma distribution.

### Unmanned Aerial Vehicle

The aforementioned methodologies all make the assumption that relay nodes are buffer-less and stationary. Moving UAVs outfitted with buffers are employed by Fawaz et al. [172] to act as relay nodes in a relay-assisted heterogeneous network that includes both fixed and moving relay nodes. Liu et al. [173] examine an uplink transmission in a SATN since an aerial platform operates as an AF relay that helps the communication between the multi-users and satellite, as shown before in Figure 6. Users communicate with the relay using RF transmission. However, communication between the relay and the satellite is a FSO link. The authors propose a BF technique that boosts the minimum average signal-to-interference-plus-noise ratio (SINR) of the users. The performance of the communication network is measured by the outage probability of the SATN. Results show that the proposed algorithm improves the performance and robustness of current systems by designing a BF scheme that boosts the average SINR of the users. Nevertheless, Aghaei et al. [174] compare the Malaga-M distribution with the gamma-gamma distribution in terms of the switching threshold, transmission rate, frame error rate, and OP. The Malaga-M distribution is an advanced and all-inclusive model announced by the University of Malaga-M to appraise FSO/RF hybrid systems under different weather conditions with two different modes, classic adaptive rate transmission and a combination of adaptive rate transmission and automatic repeat request. Results show that the Malaga-M distribution performs better and is more comprehensive than the gamma-gamma distribution for all turbulence conditions. On the other hand, gamma-gamma is only accepted in only weak, moderate, and strong turbulence conditions.

Wu and Kavehrad [175], establish a model that combines FSO and RF channels that uses QAM modulation. As shown in Figure 20, there are two types of links in this communication system; ground-to-air links (such as A–C and B–E links) and air-to-air links (such as links between C, D, and E), where the airborne vehicles are roughly 2000 m away from the ground. A and B are the sender/destination nodes. However, C, D and E are relay nodes in the communication system. The authors focus on ground-to-air links; each link consists of RF and FSO sub-links. RF links operate at 15 GHz, and FSO links use a 1550 nm wavelength. The probability distribution of capacity is calculated for both channels. The results confirm that using only FSO links is unreliable as FSO communication is easily affected by clouds and achieves diminutive availability. On the other hand, RF links are resistant to clouds. In this proposed system, the authors focus on testing the availability of both links by taking some factors into consideration such as scattering, absorption, scintillation, etc. Moreover, multiplexing can enhance FSO channel capacity and availability for thin clouds. In addition, utilizing a backup RF link is a great solution to be used during the FSO downtime. However, the capacity of the overall system is limited due to the bandwidth limitations on the RF link.

Lee et al. [176], investigate a dual-hop mixed FSO/RF system with a mobile DF relay aid by an unmanned aerial vehicle (UAV) that has a fixed buffer size, controls throughput delay and UAV mobility. The system has a FSO link between source and R, and a RF link between destination and relay as shown below in Figure 21, where both links have different transmission rates. Based on the delay requirements, there are two scenarios; limited delay transmission and tolerant delay transmission. The results illustrate that the model attains 223.3% throughput gain in contrast with the traditional static relating systems. Furthermore, the proposed algorithm clearly shows its superiority over conventional schemes as it works in full-duplex. The packets in the queue work in a first-in-first-out scheme as it transmits the packets that arrive earlier in order [177].

Pai and Sainath [178], analyze a hybrid PHY layer (RF or MMW or FSO) link switching policy that has tethered UAV that improves the end-to-end performance of all links between BSs and user equipment as UAVs have significant advantages such as mobility and flexibility. The authors propose a simple novel hybrid FSO/RF assisted with CSI-free UAV. An outage probability analysis is developed to examine the performance of the proposed project. UAV selection policy by choosing the most suitable UAV among many UAVs, which will reduce the power loss and choosing the suitable link between FSO and RF based on their outage probability, is a novel contribution. However, the proposed model works in half-duplex communication. By comparing selection combining and threshold combining together, results confirm that selection combining is more reliable and more stable in terms of performance than threshold combining. However, threshold combining could have better performance by carefully choosing the switching threshold. Their future work will cover developing the gain of the relay with various combinations schemes of the PHY layers, more studies about the BER, in-depth study on the interference and more performance analysis.

Gu et al. [179] test two different network configurations for FSO-based fronthaul/backhaul in 5G+ where all the connections are using either fiber, FSO or RF/FSO. It is assisted with networked flying platforms (NFPs) such as UAVs and drones. The first one is the proactive network configuration which optimizes the network topology for a specific time based on the statistics of the network. Secondly, reactive reconfiguration adjusts the topology based on the traffic demands or the failure of the link. Both proactive and reactive reconfiguration algorithms are compared together in terms of network throughput and cost. Simulation results prove that the proposed greedy matching based algorithm that selects the active links in order to build the best topology of the FSO-based platform improves the throughput of the network and also reduces the cost. Furthermore, the results confirm that the reactive topology reconfiguration algorithm for link failures and the reactive topology reconfiguration algorithm for traffic events boost the performance when a failure occurs and traffic is noticed. Additionally, Alzenad et al. [180], investigate the practicality of a novel vertical fronthaul/backhaul 5G+ networks with the assistance of NFPs that transmit data between the access and core networks by using P2P FSO links. The authors in this article describe any floating and moving unmanned flying platforms as NFPs such as UAVs, drones, balloons, and high-altitude/medium-altitude/low-altitude platforms. All connections in the entire network are wireless, which is one of the main essentials of the 5G architectures [181]. The performance of the proposed system is investigated in terms of link budget and data rate achieved. Simulation results prove that by decreasing the divergence angle, the system’s performance is remarkably improved. The main advantage of using FSO and NFPs concepts is that the cost of the system decreases as the cost of a vertical network is so expensive. Moreover, Safi et al. [182], use a small-scale UAV in UAV-based FSO links as an access link using MMW/WiFi for high traffic demand network. The proposed system utilizes quad-detectors that consist of four APDs that work with IM/DD and OOK signaling schemes. The authors derive closed-form expressions of bit error rate and tracking error to evaluate the system’s performance. This project can be used for the 5G network and front hauling to achieve the 5G requirements.

Erdogan et al. [183] propose a new model that merges cognitive radio (CR) enabled RF and FSO together for aerial relay networks. CR is used for ground-to-air transmission with various advantages such as spectrum efficiency, multi-user connection, and spatial diversity. However, the air-to-air transmission uses FSO communication since there are no obstacles between the nodes in the air and the FSO link requires line of sight communication. Lastly, air-to-ground links are a hybrid RF/FSO communication; RF link in hybrid RF/FSO works as a backup for the primary FSO link when it experiences terrible weather conditions. The proposed system successfully fully uses the frequency spectrum even with RF problems such as spectrum mobility and under-usage. Moreover, multiple aerial vehicles ease emergency communication, such as the occurrence of earthquakes. The whole communication system has many advantages such as a long battery lifetime, high performance, Efficient Spectrum Usage, Cost-Effective and Inherent Security.

Shah et al. [184] employ a cloud radio access network (C-RAN) cost-effective and upgradable network that uses NFPs, which are used as fronthaul hubs that connect the small cells to the core network via baseband unit pool; all the NFPs are connected using FSO links. However, they use MMW links to connect to the small cells. In addition, the authors use two greedy algorithms. The first algorithm, called Centralized Maximal Cells Algorithm (CMCA) is used when the small cells have limited processing power, and the second one, called Distributed Maximal Cells Algorithm used when the small cells have enough processing power because it is faster than the CMCA. Both algorithms are used for C-RAN architecture, aiming to increase the number of small cells. Moreover, performance results confirm that both algorithms can be implemented due to their low complexity compared to the branch and bound method. However, all of them have almost the same performance.

## 6. Hybrid FSO Networks

In this section, we will highlight different models that utilize various sub-systems with FSO communication systems. To fully utilize the benefits of hybrid FSO systems, various issues must be resolved. The transition and realization of seamless mobility for mobile users is one of the major issues. A mobile wireless light-based communication system called LiFi is a fast, bidirectional network. For instance, in a LiFi-WiFi network, a user should be able to effortlessly switch between LiFi cells and WiFi networks (vertical handover) as well as between LiFi and WiFi networks. The general characteristics of heterogeneous (LiFi + WiFi) networks are presented by Ayyash et al. [185], who also create a framework for the coexistence of LiFi and WiFi technologies. The network consists of optical small cells, RF macrocells, and RF small cells. A hybrid FSO/RF system is discussed in [185], LOS/Short link models make up each of the LiFi luminaires (lights). It should be mentioned that, instead of using cables to connect the many BSs in a big room, LOS FSO links can be used. This network can accommodate a network of nodes outfitted with LiFi receivers, which makes it a great fit for the IoT concept.

Makki et al. [171] examine the performance of a multi-hop and mesh network with short and long codewords that consist of MMW RF and FSO links with and without Hybrid automatic-repeat request (HARQ). The RF link follows the MMW characteristics, and FSO link utilizes HD technique while a DF relay is used in the network. The performance is measured in terms of outage probability and ergodic achievable rates based on different parameters, including the number of antennas and various consistency times of both links. The results support the minimum number of antennas required for the RF link to provide the same transmission rate at the RF/FSO hops. Furthermore, HARQ improves energy efficiency and reliability. Several RF antennas do not affect the OP in multi-hop networks. Nonetheless, it affects the ergodic rate as, when the number of antennas increases, the ergodic rate increases. Makki et al. [186], study both cases with and without HARQ in hybrid RF/FSO link while considering the CSI is ideal at the destination, derive an expression for the probability of decoding the message, OP, and throughput of the RF/FSO system in different scenarios. Moreover, they study the effect of adaptive power allocation between both links on throughput and OP. Lastly, they examine the performance of the RF/FSO system in different conditions of the channel to compare them with RF and FSO link solely. The signals are encoded over parallel FSO and RF links, FSO link uses IM/DD while RF utilizes MMW wavelengths varying from 30 to 300 GHz of the carrier frequency. Both FSO and RF links are working individually at the same time to transmit data to the receiver, RF signal is down-converted to the baseband and the FSO signal is gathered by an aperture to be transformed into an electrical signal by a photodetector at the receiver. The decoder is used to decode all the received signals to get the original signals. HARQ, adaptive power allocation and data diversity increase the performance and efficiency of the system to recover the lost data of the link that experiences some difficulties by referring to the other link. The authors will focus on block error rate analysis for FSO/RF links using finite-length codewords in their future work.

Inter-cell interference is unavoidable given the growing number of FSO cells deployed for coverage. In the RF domain, inter-cell interference coordination and mitigation methods have long been researched [187]. The successful strategies utilized in the RF domain are being applied by researchers in the FSO domain [188,189,190]. It is crucial to comprehend how to manage the interference between FSO links since FSO technology is becoming a feature of future hybrid networks, expressly to relieve the spectrum congestion caused by RF interference. Usman et al. [191], present and analyze a low complexity hard switching system for hybrid FSO/RF systems. Either FSO or the RF links will be activated for a specific duration, while the FSO is the primary link to be activated as long as the weather is clear for high data rate transmission. In other words, if the link quality is higher than a specific threshold. The decision to activate the more suitable link is based on the threshold of the received signal. If the FSO link is not suitable, the system will automatically switch to the RF link. Using only one link at a time will reduce the consumed energy at the source and will eliminate the use of multiplexing or combining schemes at the destination. Both FSO and RF links obtain OP, BER, and capacity. These results illustrate that the hybrid FSO/RF system accomplishes higher performance and reliability compared to a system that utilizes only the FSO link. Having both FSO thresholds will not downgrade the system performance, but it will also enlarge the lifespan of the FSO link. Both links are working parallelly to allow smooth implementation at the receiver. While the communication occurs only at one of the links, the FSO link which is the primary link of the system will be deployed as long as the link quality is sufficient. The RF link will be used immediately if the threshold is reached to continue data transmission until the received signal exceeds the threshold again. FSO link is utilizing IM/DD with a quadrature modulation scheme. Numerical results clearly show the improvement of using a hybrid scheme. In addition, the dual FSO threshold has the same performance as the single FSO threshold, but it increases the lifetime of the FSO communication link. 

Zhang et al. [39], utilize the switching system between the FSO link and RF. The switching process is determined after analyzing each path based on the threshold previously calculated. Hence, the data will be transmitted using the most suitable path, FSO or RF. The model utilizes the FSO and RF paths to avoid losses caused by the FSO link. Furthermore, the authors evaluate the BER to observe how reliable the system is. In the hybrid FSO/RF model system, based on the climate conditions, the threshold value will be computed to decide which transceivers will be deployed by the switch. The FSO path consists of a pair of FSO transceivers. The FSO transceivers are the primary path to be used in normal conditions to sustain a full-duplex communication, where the RF link is used as a backup link when the FSO link is under rough weather conditions such as fog and rain. Nevertheless, poor setting and equipment are avoided by a proper analysis to keep the system reliable all the time by avoiding delay and any failure. The threshold value in this system is 6dBm, where the system switches to the backup link when the received SNR falls below 6dBm. However, the system does not switch back to the primary link immediately. A 3dBm margin is set to switch back to the FSO link, which means that the received SNR at the FSO link is monitored to activate it back when it is 3dBm higher than the threshold.

Furthermore, Chen et al. [192], discuss multiuser diversity over a parallel hybrid FSO/RF point-to-multipoint communication system that consists of a hybrid access point (HAP) which allows multiple FSO users and multiple RF users to communicate with the AP, as illustrated below in Figure 22. For each transmission block, FSO or RF user is scheduled to communicate with HAP. The FSO link uses quadrature modulation along with IM/DD [81]. First, a quadrature modulation algorithm is used to modulate the information. The transmitter laser intensity is afterward directly modified by the modulated electrical signal. The electrical signal is demodulated to produce the discrete-time equivalent baseband signal once the DC bias has been removed. An efficient user scheduling strategy is proposed to select the user based on his/her link quality. OP and average bit error rate are analyzed to evaluate the performance of the whole communication system. Furthermore, novel asymptotic closed-form expressions at high signal to noise and new closed-form expressions for PDF and the CDF of the selective user SNR are derived. The results confirm that, as the number of FSO and RF users increase, the overall performance increase where the HAP operates in time division multiple address mode, which means that it chooses one RF user or one FSO user to fill out each time slot. This model can be enhanced by using soft-switching instead of hard-switching, where it cannot transmit data over both RF and FSO links simultaneously.

Dat et al. [81] establish a hybrid FSO/MMW system that provides a high-speed and stable fronthaul transmission. They develop new worthwhile voice coil mirrors (VCM) for a fine tracking system. The authors apply an adaptive combining technique using MRC scheme. With this switching approach, the RF link is only enabled via a 1-bit feedback signal when the instantaneous SNR of the FSO connection at the receiver is less than a switching threshold. The FSO link remains active throughout. VCM expands signal at transmitter and signal coupling at receiver. Fine tracking is essential for this project as the signal transmitted from the beam splitter has to be directed to a quadrant photodetector (QPD) to implement the system. A proportional integral derivative servo controller controls the VCM and utilizes the information of the received signal from the QPD. The system transmits 100 Gbps over the FSO system and 50 Gbps over the hybrid FSO/MMW system using Nyquist-Subcarrier multiplexing signals. According to the authors, this experiment transmits signals over three different cases. The first case utilizes FSO system with two different wavelengths. Secondly, a hybrid FSO with intermediate-frequency-over-fiber MMW system deploys an electrical-up conversion. Lastly, a hybrid FSO with radio over frequency MMW system utilizes photonic-up conversion. The total bandwidth of the system is 10 GHz, with 10 subcarriers and an arbitrary waveform generator (AWG) generates the centered frequency at 7.5 GHz in MATLAB. A driver amplifier amplifies the signal, followed by modulating the optical signal from two different LDs. An optical IQ modulator generates an optical single-sideband to lower the dispersion occurring during the transmission. An Erbium-Doped Fiber Amplifier (EDFA) amplifies the modulated signal to be sent over a 20 km SMF to a remote antenna unit (RAU) to be transmitted through 5 m of free space using a FSO antenna received by another FSO antenna to be directed immediately to an SMF fiber. The EDFA amplifies the signal. Nevertheless, the photodetector detects and filters the amplified signal. Finally, the signal is transmitted to a real-time oscilloscope and evaluated offline.

The second experiment acquires only one laser diode and utilizes only a single wavelength for data transmission. At the RAU part, the optical signal splits into two parts using a 3 dB optical coupler. The first part transmits over exactly the same as the first experiment. The second part passes to a photodetector to perform an intermediate frequency then the electrical mixer transforms it to 90 GHz. Moreover, it passes through a band pass filter and the desired band will be transferred using a 43 dB antenna. The vice versa of the process accomplishes at the receiving part to transmit the original signal by the sender.

In the last stage, the MATLAB generates an intermediate frequency and passes through two distinct synchronized AWG. One of them transmits the signals over a FSO system as deployed in the second experiment. The second one transmits over a radio over fiber (RoF) MMW system at 90 GHz. It deploys high-gain antennas to extend the transmission space of the MMW link. The collected IF signals pass to two inputs of OSC and are inspected offline. The proposed system affords an excellent solution for the future technology for fronthaul mobile communication, especially ultra-dense small cells for 5G and beyond networks where the fiber cable is unsuitable. In addition, to verify the effectiveness of the developed FSO and hybrid systems, they will undertake experiments in an outdoor setting in the upcoming works under various weather conditions.

Khalid et al. [193] analyze hard switching between FSO Link and RF Link. FSO link has the higher priority as a primary link between sender and receiver until SNR falls below a specific threshold value. The backup link, which is the RF link, will be activated to continue the transmission until the ideal condition comes back. Hybrid FSO/RF systems increase the accuracy and reliability of the network. Moreover, the author implements OOK for IM/DD and then analyses the BER and average capacity. As long as the received value is above the threshold, the FSO link will keep working for data transmission until the quality reduces then the RF link will work immediately for backup. Only one link will be activated at a time, so the system will not be interrupted. As the weather changes rapidly, the switching device needs to switch between both links as fast as possible to decrease the outage in data transmission. System performance is analyzed based on OP, BER, and ergodic capacity. Results show that the proposed hybrid FSO/RF model has significant improvement and higher performance in terms of reliability and availability than the conventional single FSO model.

Eslami et al. [194] propose a new model called “Hybrid Channel Coding” that can be applied for both links to achieve reliability. Hybrid Channel Coding associates non-uniform coding and rate-adaptive coding using LDPC codes. Non-uniform codes provide high efficiency and reliability for communication over parallel channels using LDPC codes. They are designed for fixed channels such as time-invariant. The main objective of this project is to transmit data over parallel channels, FSO and RF channels, with error-correcting code for each channel of both. The codes in this model provide magnificent performance enhancements compared to the existing models in terms of availability. Future study will involve using Hybrid Channel Codes with effective Very Large-Scale Integration (VLSI) Architectures and a testbed to evaluate the performance of the proposed system to that of the current systems.

In Yasir et al. [195], the authors study fog and dust concentration to model a high-efficiency FSO communication link that utilizes an adaptive technique and switch over-analysis relying on the power received to maintain high data transmission rate. This project deploys the Kruse attenuation model for estimating the optical attenuation as it is more sensitive to 1550 nm wavelength. Regarding the concentration of the dust, by utilizing the V-TSD model for estimating the visibility and CCIR formula for estimating the optical attenuation. MATLAB estimates attenuation and link parameters such as power received, BER, and SNR in all simulations. The proposed adaptive FSO system will increase the transmitted power by 5 mW if the FSO link is facing harsh weather conditions until it reaches 100 mW as the maximum limit. All of these factors develop an adaptive technique on MATLAB to boost the efficiency of the FSO communication by switching to RF communication link when the threshold value is reached to maintain the link margin for the data rate. In addition, it compares diverse adaptive techniques where most of them utilize coding techniques for modulation. However, the switching technique is simpler and easier to comprehend as it is plainly mathematical. As proved by the numerical results, one of the simple effective techniques to achieve higher SNR is increasing the transmitter power. Nevertheless, the existing techniques such as SIMO, MIMO, WDM, adaptive coding, and modulation make the system more complex. Moreover, increasing the transmitter power allows the FSO link to maintain a high data rate of up to 23.1 Gbps in diverse weather conditions. This model can be improved to maintain higher link availability by applying automatic repeat request (ARQ) based on the receiver’s RF feedback system to utilize the capacity of the main link as much as possible [196].

Nadeem et al. [197], utilize a system that uses 850 nm for the FSO main link and a backup RF link operates at 40 GHz as an alternative to 10 GHz. This is done as 10 GHz is utilized for long-distance communication that can reach up to 50 km with a lower data rate than 40 GHz, up to 10 km only. However, propagation at 40 GHz has a higher data rate than the 10 GHz. A hybrid FSO/RF communication link provides a secondary back up link to be used for a period of time while the primary link is not fully operational due to interruption happens due to some changes in the weather. In addition, researchers investigate the effects of fog, rain, and snow on the primary and secondary links. Firstly, fog is the most influential factor in optical signals. It causes a high attenuation by Mie scattering during the collision of the optical light and the fog particle. The size of the fog particle is almost the same as the wavelength of the optical light and close to the infrared waves. The attenuation can be predicted with high accuracy by applying the Mie scattering theory. On the other hand, it needs some comprehensive information about the fog parameters, such as the size of the fog particle, and the refractive index is not readily available all the time, especially at the desired site of installation. Moreover, the tiny fog droplets will also affect the backup RF link as long as the frequency is greater than 10 GHz. Rayleigh approximation is authentic as long as the size of the fog droplets is less than 0.01 cm and propagates with frequencies below 200 GHz. Secondly, rain attenuation plays a great effect on the network because its size is sufficient to cause reflection and refraction. Finally, snow also affects data rate by attenuating the transmitted power. Snow could be dry and wet; each has its parameters. The numerical results confirm that the hybrid system has much higher availability than using each link alone. The future work will focus on pinpointing the best parameters for different seasons and locations and then developing an adaptive algorithm to achieve optimum performance in various weather conditions.

This hybrid RF/FSO system in [198], transmits private information over two independent parallel links at the same time, which are FSO and RF links to a receiver. However, a vicious eavesdropper is trying to hijack the RF link. A switching scheme eliminates data loss, a secondary user receiver utilizes different techniques to select the desired data stream with the considerable value of SNR from both links and uses heterodyne detection to identify the data sent over the FSO link. Furthermore, pointing errors and link blockage in the FSO link are considered by the authors. In addition, Amirabadi et al. [199] propose a novel single-hop hybrid FSO/RF with two parallel links. Receive diversity is used in this project for the first time. The authors inspect the single-hop hybrid FSO/RF system at saturated atmospheric turbulence. OP was expressed to indicate the performance of the system. The authors utilize equal gain combining and MRC for FSO and RF links, respectively. FSO link utilizes IM/DD Merging FSO and RF links in the same system gives a huge advantage to the mixed system, such as reliability, security and high capacity. In contrast with any other single-hop FSO/RF system, the design of multi-receive in this project helps the receiver to attain high accuracy by blending the various copies of the originally sent signal which enhance the performance of the project. The authors suggest using the presented system in Mediterranean weather as it has heavy rain or dense fog, even though this system is more complex due to the received diversity. On the other hand, it also reduces power consumption and enhances the system’s performance.

Song et al. [200] authors construct a switching structure that consists of a pair of FSO and RF transceivers. The RF link utilizes 16-QAM modulation, and FSO uses IM/DD that achieve 340 Mbits/s and 900 Mbits/s, respectively. The authors observe the optical received signal by an optical power detection module. A pair of smart-managed switches execute switching between nodes to allow end devices to receive the optical signals. This mixed FSO/RF hybrid system can automatically shift between links according to the threshold and the received optical signal. Thus, the system can be more stable and handle communication in a challenging climate by boosting its performance. The adaptive link selection and the switching mechanism increase the stability. Moreover, communication between the upper computer and the lower computer is done via serial-port communication, with the aid of the iPerf3 network test tool for monitoring the system transmission rate.

Grigoriu et al. [201] examine a system with two wireless links, FSO and RF. Both links can be represented by finite-state Markov chain (FSMC) models; FSO links are the main link with a high data rate. On the other hand, the RF link is a backup link for the main link in case it cannot work properly due to atmospheric conditions. The main link has FEC to enhance data reliability by adding redundancy to the transmitted signals and stop-and-wait ARQ. High priority packets will be transmitted over the main link at the beginning, if they fail to be transmitted precisely, they will be switched to the backup RF link to be transmitted immediately as the RF link is already working simultaneously with the main link to transmit low priority packets. Cyclic redundancy check code is added to all packets at the sender side to allow the receiver to recognize whether the received data have arrived correctly or not. The most significant advantage of using a point-to-point FSO communication system is that it serves high data speeds reaching 10 Gbps. However, it can be easily affected by the weather. Thus, the RF link is used with the main FSO link to provide higher availability for the system. Hybrid FSO/RF system is an achievable scheme for stable P2P wireless communication systems in different weather conditions. In equatorial countries, RF links are unstable due to rain. Basahel et al. [40], study the tentative models to estimate the availability of RF and hard switching hybrid FSO/RF systems for data transmission up to 5 km. The prediction of the availability of the models is based on long-dated statistics of weather attenuation in tropical climate zones. The frequency of the RF link ranges from 10 to 100 GHz using the International Telecommunication Union Radio rain attenuation model. These models can wisely choose the suitable RF for FSO’s backup. Results prove that the frequency used increases with shorter distances and has better performance in terms of outage probability and availability. Moreover, a mixed FSO system with RF link performs much better than a standalone communication system.

Chatzidiamantis et al. [202], proposes a new hybrid FSO/RF communication system to work efficiently even if there is no feedback or CSI at the transmitter. The authors assume that both links have the same data rate speed, and desired data to be transmitted will be sent over both links with the same modulation. The modulation used at both links is the PSK because it is easy to be performed at both FSO and RF links. Since data are collected individually at each link, combining methods are used to combine data in the electrical domain on a symbol-by-symbol basis. The first method is the SC, that only processes the link between both, which has greater SNR. Secondly, MRC processes the output of both links to calculate the overall SNR. BER is computed to examine the performance of the proposed system. Results show that MRC has higher performance compared to SC combining scheme.

Touati et al. [203] appraise the performance of a parallel soft-switching FSO/RF system placed at Qatar University under both dry and brutal weather. FSO uses IM/DD with OOK, and RF uses LOS link with 16-QAM. The lowest temperature that could be reached in Qatar is 17 °C which is during the winter, while it can exceed 46 °C in summer. Furthermore, Qatar’s weather is humid, so the humidity can reach more than 90% in Autumn. On the other hand, fog and rain presence are abnormal and extraordinary in Qatar compared to other countries [204]. Therefore, Qatar does not face rain and fog attenuation problems for FSO communication. However, haze can decrease the system performance. High temperature and high humidity can cause high instability due to the change in the refractive index. Researchers use FSMC to modulate the signals and determine mathematical expressions to calculate the OP of the soft switching. The distance between sender and receiver is 600 m. The system is P2P which transmits data wirelessly using infrared laser lights with a 1550 nm wavelength. The authors confirm that the system is working fine even when the FSO link is in an outage state due to low power transmission and/or strong turbulence as the system switches to the RF link till the FSO link is available again.

Switching between FSO and RF links in hybrid FSO/RF systems plays a significant role as it relies on the value of the SNR threshold of both links. Thus, choosing optimum thresholds of both links gives higher reliability to the system. Shrivastava et al. [205], propose a new switching technique to decrease the misuse of optical power and BER. In addition, boosts the performance of the system. Figure 23 shows the flowchart of three different weather conditions, “clear weather, moderate fog, and dense fog” with two threshold values. In clear weather conditions, all data are transmitted by the optical link only while the RF link is in standby mode. When the quality of weather and SNR of the received data decreases below the upper threshold, the RF link is switched to work simultaneously with the FSO link to transmit data together in moderation.

When the quality of optical link degrades and the received optical SNR falls below an upper threshold level, the RF link activates. In this situation, both signals reach the receiver separately and combine through MRC. Data duplication on the RF and FSO links provide a diversity of order two. When received optical SNR falls below the lower threshold level, then the optical link is put in standby mode and only RF link sustains the required transmission. If the RF SNR is also lower than the RF threshold, outage is declared and no transmission takes place. The optical link activates as soon as the environmental conditions become favorable and optical SNR improves. Thus, the switching scheme conserves optical power, conserves RF power, prevents generation of unnecessary RF interference, improves system performance and overcomes the drawbacks of existing switching schemes [205]. Shrivastava et al. [206] confirm that thresholds cannot be chosen randomly; determining the optimum threshold depends on the average FSO and RF SNR.

Rakia et al. [207], have a novelty for the CDF of the received SNR for the hybrid FSO/RF system with an adaptive combining technique that uses PSK modulation. The FSO link works individually if the received SNR exceeds the determined threshold. However, if the quality is insufficient, the system switches on the RF link and receives the data using MRC at the receiver to collect data sent over both links. To simplify diversity reception, the system might decrease the transmission rate of the FSO link so that it can be the same as the RF link as long as both of the links are working at the same time until the quality becomes high and tolerable again. Thus, the RF link is put on standby mode to minimize the power consumption. The CDF improves system reliability, which is free from switchover problems, does not generate unwanted RF interference, saves power, and uses FSO link most of the time due to the high transmission as it utilizes IM/DD scheme. The results show that the proposed hybrid FSO/RF with an adaptive combining transmission scheme has higher performance than the FSO-only and RF-only system during all the weather conditions in terms of OP.

Nonetheless, He and Schober [208] propose a novelty in an architecture for a hybrid FSO/RF wireless communication system modeled as a heterogeneous block fading channel. The system utilizes a bit-interleaved coded modulation (BICM) of all the bits transmitted over RF and FSO links. The FSO link uses IM/DD with OOK transmitted symbols. The results obtained by the simulation show that the hybrid FSO/RF system with BICM has a better performance compared to a system with a simple repetition code and selection diversity. The authors consider that FSO and RF links have only one transmitter and one receiver aperture, respectively. Transmitters and receivers of both subsystems are placed close to each other with a direct line of sight to ease the joint processing of received bits of both links. The frame error rate and cutoff rate results confirm that the proposed system has excellent performance over different weather conditions. However, it can be improved by replacing the convolution encoder with more vital LDPC codes or turbo codes, utilizing MIMO technology in both sub-systems, and applying multicarrier techniques to achieve full diversity. Furthermore, the system can assign the optimum power for both sub-systems.

Kumar and Borah [209], inspect the performance of hybrid FSO/RF symbol mappers using BICM with iterative decoding (BICM-ID), since the system has different data rate speeds at both FSO and RF links. In the beginning, the data bits are encoded by a channel encoder and then interleaved to be mapped to hybrid FSO/RF symbols using conventional RF Quadrature Phase Shift Keying (QPSK) symbols and OOK FSO transmission. On the other hand, iterative decoding is placed at the receiver to decode the received data bits. Moreover, to find the best symbol mappings, an orthogonal evolutionary method is utilized. The orthogonal evolutionary algorithm is presented with two cost functions. Extrinsic information transfer (EXIT) charts are utilized for the cost function in the high SNR domain and analytical error-bound formulas for the low SNR regime. The FSO link uses ten times the channels used by the RF link. The results demonstrate that, even with a substantial data rate imbalance between the two links, hybrid symbol mappings achieve a considerable performance gain in FSO/RF channels. 

However, Zhang and Hranilovic in [210] evolve a soft-switching design to be used for hybrid FSO/RF communication system by utilizing short-length Raptor codes which accomplish high data rate speed with low decoding cost. The Raptor encodes data which are sent at the same time over both links where the codes can modify based on either link with finite feedback. Short-length Raptor codes have different values where K can be from 16 to 1024. In addition, it is agreeable for high-speed implementation. A field programmable gate array (FPGA) is used for implementing the Raptor encoder (K = 16) and decoder which can support a data rate of 714 Mbps with 97 mW of power consumption and 26,360 gate circuit scale. The proposed soft-switching technique using Raptor code has achieved an average 472 Mbps data rate compared to 112 Mbps on average for the hard-switching technique, where the FSO link uses intensity modulation with direct detection IM/DD technique using OOK. However, the RF channel is a WiMAX point-to-point link that works at 5.8 GHz with 64-QAM modulation and is modeled as a fading-free, additive white Gaussian noise (AWGN) channel since the range is less than 2 km. Raptor encoder and decoder can be easily added to any current system with minimal extra cost. Two alternatives for a hybrid FSO/RF link setup are shown in Figure 24. For data transmission in the hard-switching setup, the transmitter and receiver jointly choose the FSO or RF channel. The high-data-rate FSO link is only chosen if the channel circumstances allow for dependable communications; otherwise, all data are sent over the RF channel [211]. One link is always idle while using hard switching, which is a significant drawback. In real life, the sensitive FSO channel will only be chosen infrequently due to scintillation or loss. The FSO link’s channel capacity is squandered once the RF link has been chosen. On the other hand, Hilt in [212] confirms that different modulation types have different thresholds for radio networks employing adaptive modulation. When the fading margin makes up for the loss brought on by rain, the low modulation mode is used during wet minutes. During sunny periods when the fade margin is available, the link can jump to higher modulation modes up to 512-QAM; thus, more data can be transmitted [213,214]. Results in [212] only display QAM up to 512 for simplicity. However, modern transceivers may achieve several Gigabit/s data speeds by using up to 4096-QAM. Nevertheless, the findings presented in [215] make it abundantly evident that raising the state of QAM modulation mode not only enhances the throughput but also reduces the amount of power it needs to operate.

A solution to the hard-switching drawback is to use channel coding to coordinate data delivery over both connections, as shown in Figure 24. According to [216], data are encoded using a single LDPC code, with a portion of the codeword split between the FSO and RF connections. The rate is then modified via puncturing per the instantaneous channel circumstances. Although this method outperforms hard switching, it is challenging to use at FSO data rates due to the need for complicated soft decoding and the requirement that transmitter and receiver channel characteristics be known. For hybrid FSO/RF lines, raptor codes have also been considered [217,218]. These codes can adjust their rate to channel conditions with just one bit of feedback for every message, and they do not require transmitters to be channel experts [218] considers a bit-wise Raptor coding system in which the bits sent on FSO and RF lines are arbitrary linear combinations of message bits. A well-known code is used [217], and it is assumed that both the transmitter and receiver have the knowledge of how each bit is connected to the others. Additionally, soft iterative detection at the receiver is necessary, and the influence of changing atmospheric conditions is not considered.

Khan et al. [219], propose an adaptive FSO/RF communication system with a novel throughput maximization algorithm (TMA) that avoids the main problems of individual FSO and RF systems. The authors evaluate the performance of the proposed adaptive system using regular and right-regular LDPC code during different weather conditions. The simulation results prove that the TMA using the right-regular LDPC code greatly achieved high performance during all weather conditions in terms of performance gain. The FSO link applies IM/DD with 2-PAM or 4-PAM. However, the RF link utilizes BPSK or QPSK. The results prove that TMA is stable under different weather conditions. In addition, using right-regular LDPC code increases the performance gain up to 2.25 dB under all channel conditions. What makes this project unique is that it selects its parameters based on the channel conditions.

Rakia et al. [220], establish and test a novel P2MP hybrid FSO/RF network that consists of many remote nodes around the central node, as shown in Figure 25. Each remote node has an individual coherent/heterodyne primary FSO link and a common backup RF link with the central node. Using a common RF link among all the remote nodes greatly benefits the system, such by sharing the RF spectrum, avoiding generating useless RF interference, and saving the RF power. It analyzes the performance of an individual remote node (tagged node) instead of considering the whole system’s performance. Focusing on the tagged node grant the authors to inspect specific criteria, including throughput from the central node to the tagged node, average transmit buffer size, symbol queuing delay in the transmit buffer, and the efficiency of the queuing system, the symbol loss probability, and the RF link utilization. The central node is aware of the quality of all FSO links between the remote nodes and switches to the RF link with the node that experiences a failure in the FSO link. The coded digital baseband signal generated by the signal source is converted into an analog signal via an M-square QAM electrical modulator. The FSO link uses a heterodyne detection for the received signal and the RF link upconverts the QAM electrical signal using a 60 GHz RF carrier generated by the RF local oscillator. The numerical results prove that the P2MP hybrid FSO/RF system accomplishes great enhancements over the P2MP FSO-only communication system with the help of a discrete-time Markov chain model for buffer transmission.

Abadi et al. [221], review the performance of a hybrid FSO/RF communication system with hard switching in terms of data rate, link distance, and link availability. The authors used two methods in the investigation; time hysteresis (TH) and power hysteresis (PH) switching protocols. The proposed system consists of two coupled laser sources at two different wavelengths (780 nm and 830 nm) which use a non-return to zero on-off keying (NRZ-OOK) and its inverted version for intensity modulation at both laser sources respectively. An optical lens is used to launch the combined light beams from the SMF into a free space channel. However, there is another optical lens at the receiving side to capture the transmitted light into two multimode fibers that consist of a higher aperture by an optical splitter. A pair of optical filters are used to differentiate between the two wavelengths followed by a differential detector to regenerate the NRZ-OOK signal. On the other hand, the RF channel uses BPSK modulation followed by a mixer, amplifier, and band-pass filter BPF to up-convert the transmitted signal. Moreover, at the receiver side, the transmitted signal must be down-converted using a low-noise amplifier, mixer, BPF, and subsequently, a BPSK demodulator to get back the original signal. Figure 26 and Figure 27 explain the algorithms of both PH and TH methods, respectively. The switching process for TH has a delay of 10 s and PH with a power margin of 3 dB. The results prove that the PH scheme performs better than the TH regarding data rate.

Shakir et al. [222], proposes a new hybrid FSO/RF communication system that does not need CSI. The FSO and the 60 GHz RF sub-systems use BPSK modulation with IM/DD. The hybrid system sends the same data with the same data rate over both FSO and RF links simultaneously. However, the system applies the SC scheme at the receiver to merge the received signal from both links for higher reliability. The authors examine the performance of the proposed system under different weather conditions using average bit error rate and OP. The analysis shows that the proposed system can work efficiently in intense weather conditions and has excellent performance compared to FSO-only systems. In addition, BPSK has better performance under harsh weather conditions than the QPSK and 8-PSK modulation techniques. Over and above that, Song et al. [223], blend power control and link switching via Recurrent Neural Networks to anticipate the Received Signal Strength Indication (RSSI). The authors develop a new brilliant power control and link switching technique that boosts the hybrid FSO/RF system’s efficiency. The results validate that the proposed project can precisely predict the RSSI with dramatic enhancement by the power control scheme in terms of transmission quality in the FSO link by keeping it operating as much as possible with changing the transmission power. The RF link consists of a QAM modulator where the transmission speed of the RF link is about 340 Mbps using 16-QAM, and the FSO link supports up to 10 Gbps for data transmission. This model can predict RSSI and keep the FSO link active as much as possible by adjusting its transmitting power, as proved by the experimental results.

Khan et al. [224], propose an adaptive hybrid FSO/RF communication system based on the atmospheric condition with a novel algorithm that relies on a puncturing technique that uses LDPC code. The authors present new results of an EXIT chart for both binary and quaternary modulations techniques of the proposed system. Moreover, the proposed communication system is simulated and well-studied in terms of performance analysis. This algorithm is affordable with higher performance gain in different weather conditions and provides efficient high data rate transmission with improved flexibility (less complex). Additionally, it is well adapted to various weather scenarios as it provides the optimum threshold and the corresponding puncturing patterns. Furthermore, Odeyemi and Owolawi [225], investigate the performance of a hybrid FSO/RF system that utilizes SC at the receiver. The FSO link employs IM/DD, the RF link operates at 60 GHz, and both use binary modulation schemes. The authors take pointing errors into consideration to obtain outage probability at high SNR and ABER. The results prove that the proposed system can overcome the defects of the FSO link under weather attenuation and pointing errors. Moreover, the performance of the hybrid system depends on the ratio between the receiver beam radius and pointing error displacement standard deviation (jitter) at the receiver side. In other words, the higher the ratio, the higher the performance. All the simulations justify the truthfulness of the derived expressions. Results show that the hybrid FSO/RF system performs better than the FSO-only system in different weather conditions and pointing errors.

Haluška et al. [226], use RSSI prediction analysis by using a hard switching technique between the FSO link and the RF link. What makes it unique from others, is that they are using machine learning to anticipate the received optical power. RSSI is essential for hard switching in hybrid FSO/RF to gain information on RSSI value in real-time, it is an index to warn before a full outage of the FSO subsystem in the communication system. In this project, they use Raspberry Pi3 Model B minicomputers and sensors to study the weather conditions and received signals. All the results obtained are used as an input for machine learning so that it can predict the received optical power with higher accuracy. On top of that, Moradi et al. [227], reconfigure the path of a hybrid FSO/RF communication system to ensure that the FSO link will be queued again with high priority when the FSO link is inactive due to high loss. By reconfiguring, the communication system will have high efficiency and high reliability with the most extended duration of active FSO link that provides high data rate transmission. However, path configuration may increase the delay and packet loss and decrease the throughput. The authors study the reconfiguration probability to evaluate the performance of the proposed system. Results obtained show that the increase of turbulence-induced lognormal will not ineluctably increase the reconfiguration probability. OOK modulation scheme is used for data transmission. The authors model the reconfiguration communication system as FSMC because it is a convenient method.

Tokgoz et al. [228], examine a hybrid FSO/MMW system in terms of physical-layer security for the first time in the presence of wiretapping the data transmission sent over both links simultaneously. Receiver/eavesdropper utilizes MRC combining technique where it is one km away from the sender. The system’s performance is analyzed in terms of the probability of SPSC of the wiretap channels. There are different eavesdroppers such as RF, FSO, and hybrid Eve. The location of the eavesdropper could be either close to the transmitter or the receiver. The performance of all three is compared in terms of physical layer security. The results show that the Hybrid-type eavesdropper has the lowest performance compared to FSO and RF because Eve uses both FSO and RF links simultaneously in hybrid communication. Analysis can be essential for designers to secure data sent through the system by boosting confidentiality. Tokgoz et al. [229] propose a novel selection mechanism for hybrid FSO/MMW systems using the PSK modulation technique for both FSO and MMW links without requiring any feedback or CSI at the sender side. The index modulation scheme determines which link to activate, the FSO or the MMW link. The performance of the proposed index modulation-based link selection mechanism is evaluated in terms of spectral efficiency, average BER, outage probability, and ergodic capacity. In addition, results are compared to the classical switching mechanisms with different modulation schemes, link distances, and weather turbulence. The numerical and simulation results confirm that the proposed project boosts the overall system performance in terms of efficiency and average BER. The first bit of each transmitted block determines which link will be used for data transmission. On the other hand, maximum likelihood detection is used at the destination. Moreover, FSO subsystem uses IM/DD technique. This model consumes low power and does not require combining or multiplexing methods at the receiver side as it activates only one link simultaneously. Additionally, it performs better than the threshold-based scheme concerning BER and spectral efficiency.

Li et al. [230] demonstrate a flexible bidirectional fiber-FSO-5G wireless communication system that uses a dual-arm Mach-Zehnder modulator for down streaming using intensity modulation and phase modulation for upstreaming. In addition to the presence of 5G sub-6 GHz links used for BSs to communicate together. Since that 5G MMW technology is not suitable for rural and suburban areas, 5G sub-6 GHz solves this problem. SMF in this project is used to transmit data over 25 km, FSO link is 600 m range and 10 m/4 m RF connectivity. As a remotely injection-locked DFB LD is used to convert phase modulation to intensity modulation, the communication of this proposed project is done in a feasible way that delivers 10 Gbps. These results obtained can be used in 5G networks as it meets the requirements of 5G communications; it has two-way high-speed, and long-haul communication due to the integration between the optical fiber and optical 5G wireless networks as 5G services require low latency and high bandwidth. Kim et al. [231], represent a passive optical network that provides high speed with low latency enabled by time-controlled-tactile optical access (TIC-TOC) technology. This strategy supports low latency and bandwidth-intensive services for 5G mobile networks by using channel bonding and low latency-oriented dynamic bandwidth allocation. The authors implement FPGA-based optical line terminal, and multi-speed optical network unit prototypes to verify the practicality of TIC-TOC. The results prove that the transmission can reach up to 50 Gbps without loss with less than 400 μs latency over 20 km of SMF for 48 h. Moreover, the communication capacity can reach 100 Gbps by adding more channels to the communication system.

Moreover, Li et al. [232] propose a hybrid Internet/cable television (CATV)/5G fiber-FSO integrated communication system for the first time; the SMF and FSO links are 40 km and 200 m, respectively. The system utilizes a triple-wavelength polarization multiplexing scheme; the lights in this project are modulated with 60 Gbps 50–550 MHz vestigial sideband (VSB) CATV even-channel with 4-PAM for the internet. On the other hand, 70 Gbps VSB with 16-QAM-OFDM for transmitting 5G signals to convince the services, and 4-PAM has a high capacity as its spectrum is half the spectrum of the non-return-to-zero signal [233,234]. Moreover, 16-QAM-OFDM modulation has high spectrum efficiency, high capacity, and a robust dispersion tolerance [235]. A Hybrid Internet/CATV/5G fiber-FSO integrated system is suitable for broadband networks as it affords high-speed services such as internet and 5G mobile telecommunication. Channel performance and capacity can be enhanced by using a Tunable optical bandpass filter which is a straightforward and beneficial polarization demultiplexing mechanism. Moreover, Illi et al. [236], study the performance of a hybrid 5G RF/FSO system. Data are sent over both channels independently to be combined at the receiver using MRC technique. Then, the MGF is calculated for the total SNR. ASER is presented of the proposed RF/FSO system for different modulations such as QPSK, BFSK, and BPSK using bivariate Fox’s H-Function. The numerical results show that BPSK has the best performance compared to QPSK and BFSK.

## 7. Using FSO Links Only

Kiasaleh [237], investigate a FSO communication system that utilizes a HARQ scheme with BPPM where the receiver identifies the received signal using direct detection technique while shot- noise, background noise, and thermal noise are taken into consideration. The performance of the proposed HARQ is determined in terms of packet error rate, while the results conclude that the HARQ-FSO system is more stable and reliable during weather turbulence than the conventional ARQ-FSO system. The results validate the performance of HARQ compared to the standard receivers in terms of packet error rate for a moderate level of turbulence ($\sigma_{SC}^{2}$ = 0.75). However, there is a great improvement when the turbulence level is reduced.

Gurjit et al. propose [238] a hybrid Polarization Division Multiplexing (PDM)/OFDM, which sends different signals over orthogonal states of polarization by a polarization splitter that splits the optical signal into two polarized orthogonal signals which are carrier and modulated data of users, this method boosts the transmission effectiveness. Both orthogonal signals are split and demodulated to retrieve the original signals at the destination. The proposed system is tested under different weather conditions. Results prove that PDM enhances the overall performance of the system by increasing the user capacity and spectral efficiency by considering a data set of Delhi weather conditions. In addition, the OFDM modulation technique decreases the multipath fading occurrence and avoids random fading effects due to the FSO transmission. The results demonstrate that the PDM and OFDM combined in one system enhance the overall performance of the system in terms of quality and reliability. However, in low visibility weather conditions, the availability of the FSO link decreases due to the increase of the concentration and the size of the fog particle compared to clear weather conditions with high visibility. Regarding numerical results, the SNR in clear and cloudy weather is 35 dB and 18 dB at 5 km, respectively. However, Sharma and Grewal [239] propose a FSO communication system under lognormal turbulence that uses pulse position modulation–Gaussian minimum shift keying (PPM–GMSK) scheme as shown in Figure 28. The system’s performance is evaluated in terms of ABER via a simple method using Gauss–Hermite. Results confirm that as the modulation order of PPM increases, the error rate and energy efficiency increase. In other words, 2-PPM–GMSK performs better than 16-PPM–GMSK but consumes a lot of energy.

Nam et al. [240], analyze the system performance of a threshold-based parallel FSO-based communication system using a selection scheme of multiple beams with wavelength division multiplexing (WDM) with considering pointing error. The authors analyze conventional detection schemes such as heterodyne detection and IM/DD techniques for adaptive modulation (AM) and non-AM such as coherent/non-coherent binary modulation. The system’s performance is evaluated in terms of OP which determines when the received power falls below the threshold [241], average spectral efficiency (ASE), and BER. Results show that the performance has high spectral efficiency with low complexity and low loss by applying the selection-based beam selection scheme. Moreover, AM case has higher ASE and lower BER compared to the non-AM cases.

Najafi et al. [242], authors introduce a dynamic NOMA technique that decreases the OP of two BSs which send their data in the same frequency band and at the same time to the single central unit (CU) over FSO backhaul links using IM/DD by resolving the best decoding. The CU utilizes successive interference cancellation (SIC) decoding scheme to decode the signals received from the BS. The two BSs and the CU are mounted on buildings, communicating over a line of sight. The simulation results show that the system has successfully decoded under perfect, imperfect, and worst SIC conditions with high performance compared to Orthogonal Multiple Access. Furthermore, 4G networks now can reach up to 100 Mbps of download speed, and 5G connectivity is expected to reach 20 Gbps of download speed. Esmail et al. [243] experimentally test how dust storms affect the performance of an all-optical FSO link using 1550 and 520 nm wavelengths. Keysight M8190A 12-GSa/s arbitrary waveform generator is used to generate A 1-Gbaud/16-QAM 5G signal, then up-converted to a 28 GHz via a Keysight E8267D vector signal generator. The results obtained show that the dust does not affect the RF link as its wavelength is larger than the dust particle size, and the 1550 nm has better performance than the 520 nm for the FSO link. In addition, FSO is an admirable alternative to fiber cable to allow wireless backhaul/fronthaul connectivity. Consequently, a hybrid FSO/RF system is a great solution to allow continuity of the system when the FSO link is experimenting with strong weather conditions. Table 6 and Table 7 categorize the models that share the same characteristics, such as the simulation software that was employed and the fading distributions for FSO and RF links.

## 8. Fourth, Fifth and Sixth Generations

As stated in Section 4.1, Section 5, Section 6 and Section 7 that [102,156,157,159,160,161,162,180,182,230,231,232,236,243,244] utilizes MIMO, Relaying, Hybrid FSO/X and FSO only techniques, respectively to achieve 5G connection. Furthermore, Chowdhury et al. [245], compare the main differences between 4G, 5G, and 6G, as shown in Table 8. Despite the advantages of the 5G communication systems such as low delay, high efficiency, and low hardware complexity. Moreover, 5G systems will not be able to fulfil the market demand after 2030; 6G will be able to fill the gap caused by the market. The 6G communication system can be summarized in a few points; the communication will be integrated with artificial intelligence, has high availability and reliability with high energy efficiency, low data loss, and high security and privacy. Since that high-capacity backhaul is one of the main requirements for networks, high-speed optical fiber and FSO systems are used to support high-capacity networks. Where FSO is a perfect solution for stable fronthaul/backhaul connection for 5G and 6G, especially for long-range, it can be used for remote and non-remote areas such as space and isolated islands. Moreover, it supports cellular BS connection as it can handle many users with high bandwidth by producing a focused narrow light beam via the light diode transmitter, which allows p2p communication with a high data rate. Therefore, hybrid FSO/RF is the best solution in 6G connectivity to overcome the limitations caused by the atmospheric changes [18].

As small cells are the main key technique for 5G networks instead of the macrocells, it has some challenges to provide omnipresent backhaul connectivity to all small cells. Siddique et al. [246] provide various backhaul solutions to the challenges caused in the backhaul small cells. This article focuses on RAN architecture, where the small cells communicate with the macro BS, not the cloud for backhauling. Additionally, the 5G promises an update in RAN that will support the huge wireless data traffic. Lopes et al. [247] implement and test two independent hybrid architectures applied to 5G NR Fiber-Wireless (FiWi) systems with different optical fronthaul approaches with a 25 km fiber link. The first architecture works in non-standalone mode, described as a third-generation partnership project to transmit 4G and 5G concurrently using a unique FiWi system. A filtered OFDM signal at 778 MHz with 10 MHz bandwidth at the 5G transceiver; a long-term evolution (LTE)-advanced signal with five 20 MHz sub-bands centralized at 2.24 GHz; a 5G NR signal at 2.35 GHz with 100 MHz bandwidth are the three waveforms studied in this architecture respectively. However, the second architecture implements RoF, FSO, and wireless technologies merged into a heterogeneous network. The maximum capacity of RoF/FSO and RoF/FSO/Wireless FiWi transmission are 3 and 1.4 Gbps, respectively.

### Using FSO Links Only in 5G Communication

As floating BSs will be playing an important role in transmitting data in 5G FSO networks, Yu et al. [248] propose an atmospheric turbulence suppression algorithm that utilizes avalanche photodiode adaptive gain control (APDAGC) that transmits laser beam that improves the performance of the FSO link as atmospheric disturbance affect its performance directly. Results show the outstanding performance of the APDAGC in terms of BER and intensity fluctuation. In addition, it can be on an airship platform for 5G FSO data transmission. All the graphs confirm that adaptive gain control performs better than fixed gain control; the actual BER is better than the theoretical calculation result. In terms of numbers, the established 5G two-way FSO link provides feasibility for the air platform BS to transmit 2.5 Gbps laser signal from airship to ground with 854 and 149 Mbps downlink and uplink transmissions, respectively.

However, Singh et al. [249] analyze the performance of a 16 × 10 Gbps WDM FSO system for the execution of 5G communication in various regions in India such as the hilly, plain, coastal, and desert regions. The proposed project utilizes digital signal processing and OFDM to weaken the channel induces limitations. Furthermore, it has EDFA with optimized gain characteristics. Results confirm that the link can be 10.75 km with 10^−9^ BER in coastal areas such as Chennai with weak weather disturbances. Contrastingly, in plain and desert areas, the FSO link of ranges 6.53 km and 7.85 km have the same BER of 10^−9^. However, in hilly areas the FSO link is only 5.3 km. The performance of the system is ideal to be used in 5G mobile networks and IoT technology as it provides high-speed connectivity. The data transmission rate of this system can reach 160 Gbps. The links in this system face some difficulties such as installing an optical fiber network in some places such as mountains which can be solved by installing FSO links with a backup hybrid RF/FSO system.

Zhao et al. [250], experimentally propose and demonstrate a 200 Gbps FSO WDM 5G communication system based on multiwavelength directly modulated TOSA, as it simply transmits eight laser beams with different wavelengths varying from 1545 nm to 1557 nm. The authors confirm that 4-PAM modulation format enhances the system capacity. In other words, the outdoor experiment achieves 25 Gbps with 4-PAM modulation. Furthermore, one-hop and forwarding systems are tested in terms of eye diagrams and BER, respectively, with 8 × 25 Gbps for 50 m outdoor transmission between the two buildings. Results show clear eye diagrams and reliable BER performance for all the wavelengths in the one-hop system. However, the BER for the forwarding system varies from 1E-2 to 1E-3 for 30 min. The system performance of the system is suitable for FSO fronthaul/backhaul communication system for 5G wireless cellular networks and optical access network extension as it provides low power consumption with a channelization scheme with high link reliability.

Additionally, Schulz et al. [251], introduce a new work using optical links to provide a robust outdoor backhaul and fronthaul for 4G and 5G small cells. The distance of links varies from 20 to 200 m. The work was conducted in Berlin, Germany, for 5 months, where the visibility was never below 180 m and 99% of the time was above 1 km. Results confirm that rate-adaptive transmission boosts the availability of the communication system in the presence of fog or sunlight. In real-time operation, the prototype achieves 800, 500, and 225 Mb/s over 20, 100, and 215 m link distance, respectively, with 2 ms latency at 95% load using low-cost optoelectronic components and a 1 Gbps baseband chipset. This project confirms that optical wireless communication systems can be a low-cost wireless backhaul solution for small radio cells such as WiFi, LTE, and 5G. To achieve 5G connectivity, high data rate and low latency are needed, which can be reached by using low-cost laser diodes instead of LEDs.

## 9. Conclusions

We can conclude from all the mentioned projects that hybrid FSO/RF communication systems perform better than FSO or RF standalone systems as it relies on more than one link in case one link is facing some difficulties such as severe weather conditions and achieves full diversity. In addition, a hybrid system with parallel links outperforms FSO-RF serial link, as a parallel hybrid FSO/RF system uses two or more independent paths to deliver the signals to the receiver. A relay is required for serial hybrid FSO/RF systems as it converts the signals from FSO to RF or vice versa, such as access points allow RF users to access the whole network. DF relay performs better than AF relay; it enhances the BER performance compared to AF relay. However, the AF relay is more straightforward and less complex than the DF relay.

In addition, external modulation is superior to direct modulation. For instance, return to zero (RZ) modulation is suitable for long-distance communication but is both difficult and expensive. NRZ, however, is more appropriate for short links, simpler, and more affordable. As stated in Table 8, the model must meet the criteria for classification as a 4G, 5G, or 6G system.

The massive increase in internet traffic and multimedia users over the last several years has put significant pressure on RF systems that operate at modest data rates. There is a need to go from RF domain to the optical domain because of the enormous development in information technology, which is pushing the information industry to higher and higher data rates. An extremely high bandwidth LOS wireless link between faraway locations is possible using FSO communication. This technology is thought to be one that will soon be able to satisfy the extremely high speed and enormous capacity demands of the modern communication industry. However, the heterogeneous nature of the air channel presents several difficulties that must be solved to fully exploit the FSO system’s terabit capability. The FSO system is susceptible to many atmospheric phenomena, including absorption, scattering, and atmospheric turbulence. Numerous methods used at the physical and network layers reduce the negative impact of the atmosphere on the laser beam’s quality. Numerous fading mitigation strategies first developed for RF communication, including diversity, adaptive optics, error control codes, modulation, etc., work well for FSO communication. In addition, the creation of a hybrid RF/FSO system, which guarantees carrier class availability for practically all weather circumstances, was prompted by the complementary nature of RF and FSO. It follows that, given the significant progress made in FSO communication, this technology seems to have highly promising development prospects in the near term. There are now some commercial products for FSO terrestrial and space lines on the market. We hope this technology will soon usher in a global revolution in telecommunications.

## Figures and Tables

**Figure 1 entropy-24-01573-f001:**
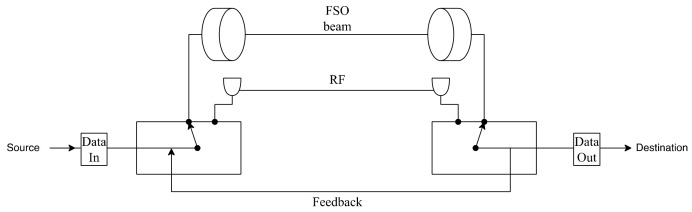
System block diagram of a hybrid RF/FSO system.

**Figure 2 entropy-24-01573-f002:**
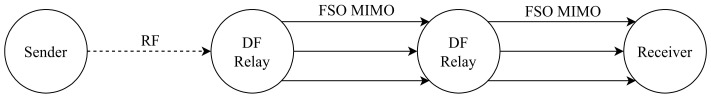
Hybrid DF multi-hop RF/MIMO FSO network with multiple optical sources and multiple optical detectors.

**Figure 3 entropy-24-01573-f003:**
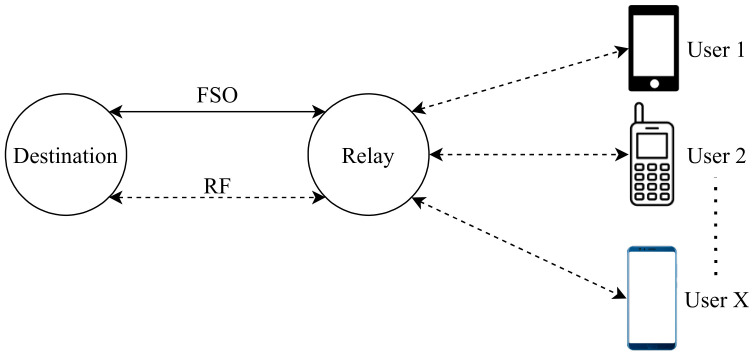
Mixed RF and hybrid RF/FSO relay network model.

**Figure 4 entropy-24-01573-f004:**
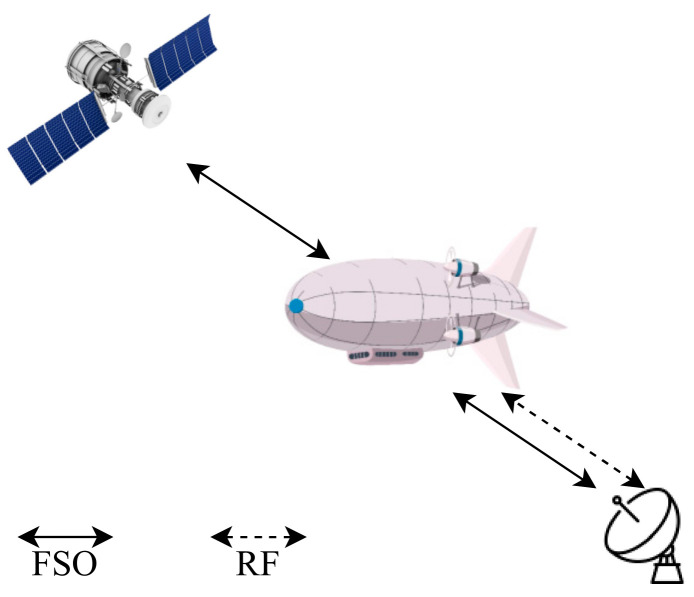
Dual-hop space-air-ground integrated network-based hybrid FSO/RF SATCOM.

**Figure 5 entropy-24-01573-f005:**
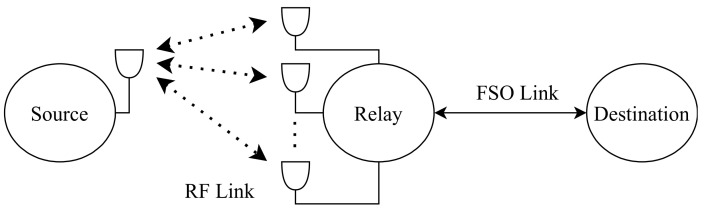
Hybrid SIMO-RF/FSO communication system.

**Figure 6 entropy-24-01573-f006:**
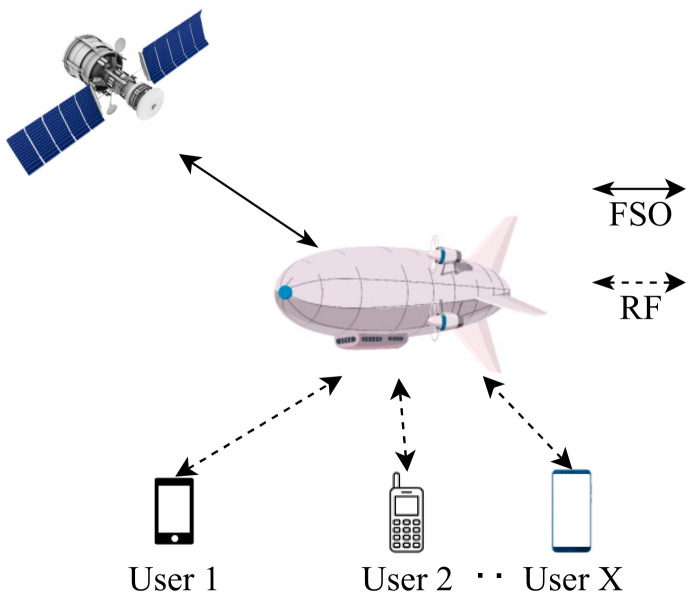
A system model of SATN.

**Figure 7 entropy-24-01573-f007:**
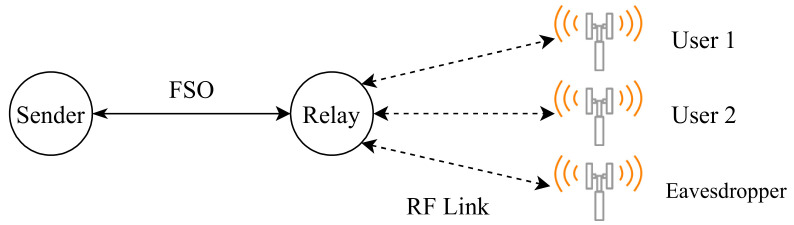
System model [140].

**Figure 8 entropy-24-01573-f008:**
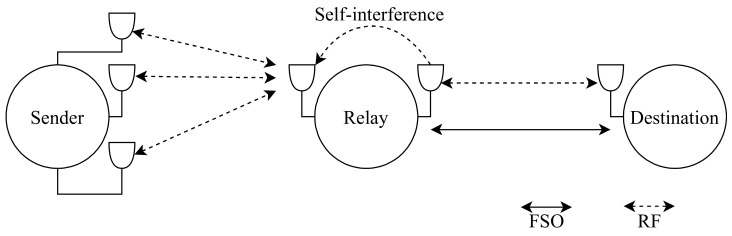
Full-Duplex relay-assisted mixed RF/FSO system with self-interference.

**Figure 9 entropy-24-01573-f009:**
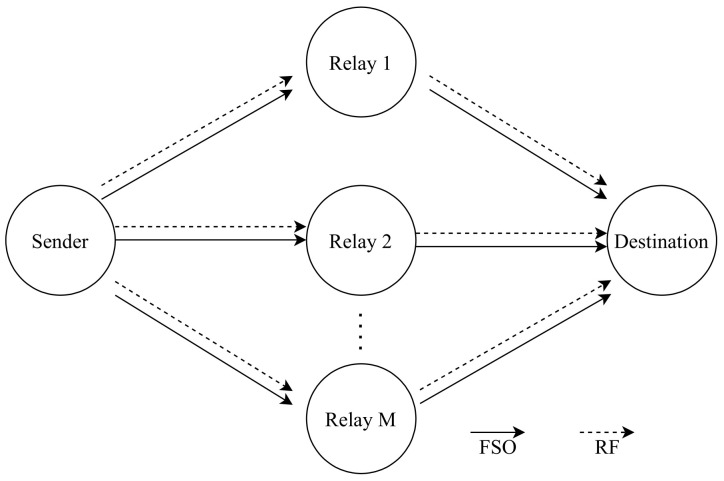
Schematic representation of a parallel hybrid RF/FSO relay system.

**Figure 10 entropy-24-01573-f010:**
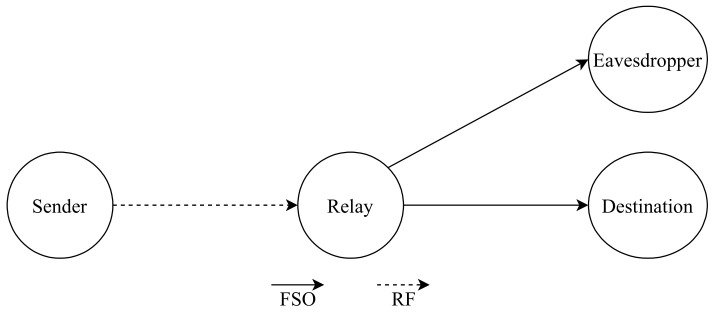
Combined RF-FSO DF-based relaying system.

**Figure 11 entropy-24-01573-f011:**
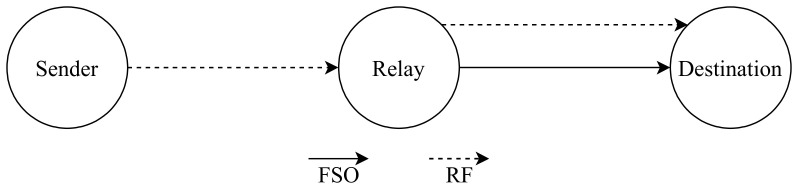
RF-RF/FSO system model.

**Figure 12 entropy-24-01573-f012:**
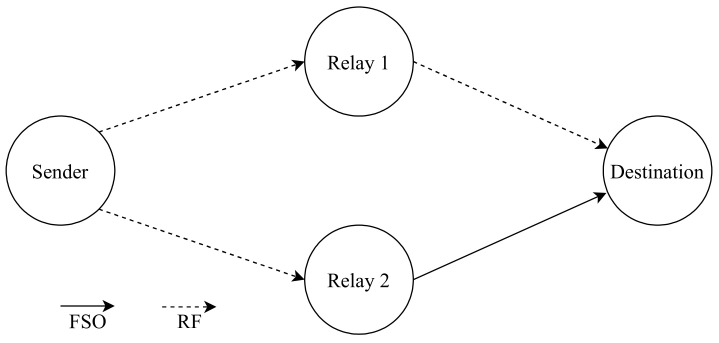
RF-RF/RF-FSO System model.

**Figure 13 entropy-24-01573-f013:**
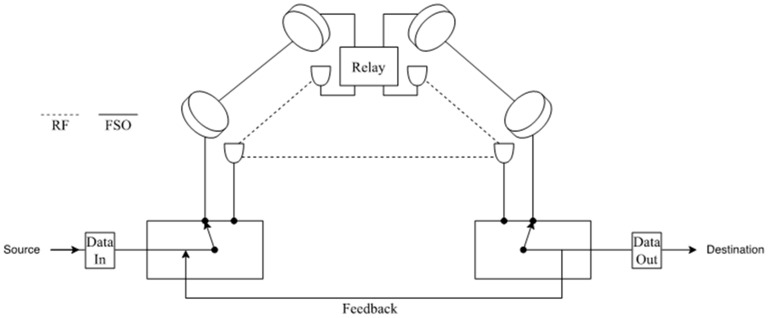
A system model of dual-hop hybrid FSO/RF system.

**Figure 14 entropy-24-01573-f014:**
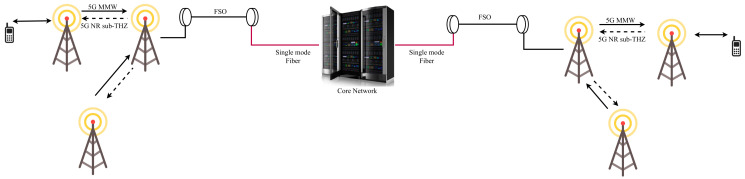
A bi-directional fiber-FSO-5G MMW/5G new radio (NR) sub-THz convergence with parallel/orthogonally polarized dual-carrier mechanism.

**Figure 15 entropy-24-01573-f015:**
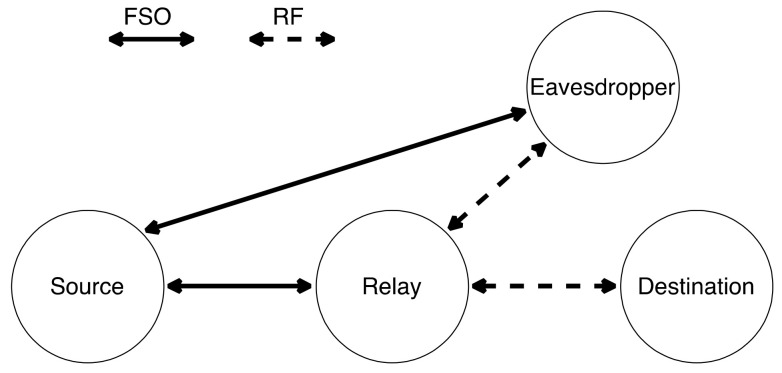
A two-way communication-based mixed FSO/RF system.

**Figure 16 entropy-24-01573-f016:**
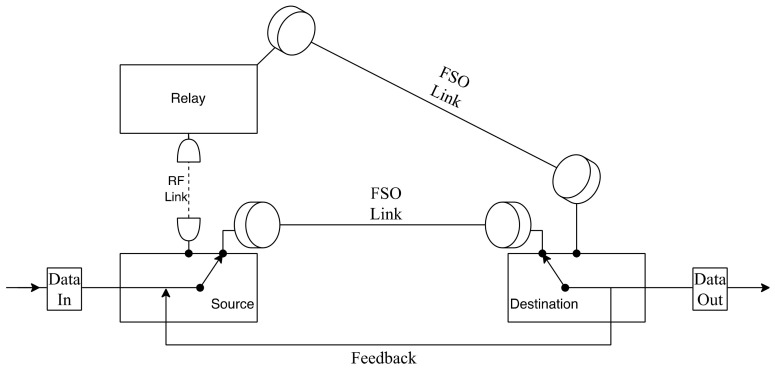
Dual-hop mixed RF-FSO backup link with source-to-relay RF link and relay-to-destination FSO link.

**Figure 17 entropy-24-01573-f017:**
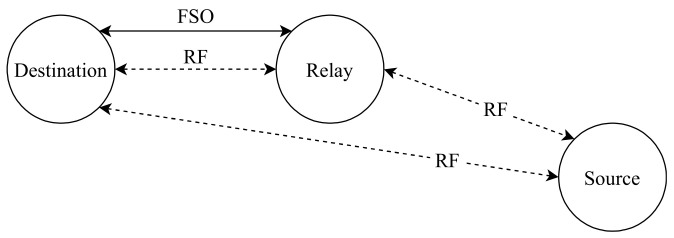
RF-FSO/RF communication system.

**Figure 18 entropy-24-01573-f018:**
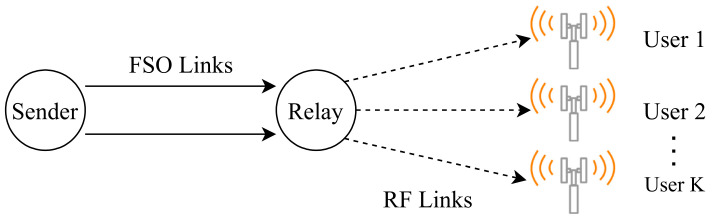
Dual-hop MMW multiuser mixed FSO-RF relay network with opportunistic user scheduling.

**Figure 19 entropy-24-01573-f019:**
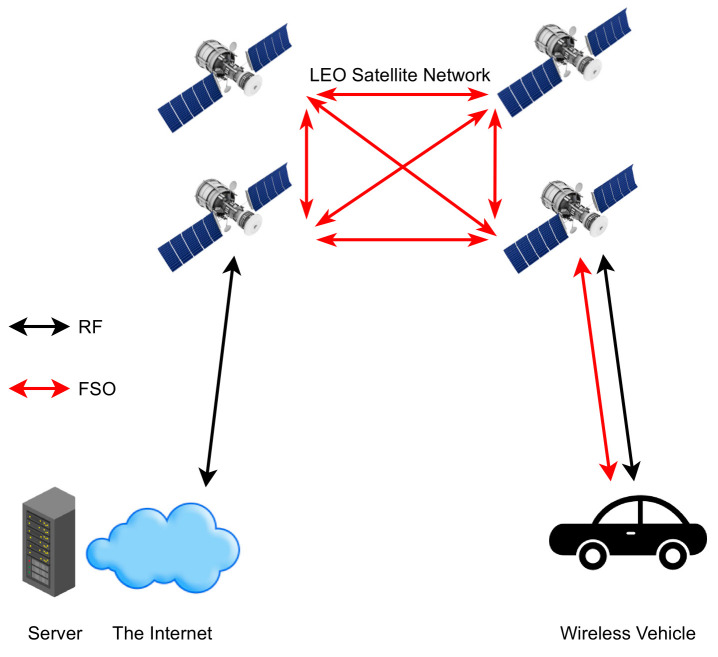
Last-mile access network.

**Figure 20 entropy-24-01573-f020:**
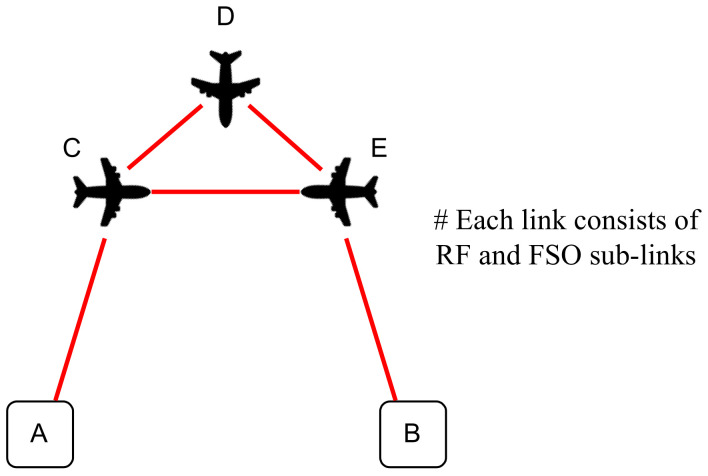
Hybrid broadband communication system.

**Figure 21 entropy-24-01573-f021:**
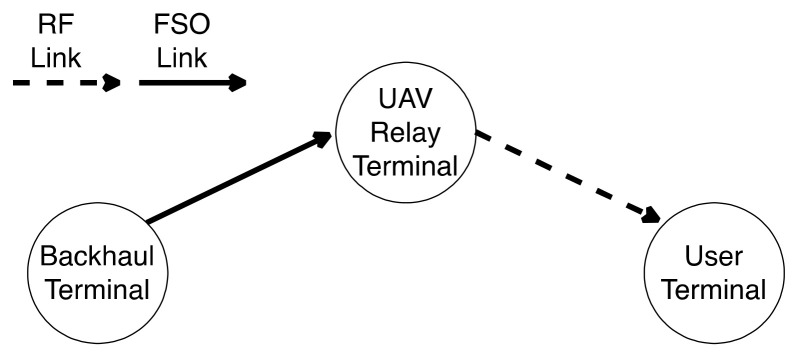
A UAV-assisted mobile relaying with dual-hop mixed FSO/RF communication.

**Figure 22 entropy-24-01573-f022:**
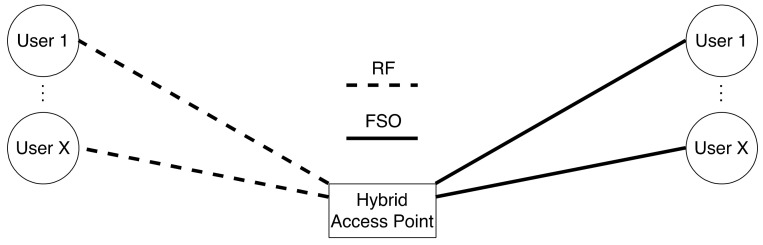
Hybrid FSO/RF multiuser system model.

**Figure 23 entropy-24-01573-f023:**
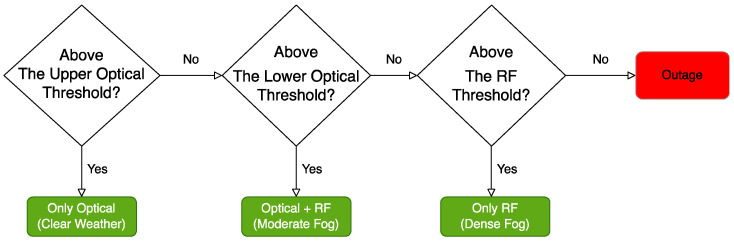
Switching scheme with two optical thresholds and one RF threshold.

**Figure 24 entropy-24-01573-f024:**
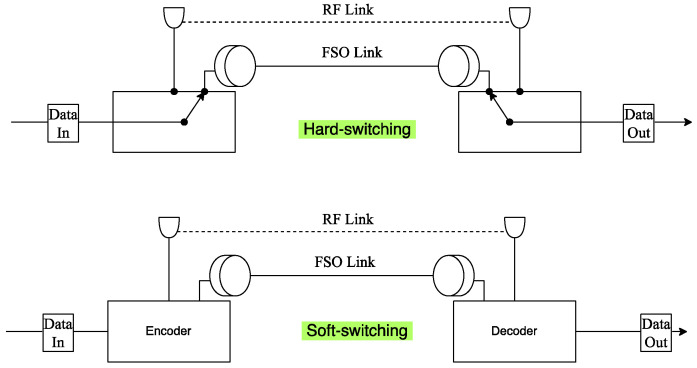
Hard- and soft-switching configurations for Hybrid FSO/RF links.

**Figure 25 entropy-24-01573-f025:**
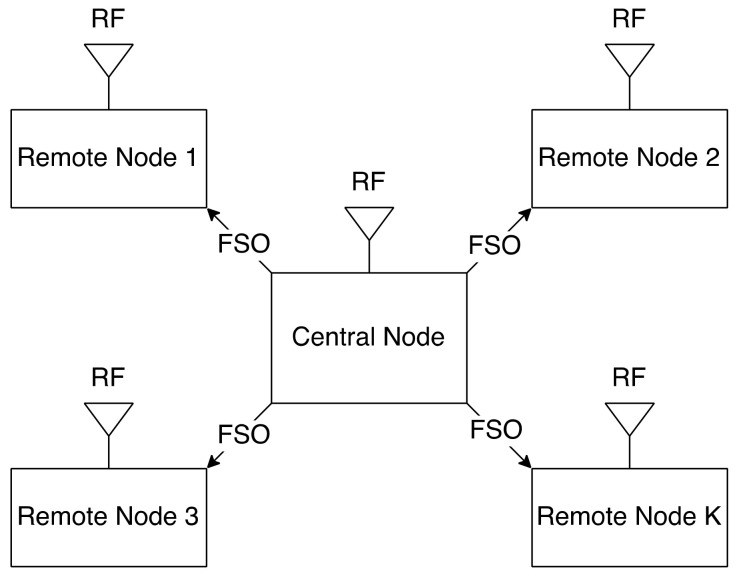
Block diagram of a P2MP hybrid FSO/RF system.

**Figure 26 entropy-24-01573-f026:**
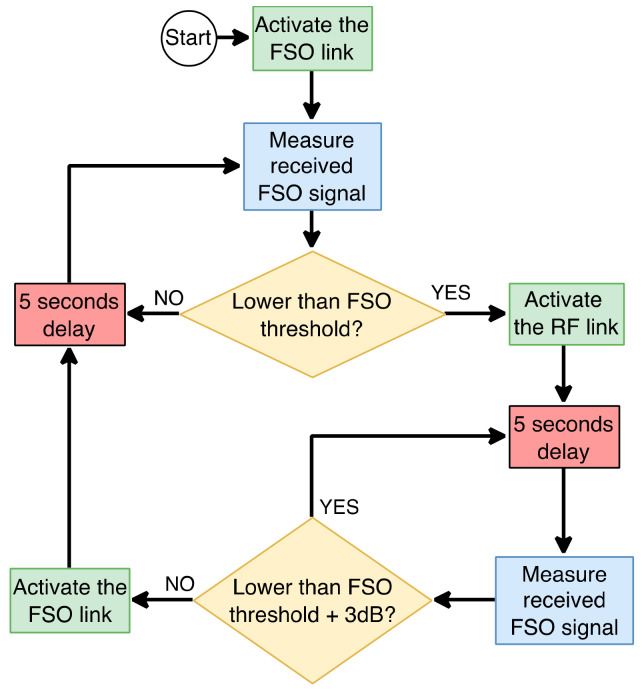
Flowchart of the PH algorithm.

**Figure 27 entropy-24-01573-f027:**
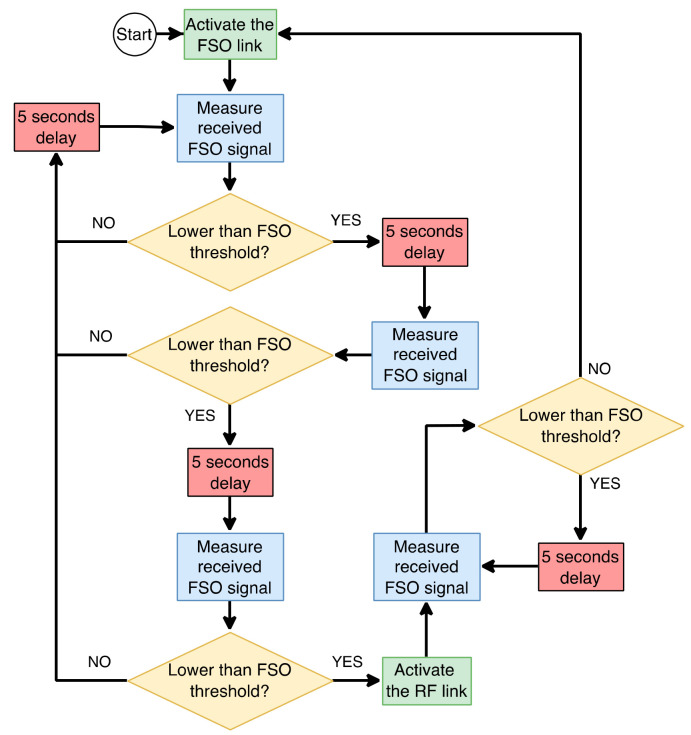
Flowchart of the TH algorithm.

**Figure 28 entropy-24-01573-f028:**
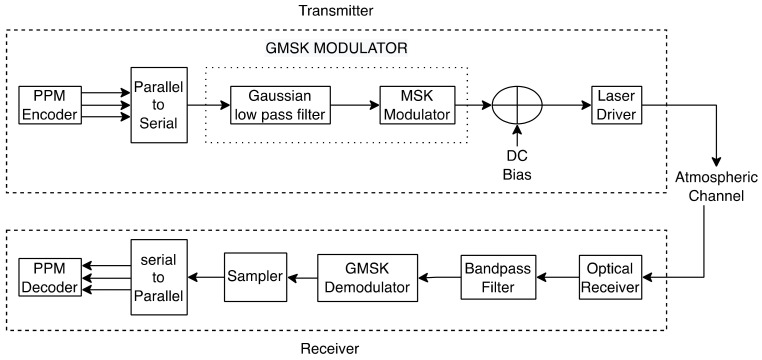
Block diagram of hybrid PPM-GMSK FSO communication system.

**Table 1 entropy-24-01573-t001:** Properties of terrestrial FSO and RF access links communications.

	FSO Links	RF Links
Typical Data Rate	100 Mbps to ~Gbps	Less than 100 Mbps
Channel Security	High	Low
Component Dimension	Small	Large
Networking Architecture	Scalable	Non-scalable
Source of Signal Degradation	Atmospheric turbulence, obscuration, pointing error and geometric losses	Multipath fading, rain, and user interferences
Link Distance	Short-range	Long-range
Superiority	Unlicensed Band	Non line of sight

**Table 3 entropy-24-01573-t003:** Comparison of small cells in 5G networks [103,104].

Type	Power (W)	Coverage Radius (km)	Users	Location
Picocell	0.1–0.2	0.01–0.05	4–16	Indoor public areas
Femtocell	0.25	0.2	32–100	High-capacity enterprise areas
Microcell	2–10	2	200	Utilized outdoor to increase signal coverage in locations that macro BSs do not or only partially reach
Macrocell	40–100	10–40	1000+	Outdoor

**Table 4 entropy-24-01573-t004:** Difference Between SISO, SIMO, MISO, and MIMO.

	SISO	SIMO	MISO	MIMO
Definition	Single-Input Single-Output	Single-Input Multiple-Output	Multiple-Input Single-Output	Multiple-Input Multiple-Output
Number of Transmitter Antennas	One	One	Multiple	Multiple
Number of Receiver Antennas	One	Multiple	One	Multiple
Channel Capacity	Low	Better than SISO	Better than SIMO	High

**Table 5 entropy-24-01573-t005:** Comparison of the relay-assisted models.

Ref.	FSO Link Distance (km)	RF Link Distance (km)	Max Data Rate	Fading Distribution	FSO Wavelength (nm)	RF Frequency (GHz)	Relay Type	Simulation Software	DC Bias
[106]	8	Na	750 Mbps	GG	785	Na	Na	Monte-Carlo	
[108]	1	1	NDA	GG	NDA	NDA	DF	NDA	
[110]	0.5	NDA	5 Byte/s/Hz	GG/Rayleigh	NDA	NDA	DF	Matlab	
[111]	NDA	NDA	7.23 Byte/s/Hz	GG/Rayleigh	1550	NDA	DF	NDA	✓
[112]	0.2	0.2	16 bit/s/Hz	GG/Rayleigh	1550	150	NDA	NDA	
[113]	25	25	NDA	GG/Rician	NDA	NDA	NDA	Monte-Carlo	
[114]	1	NDA	12 bit/s/Hz	M/Rayleigh	785	NDA	AF	Monte-Carlo	
[115]	NDA	NDA	13 bit/s/Hz	GG/Rayleigh	NDA	NDA	AF	NDA	
[133,134]	NDA	NDA	NDA	GG/Rayleigh	NDA	NDA	DF	Matlab	
[136]	NDA	20	70 bit/s/Hz	GG/Nakagami-m	1550	1.9	AF	Matlab	
[137]	3	3	1.8 bit/s/Hz	GG/Nakagami-m	1550	60	DF	Monte-Carlo	
[138]	NDA	NDA	NDA	GG/Nakagami-m	1550	60	DF	Monte-Carlo	✓
[140]	NDA	NDA	1.8 bit/s/Hz	GG/Nakagami-m	NDA	NDA	AF	Monte-Carlo	
[142]	NDA	NDA	NDA	GG/Nakagami-m	NDA	NDA	NDA	Monte-Carlo	
[144]	1	1	NDA	GG/Rayleigh	1550	NDA	AF	Monte-Carlo	✓
[146]	NDA	NDA	NDA	GG/Rayleigh	NDA	NDA	NDA	Monte-Carlo	✓
[147]	1	1	1Gbps	GG/Rician	1550	3.5	DF	NDA	
[149]	NDA	NDA	NDA	GG/Nakagami-m	NDA	NDA	AF	Monte-Carlo	✓
[150]	NDA	NDA	9.2 bit/s/Hz	GG/Nakagami-m	NDA	NDA	AF	NDA	
[151]	NDA	NDA	NDA	GG/K	NDA	NDA	AF	Monte-Carlo	
[152]	NDA	NDA	0.5 bit/s/Hz	Malaga/AKM	NDA	NDA	DF	Monte-Carlo	
[153]	3.8	3.8	4.5 bit/s/Hz	GG/Nakagami-m	1550	60	DF	Monte-Carlo	
[154]	NDA	NDA	4 bit/s/Hz	GG/Rayleigh	NDA	NDA	AF	Monte-Carlo	
[155]	NDA	NDA	NDA	GG/Rayleigh	NDA	NDA	AF	Matlab	✓
[156]	NDA	NDA	NDA	α-μ/Rayleigh	NDA	NDA	DF	NDA	
[157]	1	1	NDA	Log-normal/Rayleigh	NDA	NDA	DF	NDA	
[158]	NDA	NDA	NDA	Malaga/Weibull	NDA	NDA	DF	NDA	
[159]	NDA	NDA	1.5 Gbps	GG/Rayleigh	NDA	NDA	AF	NDA	
[160]	O.5	0.001	40 Gbps	NDA	1550	100	NA	NDA	
[161]	NDA	NDA	55 Gbps	Log-normal/Rician	1550	10	DF	NDA	
[162]	NDA	NDA	20 bit/s/Hz	Malaga/Rician	NDA	28	AF	Monte-Carlo	✓
[163]	NDA	NDA	1 bit/s/Hz	Malaga/Nakagami-m	NDA	NDA	DF	Monte-Carlo	
[171]	NDA	NDA	NDA	GG/Rician	NDA	NDA	DF	Matlab	
[165]	NDA	NDA	9 bit/s/Hz	GG/Rayleigh	NDA	NDA	AF	Monte-Carlo	
[166]	NDA	NDA	NDA	GG/Rayleigh	NDA	NDA	QER	NDA	
[167]	0.5	NDA	6 bit/s/Hz	GG/Rayleigh	NDA	NDA	NDA	Monte-Carlo	
[168]	NDA	NDA	NDA	NDA	670	2.4	Switch	Matlab	
[169]	NDA	0.005	NDA	GG/Nakagami-m	NDA	28 and 38	AF	Monte-Carlo	
[170]	600	600	900 Mb/s	GG/Nakagami-m	1550	2.4	NDA	Monte-Carlo	
[36]	NDA	NDA	NDA	GG/Negative Exponential	NDA	NDA	AF	Matlab	

**Table 6 entropy-24-01573-t006:** Summary of projects used Monte-Carlo, Matlab and DC Bias.

	Ref.
Monte-Carlo	[182,192,198,228,229,239]
Matlab	[198,199,230,244]
DC Bias	[191,207]

**Table 7 entropy-24-01573-t007:** Summary of fading distribution used in FSO and RF links.

Fading Distribution	FSO Link	RF Link
GG	[171,175,210,182,192,193,198,203,207,208,209,220,222,228,229,240,242]	
Rayleigh		[175,193,198,199,222]
Weibull	[183]	[236]
Malaga-M	[173,225]	
Nakagami-m		[183,192,201,207,220,225,228,229,191]
Log-Normal	[191,219,239,161]	
Negative Exponential	[199]	
Rician		[203,208,209,171]

**Table 8 entropy-24-01573-t008:** Comparison of 4G, 5G, and 6G systems.

Parameter	4G	5G	6G
Peak Data Rate	1 Gbps	10 Gbps	1 Tbps
Technology	LTE, Wi-max	IPv4	IPv6
End-To-End Latency	100 ms	10 ms	1 ms
Satellite Integration	No	No	Fully
Artificial Intelligence	No	Partial	Fully
Driverless Vehicle	No	Partial	Fully
Haptic	No	Partial	Fully
Mobility	350 km/h	500 km/h	1000 km/h
Connection Density	$10^{5}$ devices $/{\mathrm{km}}^{2}$	$10^{6}$ devices $/{\mathrm{km}}^{2}$	$10^{7}$ devices $/{\mathrm{km}}^{2}$
Maximum Frequency	6 GHz	90 GHz	10 THz

## Data Availability

The data presented in this study are available upon request from the corresponding author.

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
