# Peer review of "A Survey of Hybrid Free Space Optics (FSO) Communication Networks to Achieve 5G Connectivity for Backhauling"

_entropy, 2022, doi:10.3390/e24111573_

Round 1

Reviewer 1 Report

This manuscript aims to present a review of hybrid RF/Optical wireless communications research to summarize the state of the field with regard to 5G connectivity. Data capacity and number of connected users are always increasing and RF alone will not be able to meet the demand. A review like this could be valuable and timely resource, however the manuscript in its present state is not mature enough for peer-review, let alone publication. Although the authors have thoroughly researched the literature, it is clear they have not proof-read the manuscript before submission. It is full of errors, inconsistencies, and superfluous or repeated statements that severely interrupt the flow making it difficult to read. For example:

- In the introduction where figures are referenced in the text, entire captions are present in the middle of the sentence. There is also a figure with a referencing error (line 376).

- The entire final paragraph of section 3 is repeated from the introduction.

- The first paragraph of section 3 is almost nonsensical. Also: "Terrestrial FSO links can also be utilized in wireless sensor networks with many nodes spread out over a big area and requiring non-line-of-sight (NLOS) communication" - FSO requires LOS so I don't know what this sentence is saying. This paragraph should be removed entirely.

- Acronyms are used before definition, are defined repeatedly, or not defined at all. Abbreviating "source" and "destination" is unnecessary and only worsens readability. Multiple abbreviations are used for the same thing: MMW or mmWave? FSO or OWC? Be consistent.

- Information is often repeated within the same paragraph. For example, the first paragraph of section 4.1 repeats the sentence "Photodetectors in this project neglect ambient light."

- Section 2.1: "According to a recent study, diffused laser light does not improve the user experience when compared to traditional light luminaires [72]" - What does this have to do with FSO communications??

- Section 2.4 - There is no mention of coherent modulation for FSO signals, which seems like a huge oversight given coherent modulation is standard in fibre communications and is being demonstrated in FSOC.

- Table 2 says that spectrum licensing is required for optical fiber, it is obviously not.

- Last paragraph of section 4.1 doesn't seem to be in the right place.

- Referencing is inconsistent, stick with Lastname et al [XX], don't use first names. Also, don't use "This paper [XX]" to refer to other papers, especially when starting a new paragraph introducing a new paper. "This paper" refers to the paper currently being read.

- Fig 20 doesn't show RF signals?

- section 6: What is an X system? Never defined. Do you mean X-band?

- Fourth paragraph of section 6: "FSO adopts IM/DD using a quadrature modulation scheme" - QAM is a coherent modulation scheme, it is not IM/DD. Also, this paragraph (among others) describes the model but doesn't state any results or make any comments about it.

- Fifth paragraph of section 6 talks about the adaptive optics system used, but doesn't use the term "adaptive optics" or what its purpose is. There is no mention of atmospheric turbulence. It's very out of place. What is IFoF? This is not defined and does not appear anywhere else in the text.

These are just a few examples. The paper is riddled with issues like this. Some other comments:

- Some described studies have diagrams while others don't, but it would be very useful if all architectures described had diagrams. 

- When modulation is mentioned, it often doesn't state if its referring to RF or optical. Since the paper never discusses coherent optical modulation, I assume its referring to RF. 

- The only figures in the paper are system diagrams. There are no graphs or tables of results. The walls of text are monotonous and it would be very hard to find any relevant information.

I also have some more fundamental concerns regarding this manuscript. A review paper should serve as a useful first reference that collects current results and summarizes the state-of-the-art. Ideally it should also distil the collected references into a self-contained introduction to the topic. A reader should only have to look up a referred paper to find more in-depth technical information. This paper however, almost never presents any quantitative results; a rote description of each study is presented, simply followed up with a vague statement to the effect of "results confirm a hybrid system is more reliable than a non-hybrid system". It isn't necessary in a review article to state for every study cited that "FSO and RF obeyed gamma-gamma and Rayleigh distributions", that's the level of technical information someone can look up in the reference if needed. It's also unnecessary to repeat multiple times in random paragraphs that FSO is affected by clouds etc. There is too much irrelevant methodology information stated and no quantitative results. Some paragraphs include component details with manufacturer names, others do not. It's also not clear from the text if the studies being described are theoretical or experimental.

A lot of studies are cited, but the manuscript is just a collection of unrelated paragraphs, with no effort to provide context or weave the information into a coherent presentation and draw overall conclusions; there is no summary paragraph or table at the end of each section comparing difference schemes. The first paragraph of section 7 is particularly jarring and offers no context.

Ultimately, the concept of the manuscript is valid, but in its current state it is far from publishable. The authors need to understand the referenced material and present the important results it in a coherent manner, not merely recite a list of studies with no interpretation or comparison of results. The conclusion is little more than common sense, I have no more insight now than I did before reading the manuscript and certainly wouldn't refer to it for any specific information.

I cannot recommend this manuscript for publication without substantial reworking to the presentation and interpretation of the papers reviewed. It must also be proof-read for simple errors as well as overall flow, by the authors and preferably other colleagues of theirs before being resubmitted for peer review. Why should reviewers spend days reading this when the authors haven't read it themselves?

Author Response

Response to Reviewer 1 Comments

Point 1:     This manuscript aims to present a review of hybrid RF/Optical wireless communications research to summarize the state of the field with regard to 5G connectivity. Data capacity and number of connected users are always increasing and RF alone will not be able to meet the demand. A review like this could be valuable and timely resource, however the manuscript in its present state is not mature enough for peer-review, let alone publication. Although the authors have thoroughly researched the literature, it is clear they have not proof-read the manuscript before submission. It is full of errors, inconsistencies, and superfluous or repeated statements that severely interrupt the flow making it difficult to read. For example:
In the introduction where figures are referenced in the text, entire captions are present in the middle of the sentence. There is also a figure with a referencing error (line 376).

Response 1:          •We are appreciative of your well-considered analysis. The authors have made numerous edits to the manuscript, including adding full stops and commas where necessary.
The reference problem in the original file has been fixed (line 376) in the revised manuscript (Page 8, line 356).

Point 2:     The entire final paragraph of section 3 is repeated from the introduction.

Response 2:        •Your feedback has been carefully considered, as it has been given a lot of thought and appreciation. The authors have gotten rid of the repetitive paragraph that was in section 3 because it was unnecessary. The paragraph is just included in the introduction (Page 3, line 100).

Point 3:     The first paragraph of section 3 is almost nonsensical. Also: "Terrestrial FSO links can also be utilized in wireless sensor networks with many nodes spread out over a big area and requiring non-line-of-sight (NLOS) communication" - FSO requires LOS so I don't know what this sentence is saying. This paragraph should be removed entirely.

Response 3:          •We are grateful that you are concerned. Your feedback was taken into consideration, and the relevant paragraph was revised. The entire first paragraph has been taken out of the article (Page 7, line 306).

Point 4:     - Acronyms are used before definition, are defined repeatedly, or not defined at all. Abbreviating "source" and "destination" is unnecessary and only worsens readability. Multiple abbreviations are used for the same thing: MMW or mmWave? FSO or OWC? Be consistent.

Response 4:          •We would like to offer our profound gratitude for the effort that you have given in providing this comment. Throughout the entire work, the authors changed mmWave to MMW (Page 20, line 838)
•and changed S to source, R to relay and D to destination (Page 10, line 423-424)
•and OWC to FSO. (Page 2, line 63)
•The only place where we have briefly referenced OWC is in section 2.1. (Page 4, line 178-185).

Point 5:     Information is often repeated within the same paragraph. For example, the first paragraph of section 4.1 repeats the sentence "Photodetectors in this project neglect ambient light."

Response 5:          •Authors would like to express our appreciation for your review.
The authors have removed the repeated sentence and it only appears once (Page 9, line 402).

Point 6:     Section 2.1: "According to a recent study, diffused laser light does not improve the user experience when compared to traditional light luminaires [72]" - What does this have to do with FSO communications??

Response 6:          •Thank you for your comment after reviewing the paper, the Authors have removed the sentence accordingly (Page 5, line 209).

Point 7:     Section 2.4 - There is no mention of coherent modulation for FSO signals, which seems like a huge oversight given coherent modulation is standard in fibre communications and is being demonstrated in FSOC.

Response 7:          •Your comment is much appreciated! Authors have added “There are coherent and non-coherent optical communication systems. Non-coherent optical transmission systems use amplitude and differential phase modulations, which do not require coherent local oscillator light, whereas coherent optical transmission systems use phase and quadrature amplitude modulations for coherent detection.” To the beginning of the second paragraph in section 2.4 (Page 6, line 273).

Point 8:     Table 2 says that spectrum licensing is required for optical fiber, it is obviously not.

Response 8:          •We have doubled checked our information. Your cooperation is highly appreciated! Authors have changed “spectrum licensing is required for optical fiber” to “not required” (Page 8, line 357).

Point 9:     Last paragraph of section 4.1 doesn't seem to be in the right place.

Response 9:          •Thank you for your response. The authors have Moved the paragraph to the end of section 3 (Page 8, line 362)

Point 10:  Referencing is inconsistent, stick with Lastname et al [XX], don't use first names. Also, don't use "This paper [XX]" to refer to other papers, especially when starting a new paragraph introducing a new paper. "This paper" refers to the paper currently being read.

Response 10:        •Your coordination is much appreciated! Authors have Changed all references to “Lastname et al. [XX]” (Page 5, line 218)

“Lastname in [XX]” for only one author (Page 8, line 359)
and “1stLastName and 2ndLastName [XX]” if they are two authors (Page 14, line 614)

Point 11:  Fig 20 doesn't show RF signals?

Response 11:        •Thank you for your comment! Authors have declared “Each link consists of RF and FSO sub-links” in figure 20. (Page 26, line 1050)

Point 12:  section 6: What is an X system? Never defined. Do you mean X-band?

Response 12:        •Thank you for your inquiry. The authors have Changed section 6 to “Hybrid FSO Networks” (Page 28, line 1131)
•and added “In this section, we will highlight different models that utilize various sub-systems with FSO communication systems. To fully utilize the benefits of hybrid FSO systems, various issues must be resolved.” At the beginning of section 6. (Page 28, line 1132)

Point 13:  Fourth paragraph of section 6: "FSO adopts IM/DD using a quadrature modulation scheme" - QAM is a coherent modulation scheme, it is not IM/DD. Also, this paragraph (among others) describes the model but doesn't state any results or make any comments about it.

Response 13:        •Your assistance in reviewing this paper is much appreciated! The authors have changed the sentence to “The FSO link uses quadrature modulation along with IM/DD” (Page 30, line 1219)
•and added the reference. Explained the results briefly, and how the system’s performance can be improved. (Page 30, line 1227)

Point 14:  Fifth paragraph of section 6 talks about the adaptive optics system used, but doesn't use the term "adaptive optics" or what its purpose is. There is no mention of atmospheric turbulence. It's very out of place. What is IFoF? This is not defined and does not appear anywhere else in the text.         

Response 14:        •Thank you for your deep review of the paper. Authors have added “With this switching approach, the RF link is only enabled via a 1-bit feedback signal when the instantaneous SNR of the FSO connection at the receiver is less than a switching threshold. •The FSO link remains active throughout.” (Page 30, line 1238)
•intermediate-frequency-over-fiber (Page 30, line 1248)
• Erbium-Doped Fiber Amplifier (EDFA) (Page 31, line 1255)
•and “In addition, to verify the effectiveness of the developed FSO and hybrid systems, they will undertake experiments in an outdoor setting in the upcoming works under various weather conditions.” (Page 31, line 1275)

Point 15:  Some described studies have diagrams while others don't, but it would be very useful if all architectures described had diagrams.           

Response 15:        •Thank you for your concern, Authors am explaining with figures when it is using a new concept among others. Authors would love to explain each model by a diagram and flowchart. However, if Authors describe all models with diagrams, the review paper will be too lengthy and the paper was already 56 pages, and now its longer after the amendments.

Point 16:  When modulation is mentioned, it often doesn't state if its referring to RF or optical. Since the paper never discusses coherent optical modulation, I assume its referring to RF.

Response 16:        •Noted on your comment with thanks, Authors have explained coherent and non-coherent modulation in section 2.4 (Page 6, line 273).

Point 17:  - The only figures in the paper are system diagrams. There are no graphs or tables of results. The walls of text are monotonous and it would be very hard to find any relevant information.

Response 17:        •Authors appreciate your comments and Authors kept them in mind while making the changes. Authors have added table 5 (Page 24, line 1011) to summarize and show the features of each work in section 5, such as FSO and RF links distance, FSO wavelength and RF frequency, maximum data rate, fading distribution, type of relay, simulation software used and which project added a DC bias.
• In addition, the authors have added tables 6 and 7 on page 43 to summarize all projects in section 7.

Point 18:  I also have some more fundamental concerns regarding this manuscript. A review paper should serve as a useful first reference that collects current results and summarizes the state-of-the-art. Ideally it should also distil the collected references into a self-contained introduction to the topic. A reader should only have to look up a referred paper to find more in-depth technical information. This paper however, almost never presents any quantitative results; a rote description of each study is presented, simply followed up with a vague statement to the effect of "results confirm a hybrid system is more reliable than a non-hybrid system". It isn't necessary in a review article to state for every study cited that "FSO and RF obeyed gamma-gamma and Rayleigh distributions", that's the level of technical information someone can look up in the reference if needed. It's also unnecessary to repeat multiple times in random paragraphs that FSO is affected by clouds etc. There is too much irrelevant methodology information stated and no quantitative results. Some paragraphs include component details with manufacturer names, others do not. It's also not clear from the text if the studies being described are theoretical or experimental.

Response 18:        •Noted on your comment. Authors have mentioned all the important details, some papers mention the component details such as raspberry pi, which is important to mention, but others did not mention it.

Point 19:  A lot of studies are cited, but the manuscript is just a collection of unrelated paragraphs, with no effort to provide context or weave the information into a coherent presentation and draw overall conclusions; there is no summary paragraph or table at the end of each section comparing difference schemes. The first paragraph of section 7 is particularly jarring and offers no context.

Response 19:     •Your feedback has been taken into consideration, and the authors are currently revising the manuscript based on the changes they made after reading it and removed the first paragraph of section 7. (Page 41, line 1725)

Point 20:  Ultimately, the concept of the manuscript is valid, but in its current state it is far from publishable. The authors need to understand the referenced material and present the important results it in a coherent manner, not merely recite a list of studies with no interpretation or comparison of results. The conclusion is little more than common sense, I have no more insight now than I did before reading the manuscript and certainly wouldn't refer to it for any specific information.

Response 20:        •We would like to express our sincere gratitude for the effort and time you have spent reviewing my manuscript and giving feedback accordingly. The authors followed your comments, and we believe this manuscript version is better in shape.
•In addition, authors have removed “In order to maintain a competitive pricing point for the capital expenditure, the complexity of the hybrid FSO/RF system implementation needs to be maintained to a minimum. For the next-generation FSO network to support either 16 or 32 channels, as well as either 16 or 32 end users, the design of the network needs to incorporate a high degree of scalability.” from the conclusion and added “The massive increase in internet traffic and multimedia users over the last several years has put significant pressure on RF systems that operate at modest data rates. There is a need to go from RF domain to the optical domain because of the enormous development in information technology, which is pushing the information industry to higher and higher data rates. An extremely high bandwidth LOS wireless link between faraway locations is possible using FSO communication. This technology is thought to be one that will soon be able to satisfy the extremely high speed and enormous capacity demands of the modern communication industry. However, the heterogeneous nature of the air channel presents several difficulties that must be solved to fully exploit the FSO system's terabit capability. The FSO system is susceptible to many atmospheric phenomena, including atmospheric turbulence, absorption, scattering, and atmospheric turbulence. Numerous methods used at the physical and network layers reduce the negative impact of the atmosphere on the laser beam's quality. Numerous fading mitigation strategies first developed for RF communication, including diversity, adaptive optics, error control codes, modulation, etc., work well for FSO communication. In addition, the creation of a hybrid RF/FSO system, which guarantees carrier class availability for practically all weather circumstances, was prompted by the complementary nature of RF and FSO.
It concludes that given the significant progress made in FSO communication, this technology seems to have highly promising development prospects in the near term. There are now some commercial products for FSO terrestrial and space lines on the market. We hope this technology will soon usher in a global revolution in telecommunications.” to the conclusion (Page 46, line 1903-1924)

Point 21:  I cannot recommend this manuscript for publication without substantial reworking to the presentation and interpretation of the papers reviewed. It must also be proof-read for simple errors as well as overall flow, by the authors and preferably other colleagues of theirs before being resubmitted for peer review. Why should reviewers spend days reading this when the authors haven't read it themselves?

Response 21:        •Your final comments are kept into consideration and we understand it after going through the comments. The authors believe your comments helped us a lot to reshape my manuscript. Your cooperation and assistance are highly appreciated!

Reviewer 2 Report

After a nice start and introduction, the paper turns into a bibliographic survey of several published papers. In this point the own result of the Authors is weak. There is no own calculation neither own measurement or experimental result presented. Even though the topic is very interesting, and the cited references are good papers, some novelty should be added, e.g., in form of a systematic and nice comparison of the cited literature. Regarding 5G and mmW radios, newer references must be added, cited information must be updated.

Unfortunately, the paper in its actual form follows a verbal summary of others’ work. I would definitively recommend adding some tables around the end of the paper or into the conclusions. Main parameters could be compared, or summarized, like below – just as an example table – to list the main engineering or experimental parameters and their value or value ranges.

This could make the paper shorter, e.g. Monte-Carlo or Matlab simulation is mentioned in each paragraph as one sentence. In a table like below, there could be a line “simulation or calculation method” and simply a listing Monte Carlo [21, 23, 55] Matlab [44, 99, 77] analytical calculation/ closed form equations [102, 103] no info given [3].

In my opinion the verbal summary of the papers could be shorter and such table(s) could reduce the total length of the paper. In its actual form the paper is rather a book chapter.

FSO part                         RF part                 papers                                      notes

________________________________________________________________________________

Link length                     20 km                             100 m                             [1]. [22] and [145]       just example

Capacity                         more than 20 Gbps      3 kbit/s                          [121]                              table, add a

                                        1-20 Gbps                                                                                                      better one

Delay                              10 ms                             no data available         [7, 14, 66]

(buffer)                          not used

BER

Wavelength /               1550 nm                        na                                    [11, 33, 99]

                                        800 nm                           na

Frequency                     na                                    2.4 GHz                          [55] and [73]

Relay type DF/DetFused/not used/AP

Relay position               Terrestrial / flying / satellite                              T: [1,2,3]

                                                                                                                        S : [6.16,26]                 

etc…

This is just an example, how the verbal comparison could be turned into engineer’s helping guide, to see where we can find the paper of our interest.

Similarly, different topologies are shown in separate figures, but they are spread over the paper. There is no quick comparison available who had loop, SIMO / MIMO, single link or protected link, diversity with/without, etc. As different papers demonstrated different topologies, the direct comparison of the results is difficult for the readers. They must remember and turn 15..20 pages back or forth. From a survey paper the reader expects that at the end there is some quick comparison of the 200+ references.

In these points I miss tables similar to Table 1, 2 or 3.

Text editing comments

Spelling / Grammar

Line 68, the paper has Section 3, but has not Section III.

Line 162, missing point

Line 354, water? matter?

Line 518, incomplete sentence

Figure 10, evesdropper is with double p

Line 804, Monte-Carlo

This is a general comment: Monte-Carlo shall be written with ‘-‘ all over the paper.

Line 880, Morraa or Morra? please check reference [160]

International Telecommunication Union is an institution, its name must be written with capitals in line 1494.

Acronyms

·      Some acronyms are not resolved, e.g.,

SAGIN in Figure 4.

NOMA in lines 656, 665, 670, 674, 1908,

PD in line 732

S (most probably Source / Sender, and not Signal) in line 437, 667

EDFA

CRZ

·      Several acronyms are resolved two or more times, e.g.,

SIM in line 287, 453

PIN in line 248, 261

OFDM in lines 228 and 1872

MMW in 872 and several other lines

FSO in line 67 and 130

SMF

APD

·      Several acronyms are resolved lines after their first appearance, e.g.,

IMD used in line 301, resolved in line 302,
BS, used in line 596, explained in line 657,
D used in line 442, explained in line 524,
OP used in line 448, explained in line 460,
MRC used in line 500, 521 524, explained at 643,
QAM used in line 472, explained in line 1018,
SC used in 628 for selective combining, explained in 634,

Unluckily, SC used later for Small Cells also
SNR used in 265, 477, 644, etc… explained in line 1886,

CDF used in lines 756, 767, etc, explained in line 1322,

PDF used in lines 767, 820, 922, etc., explained in line 1322,

TWR used in line 776, explained in 929

WiFi is incorrectly written as WIFI in Lines 1221, 2024

mmWave, MMW, mm-Wave all mean the same. When it is not a citation, uniform writing is recommended all over the paper. (mmW in citation titles must keep its bibliographical form).

IPv4 and IPv6 are written IPV4 and IPV6 in Table 3.

 As a general recommendation: I would add a small chapter with all acronyms at the end of the paper. Authors must list all abbreviations in alphabetical order, and check if they are referred in IEEE format.

Acronyms must be explained at the first place of their appearance, not several lines later. Dual meaning acronyms are better to be avoided if possible, or explained in the list. Please see a good example list in one MDPI paper at:  https://www.mdpi.com/2076-3417/9/23/5240/htm

It is not the reviewer’s job to find out all the acronyms by careful reading of the manuscript and the cited references. This is a very long survey and it uses hundreds of abbreviations. In my opinion such a list may help both the writing and the reading of the paper. (The paper shall be preferably understandable without using Google or Wikipedia, the paper should help its readers.)

SI units

mW is the correct writing for milliWatt

mw is incorrect at lines 1399, 1400

°C   Celsius must be written with degree sign  - Line 1515

Gbps is well known and used in English. If SI form is used, Gbit/s is recommended. At several places Gb/s is used. Except the citations, I recommend uniform writing of Gbps everywhere. The reason is simple, in several papers write GigaByte/second as GB/s, there is always a doubt if Authors follow the rule or not when abbreviating Bytes and bits as B and b, respectively. In case of Gbit/s or Gbps there is no doubt.

As a general rule, all SI units, and their prefixes must be correct, the paper cannot be published with units written incorrectly.

As a general rule, there is non-braking space between number and unit. 340Mbps in line 1712 is correctly written as 340 Mbps.

References

Line 376 has Error! “Reference source not found” – the paper must not be published with this

Line 2146, Hecht: year of publishing of the book is missing

Line 2055, name of the Author starts with capital, please correct

Line 2155, title doubled

Line, 2196, use title case please, no reason to use all caps

Line, 2244 refers to a table of a paper of Aditi Malik and Preeti Singh, “Free Space Optics: Current Applications and Future Challenges”, Vol. 2015, Article ID 945483, DOI: 10.1155/2015/945483.

Table.2 of this article repeats Table 1 of reference [100], so please refer to the original paper, not only to the Table. Reference shall be given in title of Table 2 too in line 338 as:

“Table 2. Comparison of FSO with Different System [100]”.

Within the text Authors are sometimes referred with full name, sometimes with family names only, sometimes with shortened given name + family name e.g.:

Anirban Bhowal and Rakhesh Singh Kshetrimayum in [134], line 591

Amirabadi & Vakili in [156], line 828

V. U. Pai and B. Sainath in [180], line 1180

I recommend similar style of writing for the names everywhere across the paper.

Positioning figures and comment to figures:

Normally text refers to a figure first, then figure placed after the paragraph referring to the figure. In lines 516-517 Figure 5 is before it is discussed in line 520.

Same comment for Figure 3. Figure 3 shows User X. text refers User K.

Same comment for Figure 11.

Figure 13 could indicate D and S. Data in should have input/ingress arrow.

Figure 16 could indicate D and S. Data in should have input/ingress arrow.

Figure 20 shows airplanes as relays. Line 1147, in the next chapter,  says relay nodes are stationery. What are C, D, E in Figure 20? Stationary or moving relays?

 Tables

Table 3 is printed before its own chapter.

Doubled paragraph/sentences:

Lines 103-114 write the same as lines 378-388.

Lines 396-399 are repeated in Lines 414-418.

Line 409 is repeated in Line 418

The paper is very long, it cannot be accepted in current form with repeated paragraphs.

Exponents In technical text, exponents must not be written ‘in line’. BER of 10-9 means BER of 1.

As 10-9 = 1. Authors wanted to say 10-9, where -9 is superscript.

Conflilct of interest declaration It is missing.

Technical comments – questions:

Figure 8 shows Full-Duplex system, however arrows are unidirectional. Why only uplink is shown? For a FD system I expect bidirectional arrows as in Fig.7.

This is a general comment to all arrows. Arrowheads show signal flow direction. Unidirectional links are one arrow ended lines. Bidirectional (duplex links) must have arrow at both ends.

Photodiode and photodetector are used interchangeably at several places. Strictly speaking, the photodiode must be a ‘diode’, having a p-n junction. p-i-n diodes have junction, they are real diodes, so we must be careful how to bias it. Photodetectors are rather circuits, that may contain a photodiode plus some optical and/or electrical circuitry, like focusing lens, pre-amplifier and/or electrical transimpedance amplifier, matching and biasing circuit. But not all photodetector is obviously a photodiode. Photodetectors can be built without diodes too (e.g., using bulk semiconductor, like a photoconductor, solar-cell or metal-semiconductor-metal construction, etc. whatever can detect a ‘photon’). I recommend using the terms correctly in part 2.3. PIN and APD are real diodes. Photoreceivers built with PIN or APD but contain more than the diode itself are photodetectors, or ‘optical receivers’.

Internet access is bidirectional, in Figure 19 the vehicle has only downlink, the server has up and downlinks too. Why ?

Technical errors:

The paper refers to 5G as ‘next’ , ‘it will provide’ (e.g. lines 26-27). Feel free to use present tense. 5G has been launched already at several countries. I would keep future tense and ‘next’ for 5G+ and 6G in the paper.

Same comment for MIMO, Line 403 says MIMO systems will be ideal…in fact MIMO is already widely used. It is not ‘will be’, it ‘is’ already…

Table 1 states that RF links have capacity less than 100 Mbps. Most probably Authors think legacy backbone links, typically built at 38/39 GHz or below.  But the table in its actual form is not correct. Table title compares “FSO and RF communications” in general, not specifically the access links. For example, 5G uses also RF for subscriber access and also for fixed access, and Gbps speed can be reached in 5G, also it has been experimentally verified. So table title instead of ‘RF communications’ should specify rather like ‘RF access links’ or rather ‘legacy RF access links’. Other way is to correct the 100 Mbps to actual value, e.g. 10 Gbps. Feel free to correct old value of reference [100] (2015) and cite paper below (2022). In line 50 Authors introduce RF band as 3 kHz to 300 GHz, that means RF is used in a sense of radio wave (i.e., not lightwave) communications. Nowadays Gbps RF links are used in backbone too. Please see e.g. “Throughput Estimation of K-zone Gbps Radio Links Operating in the E-band”, Informacije MIDEM, Journal of Microelectronics, Electronic Components and Materials, Vol.52, No.1, pp.29-39, 2022. DOI: 0.33180/InfMIDEM2022.104

The spectrum license for Optical fiber is not required. License required only for signals radiated to air as they may create interference and can be hazardous for living (humans or animals). What may be needed for optical fiber is building permit to dig the roads, to lay down duct, tunnels, cable trays etc.

So instead of writing Spectrum license, I would write license and would mention what type of license is required.

 If the paper is accepted by the Academic Editor and the other reviewers, I definitively recommend updating the tables and chapters that discuss the microwave/millimeter-wave RF links. Actually, the highest QAM mentioned in the paper is 64-QAM. Please cite newer literature and recent papers too, where 256, 512, 1028 QAM or even higher are discussed. 5G backhaul uses Gbps links with adaptive modulation and wide RF bandwidth. Please see and cite papers like:

DOI: 0.33180/InfMIDEM2022.104 and DOI:10.3390/app9235240

Finally, when comparing the different solutions, or topologies, beside link capacities, throughput and delay the availability is also an essential parameter to compare. Please see a quick summary of end-to-end availability considerations in:

“Availability-Aware E-band Wireless Extension of Fiber-Access”, IEEE/IFIP Network Operations and Management Symposium, NOMS’2022, April 2022. DOI: 10.1109/NOMS54207.2022.9789802

In the table(s) recommended for comparison, the Authors could collect experimental availability figures if the references provide this information. If not, at least mentioning typical availability figures of optical fiber / FSO / and microwave links would help comparison. Just saying hybrid FSO+RF link has higher capacity than RF link is straightforward. A simple comparison could be added, how a standalone RF link availability is increased by a parallel FSO. How standalone FSO link availability is increased by adding RF link. See the simple calculations in DOI: 10.1109/NOMS54207.2022.9789802.

My recommendation is not to publish the paper in its actual form. I recommend a careful proof-reading. Several small errors can be quickly fixed to meet MDPI quality. Regarding the technical content -if Academic Editor and other Reviewers accept the paper for publishing- adding a comparison of the cited references in some tabular form, where the main technical parameters are summarized, would increase the value of the survey. In case of acceptance, the RF mmW link part shall be updated and up to date (new) references to be added. In Line 1, I recommend declaring that it is a survey paper. Based on above comments I recommend a major revision first. In the attached PDF I marked style/grammar comments yellow. Pink highlights show errors, that must be corrected before publishing.

Best Regards

2022-08-12

Author Response

Response to Reviewer 2 Comments

Point 1:     After a nice start and introduction, the paper turns into a bibliographic survey of several published papers. In this point the own result of the Authors is weak. There is no own calculation neither own measurement or experimental result presented. Even though the topic is very interesting, and the cited references are good papers, some novelty should be added, e.g., in form of a systematic and nice comparison of the cited literature. Regarding 5G and mmW radios, newer references must be added, cited information must be updated.

Unfortunately, the paper in its actual form follows a verbal summary of others’ work. I would definitively recommend adding some tables around the end of the paper or into the conclusions. Main parameters could be compared, or summarized, like below – just as an example table – to list the main engineering or experimental parameters and their value or value ranges.

This could make the paper shorter, e.g. Monte-Carlo or Matlab simulation is mentioned in each paragraph as one sentence. In a table like below, there could be a line “simulation or calculation method” and simply a listing Monte Carlo [21, 23, 55] Matlab [44, 99, 77] analytical calculation/ closed form equations [102, 103] no info given [3].

In my opinion the verbal summary of the papers could be shorter and such table(s) could reduce the total length of the paper. In its actual form the paper is rather a book chapter.

FSO part                         RF part                 papers                                      notes

______________________________________________________________________

Link length                     20 km                             100 m                             [1]. [22] and [145]       just example

Capacity                         more than 20 Gbps      3 kbit/s                          [121]                              table, add a 

                                        1-20 Gbps                                                                                                      better one

Delay                              10 ms                             no data available         [7, 14, 66]

(buffer)                          not used

BER

Wavelength /               1550 nm                        na                                    [11, 33, 99]

                                        800 nm                           na

Frequency                     na                                    2.4 GHz                          [55] and [73]

Relay type DF/DetFused/not used/AP

Relay position               Terrestrial / flying / satellite                              T: [1,2,3]

                                                                                                                        S :[6.16,26]                  

etc…

This is just an example, how the verbal comparison could be turned into engineer’s helping guide, to see where we can find the paper of our interest.

Similarly, different topologies are shown in separate figures, but they are spread over the paper. There is no quick comparison available who had loop, SIMO / MIMO, single link or protected link, diversity with/without, etc. As different papers demonstrated different topologies, the direct comparison of the results is difficult for the readers. They must remember and turn 15..20 pages back or forth. From a survey paper the reader expects that at the end there is some quick comparison of the 200+ references.

In these points I miss tables similar to Table 1, 2 or 3.

Response 1:          •Your comments are much appreciated and taken into consideration. The reference problem in the original file has been fixed (on line 376) in the revised manuscript (Page 8, line 356).
•The authors added a table to summarize and show features of all the models mentioned in section 5, as there are a lot of projects in this section; therefore, it will be easier for readers to compare them as recommended. (Page 24, line 1011) as shown in the screenshot below “The screenshot is only a part of the actual table”

•Table 5 helped a lot to remove a lot of repeated sentences in the manuscript such as “All simulations in this paper have been carried out using Monte-Carlo simulation ” on page 18, line 790 in the original manuscript.
•Also, the authors have removed all sentences that mention the fading distribution used in all the paragraphs and included them in table 5
•However, the authors have also added Table 4 on page 9, line 397 under section 4 (Multiplexing), which highlights the major differences between SISO, SIMO, MISO, and MIMO, as shown in the screenshot below.

• In addition, the authors have added tables 6 and 7 on page 43 to summarize all projects in section 7.

Point 2:     Text editing comments

Spelling / Grammar

Line 68, the paper has Section 3, but has not Section III.

Line 162, missing point

Line 354, water? matter?

Line 518, incomplete sentence

Figure 10, evesdropper is with double p

Line 804, Monte-Carlo

This is a general comment: Monte-Carlo shall be written with ‘-‘ all over the paper.

Line 880, Morraa or Morra? please check reference [160]

International Telecommunication Union is an institution, its name must be written with capitals in line 1494.

Response 2:          •Thank you for your response. Authors have changed Section III into Section 3 (Page 4, line 153)
•Added the full stop (Page 4, line 165)
•Changed mater molecules to water molecules “was a typo” (Page 7, line 334)
•Changed the sentence to “The hybrid SIMO-RF/FSO communication system is introduced by Shi et al. [115], who compare it with the conventional single input single output RF/FSO (SISO-RF/FSO) communication system, as shown in Figure 5” (Page 11, line 486)
•Wrote evesdropper instead of “evesdroper” in figure 10 (Page 17, line 728)
•Changed “Monte Carlo” to Monte-Carlo in the entire paper as it only appears in tables 5 and 6. (Page 24, line 1011) and (Page 43, line 1786)
•It's Morra, not Morraa “our apologies for the typo” (Page 19, line 802)
•We have written it as “International Telecommunication Union Radio” (Page 33, line 1399)

Point 3:     Acronyms

·      Some acronyms are not resolved, e.g.,

SAGIN in Figure 4.

NOMA in lines 656, 665, 670, 674, 1908,

PD in line 732

S (most probably Source / Sender, and not Signal) in line 437, 667

EDFA

CRZ

Response 3:          •Thank you for your comments, authors have changed SAGIN into “space-air-ground integrated network-based” in figure 4 (Page 11, line 485)
•Resolved Non-Orthogonal Multiple Access (NOMA) (Page 14, line 614)
•wrote PD as “photodetector” since we did not use PD many times (Page 16, line 674)
•We have changed S to source, R to relay and D to destination (Page 10, line 423-424)
•Moreover, we have written the full form of (EDFA) to Erbium-Doped Fiber Amplifier (Page 31, line 1255)
•and removed the first paragraph of section 7, which contains the unresolved CRZ

Point 4:     ·      Several acronyms are resolved two or more times, e.g.,

SIM in line 287, 453

PIN in line 248, 261

OFDM in lines 228 and 1872

MMW in 872 and several other lines

FSO in line 67 and 130

SMF

APD

Response 4:          •Your cooperation is highly appreciated! We have checked all of them and now are resolved once when are mentioned for the first apperance
•SIM (Page 6, line 280)
•PIN (Page 6, line 239)
•OFDM (Page 5, line 219)
•MMW (Page 8, line 365)
•FSO (Page 2, line 58)
•SMF (Page 19, line 820)
•APD (Page 6, line 240)

Point 5:     ·      Several acronyms are resolved lines after their first appearance, e.g.,

IMD used in line 301, resolved in line 302, 
BS, used in line 596, explained in line 657, 
D used in line 442, explained in line 524, 
OP used in line 448, explained in line 460, 
MRC used in line 500, 521 524, explained at 643, 
QAM used in line 472, explained in line 1018, 
SC used in 628 for selective combining, explained in 634,

Unluckily, SC used later for Small Cells also
SNR used in 265, 477, 644, etc… explained in line 1886,

CDF used in lines 756, 767, etc, explained in line 1322,

PDF used in lines 767, 820, 922, etc., explained in line 1322,

TWR used in line 776, explained in 929

WiFi is incorrectly written as WIFI in Lines 1221, 2024

mmWave, MMW, mm-Wave all mean the same. When it is not a citation, uniform writing is recommended all over the paper. (mmW in citation titles must keep its bibliographical form).

IPv4 and IPv6 are written IPV4 and IPV6 in Table 3.

As a general recommendation: I would add a small chapter with all acronyms at the end of the paper. Authors must list all abbreviations in alphabetical order, and check if they are referred in IEEE format.

Response 5:          •We would like to express our sincere gratitude for the effort you have paid to Give these comments. The authors have fixed all the mistakes.
•IMD (Page 7, line 297)
•BS (Page 8, line 375)
•We have changed S to source, R to relay and D to destination (Page 10, line 423-424)
•OP (Page 10, line 435)
•MRC (Page 11, line 470)
•QAM (Page 11, line 456)
•SC (Page 14, line 591)
•SNR (Page 6, line 255)
•CDF (Page 16, line 696)
•PDF (Page 16, line 706)
•TWR (Page 17, line 716)
•WiFi (Page 8, line 376) and written correctly all over the paper
•MMW (Page 8, line 359)
•IPv4 and IPv6 (Page 44, line 1826)
•Added table 9 at the end of the paper for a list of acronyms (Page 46, line 1925)

Point 6:     Acronyms must be explained at the first place of their appearance, not several lines later. Dual meaning acronyms are better to be avoided if possible, or explained in the list. Please see a good example list in one MDPI paper at:  https://www.mdpi.com/2076-3417/9/23/5240/htm

It is not the reviewer’s job to find out all the acronyms by careful reading of the manuscript and the cited references. This is a very long survey and it uses hundreds of abbreviations. In my opinion such a list may help both the writing and the reading of the paper. (The paper shall be preferably understandable without using Google or Wikipedia, the paper should help its readers.)

Response 6:          •We sincerely thank you for taking the time to leave these remarks. All errors have been corrected by the authors and added table 9 at the end of the paper for a list of acronyms (Page 46, line 1925)

Point 7:     SI units

mW is the correct writing for milliWatt

mw is incorrect at lines 1399, 1400

°C   Celsius must be written with degree sign  - Line 1515

Gbps is well known and used in English. If SI form is used, Gbit/s is recommended. At several places Gb/s is used. Except the citations, I recommend uniform writing of Gbps everywhere. The reason is simple, in several papers write GigaByte/second as GB/s, there is always a doubt if Authors follow the rule or not when abbreviating Bytes and bits as B and b, respectively. In case of Gbit/s or Gbps there is no doubt.

As a general rule, all SI units, and their prefixes must be correct, the paper cannot be published with units written incorrectly.

As a general rule, there is non-braking space between number and unit. 340Mbps in line 1712 is correctly written as 340 Mbps.

Response 7:          •Authors have corrected all of those mistakes. Our apologies for those mistakes.
•mW (Page 32, line 1311-1312) (Page 36, line 1504)
•°C (Page 34, line 1418)
•Gbps is used all over the paper instead of Gbit/s even in the new tables add such as table 5 (Page 24, line 1011)

Point 8:     References

Line 376 has Error! “Reference source not found” – the paper must not be published with this

Line 2146, Hecht: year of publishing of the book is missing

Line 2055, name of the Author starts with capital, please correct

Line 2155, title doubled

Line, 2196, use title case please, no reason to use all caps

Line, 2244 refers to a table of a paper of Aditi Malik and Preeti Singh, “Free Space Optics: Current Applications and Future Challenges”, Vol. 2015, Article ID 945483, DOI: 10.1155/2015/945483.

Table.2 of this article repeats Table 1 of reference [100], so please refer to the original paper, not only to the Table. Reference shall be given in title of Table 2 too in line 338 as:

“Table 2. Comparison of FSO with Different System [100]”.

Within the text Authors are sometimes referred with full name, sometimes with family names only, sometimes with shortened given name + family name e.g.:

Anirban Bhowal and Rakhesh Singh Kshetrimayum in [134], line 591

Amirabadi & Vakili in [156], line 828

V. U. Pai and B. Sainath in [180], line 1180

I recommend similar style of writing for the names everywhere across the paper.

Response 8:          •Thank you for your assistance cor reviewing this paper. I have fixed all of those mentioned above.
•The reference problem in the original file has been fixed (line 376) in the revised manuscript (Page 8, line 356).
•The year is added in “Hecht J. Understanding fiber optics. Published online 2015:790.” (Page 50, line 2042)
• “Zhang D, Zhou Z, Mumtaz SM, Rodriguez J, Sato T.” (Page 48, line 1948)
• “Sze SM, Ng KK. Physics of Semiconductor Devices. John Wiley & Sons, Inc.; 2006. doi:10.1002/0470068329.” (Page 50, line 2050)
• “Forrest SR. Optical detectors: three contenders. IEEE Spectr. 1986;23(5):76-84. doi:10.1109/MSPEC.1986.6370907” (Page 50, line 2079)
•Reference is added to the title of table 2 as you suggested (Page 8, line 357)
•Authors have Changed all references to “Lastname et al. [XX]” (Page 5, line 218)
• “Lastname in [XX]” for only one author (Page 8, line 359)
•and “1stLastName and 2ndLastName [XX]” if they are two authors (Page 14, line 614)

Point 9:     Positioning figures and comment to figures:

Normally text refers to a figure first, then figure placed after the paragraph referring to the figure. In lines 516-517 Figure 5 is before it is discussed in line 520.

Same comment for Figure 3. Figure 3 shows User X. text refers User K.

Same comment for Figure 11.

Figure 13 could indicate D and S. Data in should have input/ingress arrow.

Figure 16 could indicate D and S. Data in should have input/ingress arrow.

Figure 20 shows airplanes as relays. Line 1147, in the next chapter,  says relay nodes are stationery. What are C, D, E in Figure 20? Stationary or moving relays?

Response 9:          •Thank you for your cooperation! Authors have moved figure 20 and the respective paragraph to chapter 5.1. (Page 26, line 1051)
•All figures, such as 2, 3, 5, 6, 11, 17, 18, and 26 have been rearranged to appear after the respective paragraph.
• “Users K” is changed to “users X” in figure 3 (Page 10, line 437)
•Source and destination are added with input and output arrows to figures 13 and 16 (Page 18, line 792) (Page 21, line 894)
•Figure 20 has been moved to section 5.1 (Page 26, line 1050)
• “A and B are the sender/destination nodes. However, C, D and E are relay nodes in the communication system.” Is added to the paragraph to explain what are A, B, C, D and E (Page 26, line 1025,1038)

Point 10:  Tables

Table 3 is printed before its own chapter.

Response 10:        •Your efforts are highly appreciated! We have relocated the table and for all the figures, Table 3 is now table 8 “due to the amendments” on page 44, line 1826 right before section 8.1

Point 11:  Doubled paragraph/sentences:

Lines 103-114 write the same as lines 378-388.

Lines 396-399 are repeated in Lines 414-418.

Line 409 is repeated in Line 418

The paper is very long, it cannot be accepted in current form with repeated paragraphs.

Response 11:        •Noted with thanks! All repeated paragraphs and sentences are removed. Our apologies for these mistakes!
•The authors have gotten rid of the repetitive paragraph that was in section 3 because it was unnecessary. The paragraph is just included in the introduction (Page 3, line 100).
•lines 414-418 in the original manuscript are removed and only appear once in lines 387-390 the latest manuscript
•line 418 is removed (Line 390-391)

Point 12:  Exponents In technical text, exponents must not be written ‘in line’. BER of 10-9 means BER of 1.

As 10-9 = 1. Authors wanted to say 10-9, where -9 is superscript.

Response 12:        We have considered your comments and made the changes based on your comments. (Page 44, line 1843,1845)

Point 13:  Conflilct of interest declaration It is missing.

Response 13:        • Authors have added “Conflict of interest declatarion” before references (Page 48, line 1926-1940)

Point 14:  Technical comments – questions:

Figure 8 shows Full-Duplex system, however arrows are unidirectional. Why only uplink is shown? For a FD system I expect bidirectional arrows as in Fig.7.

This is a general comment to all arrows. Arrowheads show signal flow direction. Unidirectional links are one arrow ended lines. Bidirectional (duplex links) must have arrow at both ends.

Response 14:        •Your comment is highly appreciated! We have Drew full-duplex links to bidirectional arrows in figure 8 (Page 15, line 648) and stated that unidirectional arrows mean a one-way communication in line 410 on page 10.

Point 15:  Photodiode and photodetector are used interchangeably at several places. Strictly speaking, the photodiode must be a ‘diode’, having a p-n junction. p-i-n diodes have junction, they are real diodes, so we must be careful how to bias it. Photodetectors are rather circuits, that may contain a photodiode plus some optical and/or electrical circuitry, like focusing lens, pre-amplifier and/or electrical transimpedance amplifier, matching and biasing circuit. But not all photodetector is obviously a photodiode. Photodetectors can be built without diodes too (e.g., using bulk semiconductor, like a photoconductor, solar-cell or metal-semiconductor-metal construction, etc. whatever can detect a ‘photon’). I recommend using the terms correctly in part 2.3. PIN and APD are real diodes. Photoreceivers built with PIN or APD but contain more than the diode itself are photodetectors, or ‘optical receivers’.

Response 15:         •We would like to thank you for your assiatnce and cooperation in reviewing this paper. We have changed photodiode to photodetector in section 2.3 (page 6, line 237)

Point 16:  Internet access is bidirectional, in Figure 19 the vehicle has only downlink, the server has up and downlinks too. Why ?

Response 16:        • Noted with thanks! We have changed it to a bidirectional arrow instead of a unidirectional arrow in figure 19 (Page 24, line 985)

Point 17:  Technical errors:

The paper refers to 5G as ‘next’ , ‘it will provide’ (e.g. lines 26-27). Feel free to use present tense. 5G has been launched already at several countries. I would keep future tense and ‘next’ for 5G+ and 6G in the paper.

Same comment for MIMO, Line 403 says MIMO systems will be ideal…in fact MIMO is already widely used. It is not ‘will be’, it ‘is’ already…

Response 17:        •Thank you for your assistance! We have edited the paragraphs accordingly.
• “it will provide” changed to “provides” (Page 1, line 27)
 • “These characteristics of MIMO systems makes it ideal for modern communication technologies.” (Page 9, line 394)

Point 18:  Table 1 states that RF links have capacity less than 100 Mbps. Most probably Authors think legacy backbone links, typically built at 38/39 GHz or below.  But the table in its actual form is not correct. Table title compares “FSO and RF communications” in general, not specifically the access links. For example, 5G uses also RF for subscriber access and also for fixed access, and Gbps speed can be reached in 5G, also it has been experimentally verified. So table title instead of ‘RF communications’ should specify rather like ‘RF access links’ or rather ‘legacy RF access links’. Other way is to correct the 100 Mbps to actual value, e.g. 10 Gbps. Feel free to correct old value of reference [100] (2015) and cite paper below (2022). In line 50 Authors introduce RF band as 3 kHz to 300 GHz, that means RF is used in a sense of radio wave (i.e., not lightwave) communications. Nowadays Gbps RF links are used in backbone too. Please see e.g. “Throughput Estimation of K-zone Gbps Radio Links Operating in the E-band”, Informacije MIDEM, Journal of Microelectronics, Electronic Components and Materials, Vol.52, No.1, pp.29-39, 2022. DOI: 0.33180/InfMIDEM2022.104

Response 18:        •Your comment was very beneficial to me, we have changed table 1 title to “RF access links” (Page 3, line 129)
•and removed “The RF band spans the electromagnetic spectrum from 3 kHz to 300 GHz.” (Page 2, line 50)
•added “Fiber-optic access networks are well-suited to be extended by E-band radios (71-86 GHz). Hilt in [99] demonstrated that modern MMW radios can attain Gbps speed owing to the broad radio bandwidth made possible by the frequency allocation method [100]. The estimated lifespan of the E-band radio connections is 3-5 years. However, when optical cable reaches a radio node, the investment is not wasted.” (Page 8, line 358)

Point 19:  The spectrum license for Optical fiber is not required. License required only for signals radiated to air as they may create interference and can be hazardous for living (humans or animals). What may be needed for optical fiber is building permit to dig the roads, to lay down duct, tunnels, cable trays etc.

So instead of writing Spectrum license, I would write license and would mention what type of license is required.

Response 19:        •Noted with thanks! We have changed required to not required for Optical fiber in table 2 (Page 8, line 357)

Point 20:   If the paper is accepted by the Academic Editor and the other reviewers, I definitively recommend updating the tables and chapters that discuss the microwave/millimeter-wave RF links. Actually, the highest QAM mentioned in the paper is 64-QAM. Please cite newer literature and recent papers too, where 256, 512, 1028 QAM or even higher are discussed. 5G backhaul uses Gbps links with adaptive modulation and wide RF bandwidth. Please see and cite papers like:

DOI: 0.33180/InfMIDEM2022.104 and DOI:10.3390/app9235240

Response 20:      •Noted with thanks!
•Authors have added “On the other hand, Hilt in [213] confirms that different modulation types have different thresholds for radio networks employing adaptive modulation. When the fading margin makes up for the loss brought on by rain, the low modulation mode is used during wet minutes. During sunny periods when the fade margin is available, the link can jump to higher modulation modes up to 512-QAM; thus, more data can be transmitted [214], [215]. Results in [213] only display QAM up to 512 for simplicity. However, modern transceivers may achieve several Gigabit/s data speeds by using up to 4096-QAM.” (Page 36, line 1518-1528)

Point 21:  Finally, when comparing the different solutions, or topologies, beside link capacities, throughput and delay the availability is also an essential parameter to compare. Please see a quick summary of end-to-end availability considerations in:

“Availability-Aware E-band Wireless Extension of Fiber-Access”, IEEE/IFIP Network Operations and Management Symposium, NOMS’2022, April 2022. DOI: 10.1109/NOMS54207.2022.9789802

In the table(s) recommended for comparison, the Authors could collect experimental availability figures if the references provide this information. If not, at least mentioning typical availability figures of optical fiber / FSO / and microwave links would help comparison. Just saying hybrid FSO+RF link has higher capacity than RF link is straightforward. A simple comparison could be added, how a standalone RF link availability is increased by a parallel FSO. How standalone FSO link availability is increased by adding RF link. See the simple calculations in DOI: 10.1109/NOMS54207.2022.9789802.

My recommendation is not to publish the paper in its actual form. I recommend a careful proof-reading. Several small errors can be quickly fixed to meet MDPI quality. Regarding the technical content -if Academic Editor and other Reviewers accept the paper for publishing- adding a comparison of the cited references in some tabular form, where the main technical parameters are summarized, would increase the value of the survey. In case of acceptance, the RF mmW link part shall be updated and up to date (new) references to be added. In Line 1, I recommend declaring that it is a survey paper. Based on above comments I recommend a major revision first. In the attached PDF I marked style/grammar comments yellow. Pink highlights show errors, that must be corrected before publishing.

Response 21:        •Noted on your comment and we took it into account. We would like to finally thank you for the effort you have paid to increase the quality of this survey. It was very beneficial on my end to enhance our writing skills.
•Authors have added “Nevertheless, the findings presented in [218] make it abundantly evident that raising the state of QAM modulation mode not only enhances the throughput but also reduces the amount of power it needs to operate.” (Page 36, line 1525-1528)

Reviewer 3 Report

Dear authors,

The authors presented the results of an extremely extensive review of projects related to hybrid communication systems using free-space optics (FSO) and radio frequency (RF), which are dedicated to obtain connectivity in 5G networks.

The motivation and contribution are clearly presented. FSO performance compared to other communication methods is included. The multiplexing technique used has been carefully described.

Communication networks using relay nodes are discussed. An overview of the various hybrid FSO systems and projects using only FSO links is provided.

It is worth noting that References contains 250 items. The items listed are aptly selected and are up-to-date.

Remarks:

1.       The article probably contains valuable conclusions drawn from the work of other scientists, but within the chapters there is no binding conclusion, no attempt to compare the solutions.

2.       5G networks are based on small cells. Maybe it is worth to clearly define (maybe in the table) what range values for the FSO / RF and FSO connection are satisfactory. What does short-range and long-range mean ?

3.          In Chapter 3 (OWC Systems for 5G Backhauling Network), in line 376 has a reference recall error.

Author Response

Response to Reviewer 3 Comments

Dear authors,

The authors presented the results of an extremely extensive review of projects related to hybrid communication systems using free-space optics (FSO) and radio frequency (RF), which are dedicated to obtain connectivity in 5G networks.

The motivation and contribution are clearly presented. FSO performance compared to other communication methods is included. The multiplexing technique used has been carefully described. 

Communication networks using relay nodes are discussed. An overview of the various hybrid FSO systems and projects using only FSO links is provided. 

It is worth noting that References contains 250 items. The items listed are aptly selected and are up-to-date.

Remarks: 

Point 1:     The article probably contains valuable conclusions drawn from the work of other scientists, but within the chapters there is no binding conclusion, no attempt to compare the solutions.

Response 1:      •Your comments are much appreciated and taken into consideration.
The authors added table 5 to summarize and show the features of all the models mentioned in section 5, as there are a lot of projects in this section; therefore, it will be easier for readers to compare them as recommended. (Page 24, line 1011) as shown in the screenshot below

•Table 5 helped a lot to remove a lot of repeated sentences in the manuscript such as “All simulations in this paper have been carried out using Monte-Carlo simulation ” on page 18, line 790 in the original manuscript.
•Also, the authors have removed all sentences that mention the fading distribution used in all the paragraphs and included them in table 5

Point 2:     5G networks are based on small cells. Maybe it is worth to clearly define (maybe in the table) what range values for the FSO / RF and FSO connection are satisfactory. What does short-range and long-range mean ?

Response 2:      •Thank you for your response.
•Authors have added “Most FSO links are between 300 m and 5 km, although depending on the data rate and availability needed, greater lengths like 8-11 km may be implemented [42]. On the contrary, with a power of 1 W, the typical RF link ranges with a frequency range of 400 MHz are up to 30 km, and with a power of 10 W, up to 80 km [43].” Right before table 1 (Page 3, line 125)
•Moreover, “Small cells resemble WiFi in many ways, but they can use cellular standards, unlike WiFi. Depending on the cell's bandwidth, maximum output power, coverage area, user count, deployment scenario, and connection to the backhaul, small cells can be categorised as femto, pico, or micro. Table 3 below compares small cells with macrocells.” Is added with table 3 as shown in the screenshot below (Page 9, line 381)

Point 3:     In Chapter 3 (OWC Systems for 5G Backhauling Network), in line 376 has a reference recall error.

Response 3:      Thank you for your kind review. The reference problem in the original file has been fixed (line 376) in the revised manuscript (Page 8, line 356).

Round 2

Reviewer 1 Report

I'm happy to see that the revised manuscript is significantly improved over the original submission. I can recommend the paper for publication after correction of some minor issues:

-Author affiliations are missing

line 124 - still has full caption in text

125 - maybe add "terrestrial", it's in the caption but should also be in text

Table 1 - "Superiority" entries are flipped?

148 - "LOS" appears twice?

188 - move the reference number to after the authors names, not end of sentence

203 - "collimated" is better word than "focused"

212 - still don't know why quality for illuminating purposes is mentioned

240 -  "handles the..." can be removed

244 - "products are only two types", this sentence is unclear

277 - Should be new paragraph?

286 - remove "assist"

326 - missing "Table 2" at start of sentence

339 -  doesn't quite make sense

364-366 why is this reference mentioned? it doesnt say anything about it

551 - missing author name

655-656 "milli" and "micro" shouldn't be used since there are no units. Just use scientific notation.

Figure 10 caption needs more info

864 - delete word "authors"

Figure 17 caption has much more detail than other figures. Be consistent

1242 there's a "." where it shouldn't be

Table 6 and 7, is this information relevant?

1795, accidental new paragraph

1863-1874 repeated paragraph is the same as lines 826-838 in section 5

1898-1902 English needs to be fixed up

1913 - "atmospheric turbulence" appears twice

I would also strongly suggest another thorough proof-reading to catch any other minor issues I may have missed.

Author Response

Response to Reviewer 1 Comments

Point 1:     I'm happy to see that the revised manuscript is significantly improved over the original submission. I can recommend the paper for publication after correction of some minor issues:

Response 1:          •Your comments are much appreciated and taken into consideration.
All changes this time are highlighted in red

Point 2:     -Author affiliations are missing

Response 2:          •Thank you for your response. Authors have added affiliations as shown in the screenshot below

Point 3:     line 124 - still has full caption in text

Response 3:          Appologies for this mistake,
“To sum up, Table 1 summarizes the main differences between FSO and RF communication systems [39]–[41].” (Page 3, line 123-125)

Point 4:     125 - maybe add "terrestrial", it's in the caption but should also be in text

Response 4:          Thank you for your comment and helpful suggestion! “Most terrestrial FSO links are added between 300 m and 5 km” (Page 3, line 125 )

Point 5:     Table 1 - "Superiority" entries are flipped?

Response 5:          Thank you for your helpful comment! As shown in the screenshot, the entries are rewritten. (Page 3 , line 129 )

Point 6:     148 - "LOS" appears twice?

Response 6:          Thank you for your comment, we have considered the comment and fixed the sentence to a correct one.
“The terms line-of-sight (LOS), directed LOS, non-directed LOS and point-to-point (P2P)” (Page 4, line 147,148)

Point 7:     188 - move the reference number to after the authors names, not end of sentence

Response 7:          Thank you for your comment! We have moved the reference number after the author’s name. “Kaushal and Kaddoum [27]” please refer to (Page 5, line 187).

Point 8:     203 - "collimated" is better word than "focused"

Response 8:          Thank you for your comment and suggestion! We have replaced the word “ collimated instead of focused. “laser light is collimated and directed forward” Please refer to (Page 5, line 203).

Point 9:     212 - still don't know why quality for illuminating purposes is mentioned

Response 9:          Thank you for your comment. We have deleted “and the uncertain quality of laser light for illuminating purposes”. Please refer to (Page 5, line 211,212)

Point 10:  240 -  "handles the..." can be removed

Response 10:        Thank you for your comment and suggestion. We have removed “handles the photodetector's detecting function”. Please refer to (Page 6, line 239).

Point 11:  244 - "products are only two types", this sentence is unclear

Response 11:        Thank you for your comment. We have considered your comment and changed the sentence to “Only two varieties are accessible because of their good quantum efficiency, semiconductor design, and widespread availability in commercial-off-the-shelf (COTS) [74].”Please refer to (Page 6, line 240-242).
This sentence refers to APD and PIN photodiodes

Point 12:  277 - Should be new paragraph?

Response 12:        Thank you for your comment. We have started a new paragraph from “Both OOK and PPM are single-carrier”.
Please refer to (Page 6, line 276)

Point 13:  286 - remove "assist"

Response 13:        Thank you for your helpful comment. We have removed “assist”. Therefore, the sentence became “SIM approaches alleviate channel impairments and enable a more superficial and cost-effective implementation than single-carrier modulation schemes [81]” Please refer to (Page 15, line 284-286)

Point 14:  326 - missing "Table 2" at start of sentence

Response 14:        Thank you for your comment, we have added“Table 2” at the begging of the sentence. Please refer to (Page 7, line 325).

Point 15:  339 -  doesn't quite make sense

Response 15:        Thank you for your comment, we have changed the sentence to “FSO systems also have the risk of scattering which happens when the optical beam collides with the scatterer. As a result, it reduces the beam’s intensity for a longer distance.” Kindly refer to (Page 8, line 338).

Point 16:  364-366 why is this reference mentioned? it doesnt say anything about it

Response 16:        Thank your for your comment and fruitful suggestion! The paragraph has been changed to “Ge et al. [103] analyze wireless backhaul networks by using small cell, millimeter-wave (MMW) and Multiple-Input Multiple-Output (MIMO) communication technologies to attain Gigabit transmission rates in 5G networks strategies for promoting 5G wireless backhaul networks in a way that uses low energy and offers high throughput. The wireless backhaul traffic in upcoming 5G+ networks is set to be examined using two common small cell scenarios. Furthermore, two common small cell scenarios are used to examine the energy efficiency of wireless backhaul networks. According to numerical findings, in 5G wireless backhaul networks, the distribution solution uses less energy than the central solution. If the new distribution network design is used in the next 5G wireless backhaul networks, a real difficulty would materialize.”Kindly refer to (Page 8, line 362-372).

Point 17:  551 - missing author name

Response 17:        Thank you for your comment! “Amirabadi and Vakili” is added before the reference. Please refer to (Page 13, line 556)

Point 18:  655-656 "milli" and "micro" shouldn't be used since there are no units. Just use scientific notation.

Response 18:        Thank you for your comment! Milli and micro are changed to “For instance, the OP at 40 dB of SNR is 2.3x10-6 compared with the single RF/FSO system is 3x10-3” Kindly refer to (Page 16, line 660).

Point 19:  Figure 10 caption needs more info

Response 19:        Thank you for your comment! The caption of figure 10 has been changed to “Combined RF-FSO DF-based relaying system” Kindly refer to (Page 17, line 734).

Point 20:  864 - delete word "authors"

Response 20:        Thank you for your comment!“authors” is deleted. Therefore the sentence became “Jiang et al. [165] propose an end-to-end hybrid FSO/RF system integrated with PPM” Please refer to(Page 21, line 869)

Point 21:  Figure 17 caption has much more detail than other figures. Be consistent

Response 21:        Thank you for your comment! The caption of figure 17 have been changed to “RF-FSO/RF communication system”
Kindly refer to (Page 22, line 930).

Point 22:  1242 there's a "." where it shouldn't be

Response 22:        Thank you for your comment. We have fixed the sentence. Hence, it became “Fine tracking is essential for this project as the signal transmitted from the beam splitter has to be directed to a quadrant photodetector (QPD) to implement the system” Kindly refer to (Page 31, line 1244-1246)

Point 23:  Table 6 and 7, is this information relevant?

Response 23:        Thank you for your comments! Both tables helped a lot to remove a lot of repeated sentences in the manuscript such as “All simulations have been carried out using Monte-Carlo simulation”, “Matlab is used to verify all the results/simulation” and “RF/ link follows XX fading distribution”

Point 24:  1795, accidental new paragraph

Response 24:        Thank you for your comment! The accidental new paragraph has been fixed. Please refer to (Page 43, line 1798).

Point 25:  1863-1874 repeated paragraph is the same as lines 826-838 in section 5

Response 25:        Thank you for your comment! The repeated paragraph has been removed (Page 45, line 1866)

Point 26:  1898-1902 English needs to be fixed up

Response 26:        Thank you for your comment! The whole paragraph has been changed to “In addition, external modulation is superior to direct modulation. For instance, return to zero (RZ) modulation is suitable for long-distance communication but is both difficult and expensive. NRZ, however, is more appropriate for short links, simpler, and more affordable. As stated in Table 8, the model must meet the criteria for classification as a 4G, 5G, or 6G system.” Kindly refer to (Page 45, line 1889).

Point 27:  1913 - "atmospheric turbulence" appears twice

Response 27:        Thank you for your comment! The repeated “atmospheric turbulence” has been removed
The sentence became “The FSO system is susceptible to many atmospheric phenomena, including absorption, scattering, and atmospheric turbulence” . Kindly refer to (Page 45, line 1903,1904).

Point 28:  I would also strongly suggest another thorough proof-reading to catch any other minor issues I may have missed.

Response 28:        • We would like to take the opportunity to thank you for your comments and the time you have taken as well as effort you have paid to give this feedback on the submitted manuscript. Another proof-reading is done and fixed a lot of minor issues
The authors strongly believe that your comments and review helped us to restructure the paper in a better shape.

Reviewer 2 Report

Dear Editor and Authors,

in general the second version of the paper is better than the first version was. New tables and list of acronyms help the reading, understanding and comparison of the huge amount of cited references. The technical part is improved, e.g. severalfigures are better or completed as requested.

But in my opinion a strict and careful revision was still missing, the proof reading was weak. There are still basic editorial mistakes in the paper:
-missing figure numbers at citations (see e.g., lines 743 and 1035). 
-in line 124 Table 1 is referred together with its very long caption that results difficulty in reading. Reader believes caption is part of the sentence.
-the new tables do not follow the MDPI table style. They look like ugly copy from MS Excel, even the font types and sizes are different in the coloumns.
- top line of tables should use bold characters.
-Authors use 'On the other hand' and 'However' too often as a gap filler. As the paper is very long, there is absolutely no need for gap-filling. When 'However' is used in consequtive sentences, it is simply not nice (992-993 lines). 

There are several mistakes that can be quickly corrected, e.g. plural noun and plural verb or singular noun with singular form of the verb. Some sentences are incomplete or verb is missing (e.g. line 326). There are multiple capital vs. small case mixings. Like Earth and earth may mean different in technical text. 'Earth' is the globe, 'earth' with small case is the ground in circuits, having zero voltage. Correcting these could improve the style of the paper to fit into a professional journal like MDPI Entropy.

Some sentences are half paragraph long...it is hard to follow what Authors would like to say in such ling sentences. In general, it is better to chop very long sentences into shorter ones having clear message.

I marked with red bubbles and added pop-up notes into the PDF. There may be misunderstandings on my side, as the paper is not an easy reading. Such sentences like line 350, 880-882 or 815 are not clear or misunderstandable. There are so many comments, so sorry I do not list them all here. They are marked in the PDF I attach to this review. (Please ignore the green color highlights)

In a few cases I understand what the authors wanted to say, but strictly speaking in English they do not write down correctly what they wanted to say. See for example: 1007-1008 "Moreover, if DC bias is added by the laser diode to avoid any negative values in the optical signals where the DC bias is subtracted at the receiver side to retrieve the original signal". In IM/DD the optical signal has intensity (optical power) that is always positive. The laser diode bias current could be negative, so harmful, but the bias current is an electrical signal, not an optical one. Such sentences could be simply deleted to make the paper shorter. Or if the sentence is kept, then should be rephrased with engineer's accuracy.

I recommend fixing these in the paper before publishing in MDPI.

Best Regards

Author Response

Response to Reviewer 2 Comments

Point 1:     in general the second version of the paper is better than the first version was. New tables and list of acronyms help the reading, understanding and comparison of the huge amount of cited references. The technical part is improved, e.g. several figures are better or completed as requested.

Response 1:          •Your comments are much appreciated and taken into consideration which resulted in the better version of the resubmitted manuscript. All changes this time are highlighted in red

Point 2:     But in my opinion a strict and careful revision was still missing, the proof reading was weak. There are still basic editorial mistakes in the paper: 
-missing figure numbers at citations (see e.g., lines 743 and 1035). 
-in line 124 Table 1 is referred together with its very long caption that results difficulty in reading. Reader believes caption is part of the sentence.
-the new tables do not follow the MDPI table style. They look like ugly copy from MS Excel, even the font types and sizes are different in the coloumns.
- top line of tables should use bold characters..
-Authors use 'On the other hand' and 'However' too often as a gap filler. As the paper is very long, there is absolutely no need for gap-filling. When 'However' is used in consequtive sentences, it is simply not nice (992-993 lines). 

Response 2:          •Thank you for your comment after reviewing the paper,
Figure number at citations have been added for figure 11 (Page 17, line 744)
and figure 20 (page 25, line 1029)
• The caption of table one is removed in (Page 3, line 124)
• All the tables now follow the MDPI table style using the same font type and size
•”However” in line 986 “latest version” has been changed to “Nevertheless” and
“On the other hand” in line 893 “latest version” has been changed to “An another approach”

Point 3:     There are several mistakes that can be quickly corrected, e.g. plural noun and plural verb or singular noun with singular form of the verb. Some sentences are incomplete or verb is missing (e.g. line 326). There are multiple capital vs. small case mixings. Like Earth and earth may mean different in technical text. 'Earth' is the globe, 'earth' with small case is the ground in circuits, having zero voltage. Correcting these could improve the style of the paper to fit into a professional journal like MDPI Entropy.

Response 3:          •Your comment is much appreciated! Authors have added all the missing verbs and nouns such as “Table 2” in line 325 and “present” in line 981 for example
We strongly believe that your comments and review helped us to restructure the paper in a better shape.

Point 4:     Some sentences are half paragraph long...it is hard to follow what Authors would like to say in such ling sentences. In general, it is better to chop very long sentences into shorter ones having clear message.

Response 4:          •Thank you for your comments. Another proof-reading is done and fixed a lot of minor issues

Point 5:     I marked with red bubbles and added pop-up notes into the PDF. There may be misunderstandings on my side, as the paper is not an easy reading. Such sentences like line 350, 880-882 or 815 are not clear or misunderstandable. There are so many comments, so sorry I do not list them all here. They are marked in the PDF I attach to this review. (Please ignore the green color highlights)

In a few cases I understand what the authors wanted to say, but strictly speaking in English they do not write down correctly what they wanted to say. See for example: 1007-1008 "Moreover, if DC bias is added by the laser diode to avoid any negative values in the optical signals where the DC bias is subtracted at the receiver side to retrieve the original signal". In IM/DD the optical signal has intensity (optical power) that is always positive. The laser diode bias current could be negative, so harmful, but the bias current is an electrical signal, not an optical one. Such sentences could be simply deleted to make the paper shorter. Or if the sentence is kept, then should be rephrased with engineer's accuracy.

Response 5:          •Your coordination is much appreciated!
Authors have gone through the paper and have edited all the comments highlighted in the PDF that you have marked
• For this comment “is there difference between Na and NDA?”
In lines 1003 and 1004, we have stated “NDA and NA respectively resulted to no data available and not applicable.” For example:
Project X is using only one hop of FSO link, therefore NA will be written in “RF link distance”, ”RF frequency” and “Relay type”
